# LOGLO-FNO: Efficient Learning of Local and Global Features in Fourier Neural Operators

**Marimuthu Kalimuthu**                                      *firstname.lastname@ki.uni-stuttgart.de*
*Universität Stuttgart, Stuttgart Center for Simulation Science, SimTech*
*International Max Planck Research School for Intelligent Systems (IMPRS-IS)*

**David Holzmüller**                                           *firstname.holzmuller@inria.fr*
*INRIA Saclay, SODA Team*

**Mathias Niepert**                                           *firstname.lastname@ki.uni-stuttgart.de*
*Universität Stuttgart, SimTech, IMPRS-IS*

**Reviewed on OpenReview:** *https://openreview.net/forum?id=MQ1dRdHTpi*

## Abstract

Modeling high-frequency information is a critical challenge in scientific machine learning. For instance, fully turbulent flow simulations of the Navier-Stokes equations at Reynolds numbers 3500 and above can generate high-frequency signals due to swirling fluid motions caused by eddies and vortices. Faithfully modeling such signals using neural networks depends on the accurate reconstruction of moderate to high frequencies. However, it has been well known that deep neural nets exhibit the so-called spectral or frequency bias towards learning low-frequency components. Meanwhile, Fourier Neural Operators (FNOs) have emerged as a popular class of data-driven models in recent years for solving Partial Differential Equations (PDEs) and for surrogate modeling in general. Although impressive results have been achieved on several PDE benchmark problems, FNOs often perform poorly in learning non-dominant frequencies characterized by local features. This limitation stems from the spectral bias inherent in neural networks and the explicit exclusion of high-frequency modes in FNOs and their variants. Therefore, to mitigate these issues and improve FNO's spectral learning capabilities to represent a broad range of frequency components, we propose two key architectural enhancements: (i) a parallel branch performing local spectral convolutions and (ii) a high-frequency propagation module. Moreover, we propose a novel frequency-sensitive loss term based on radially binned spectral errors. This introduction of a parallel branch for local convolutions reduces the number of trainable parameters by up to 50% while achieving the accuracy of the baseline FNO that relies solely on global convolutions. Moreover, our findings demonstrate that the proposed model improves the stability over longer rollouts. Experiments on six challenging PDE problems in fluid mechanics, wave propagation, and biological pattern formation, and the qualitative and spectral analysis of predictions, show the effectiveness of our method over the state-of-the-art neural operator families of baselines.

## 1 Introduction

Simulating real-world physical systems and problems in thermodynamics, biology, weather and climate modeling, hydrodynamics, and astrophysics, to name a few, involves solving partial differential equations (PDEs). Numerical PDE solvers based on Finite Volume, Finite Difference, and Finite Element methods might be slow due to the need for fine discretization of the computational mesh, work only for a given set of input parameters, and must be run from scratch when the settings, such as initial and boundary conditions, change. In recent years, neural networks have been used to build generalizable surrogate models

of such dynamical systems (i.e., forward problem) and for the inverse task of the discovery of PDEs and their parameters from data (Brunton & Kutz, 2023; 2024).

Effective modeling of the entire frequency spectrum, including high frequencies, is important for tasks such as super-resolution and turbulence modeling (Wang et al., 2020; Fan et al., 2024). In the latter case, large eddies can give rise to small eddies, resulting in chaotic dynamics, which can further be influenced by forcing functions (Fan et al., 2024). Large Eddy Simulation (LES) and Direct Numerical Simulation (DNS) require prohibitive costs for high-resolution scenarios (e.g., 2048×2048; Tran et al. (2023); Fan et al. (2024)). Neural Operators (NOs) are a class of surrogate models that can be trained at low spatial and temporal resolutions, saving cost, time, and compute, and can be used to predict solutions at resolutions unobserved during training, a technique called zero-shot super-resolution. Although the predominant neural operators exhibit this property, their accuracy is limited on simulation tasks that exhibit fine-grained details and complex structures. For instance, the Navier-Stokes equation exhibits high frequencies at low viscosities (i.e., $\nu = 1e^{-4}$) or high Reynolds numbers (e.g., $Re = 5000$; Li et al. (2022b)). NOs suffer from the spectral bias of neural nets and have difficulty modeling high frequencies and local features. We review related literature concerning this challenge in Appendix A. Motivated by the limitations of existing NOs and inspired by multipath neural net architectural designs (Chi et al., 2020; Chen et al., 2021; Pan et al., 2022; Li et al., 2024; Liu-Schiaffini et al., 2024), we propose LOGLO-FNO with the following key contributions: (i) local spectral convolutions, (ii) a high-frequency propagation component, and (iii) a novel frequency-aware loss term.

## 2 Background on Neural Operators (NO)

Neural Operators learn an approximation of operators between infinite-dimensional Banach spaces of functions. A neural operator is a function $\mathcal{G} : \mathcal{A} \to \mathcal{U}$ between two Banach spaces of functions $\mathcal{A}, \mathcal{U}$ defined on bounded domains.

**Fourier Neural Operator (FNO).** FNO (Li et al., 2021; Kossaifi et al., 2024b), which is one of the effective instantiations of a *NO*, learns the operator $\mathcal{G}$ using the Discrete Fourier Transform (DFT). In compact terms, $u = \mathcal{G}(a) = \left( \mathcal{Q} \circ \mathcal{L}^L \circ \ldots \circ \mathcal{L}^1 \circ \mathcal{P} \right)(a)$, with the lifting and projection layers $\mathcal{P}, \mathcal{Q}$ and the function composition $\circ$. A Fourier layer $\mathcal{L}$ can mathematically be described as the mapping in Eqn. 1, with $\mathbf{Z} \in \mathbb{R}^{\cdots \times N_c \times N_x \times N_y}$ being the output of the *lifting* operation for the first Fourier layer and the previous Fourier layer outputs thereafter,

$$\mathcal{L}(\mathbf{Z}) := \sigma\Big(\text{ChannelMLP}^1(\mathbf{Y}_{global}) + \text{SoftGating}^1(\mathbf{Z})\Big), \quad \text{where} \quad \mathbf{Y}_{global} := \sigma\Big(\mathcal{K}_g(\mathbf{Z}) + \text{Conv1D}^1(\mathbf{Z})\Big) \quad (1)$$

where $\mathcal{K}_g(\cdot)$ is the global kernel integral operator realizing DFT with Fast Fourier Transforms (FFT) for uniformly discretized data, $\sigma$ is the GELU non-linearity[*] acting point-wise, SoftGating operation applies a learnable, feature-wise affine or linear transformation to its input, Conv1D represents $1 \times 1$ convolution, $N_x \times N_y$ is the spatial resolution of the 2D spatial data, and $N_c$ is the width or the number of hidden channels.

FNO and its variants, barring incremental FNO (George et al., 2024), retain only a fixed, albeit tunable, number of frequency modes corresponding to low frequencies and truncate the high-frequency ones (Kovachki et al., 2023; Helwig et al., 2023; Brandstetter et al., 2023; Gupta & Brandstetter, 2023). This modeling choice, although beneficial in managing the model parameters, results in suboptimal reconstruction quality of predictions since it pushes the model to ignore high frequencies in the data. This is inconsequential when the data in question contains mainly low-frequency structures. However, high-fidelity reconstruction is paramount for tasks such as turbulence modeling (e.g., LES) and resolving multiple scales in multiphysics simulations. For instance, Hassan et al. (2023) benchmark FNO and its variants on the BubbleML dataset, a multiphase and multiphysics simulation of boiling scenarios. In such cases, the temperature profile can exhibit sharp jumps, resulting in discontinuities along the bubble interfaces, a typical manifestation of high frequencies. Their study has observed that FNO variants face significant difficulties in this task.

---

[*]skipped for the last Fourier layer in the network.

# 3 Method

In this section, we describe our main contributions. The LOGLO-FNO model aims to improve the class of Fourier Neural Operators (Li et al., 2021; Kossaifi et al., 2024a;b; Tran et al., 2023) in boosting their capacities to effectively learn local patterns and non-dominant frequencies. Such a model supports high-fidelity outputs and achieves improved stability over long rollouts (McCabe et al., 2023). Towards this end, LOGLO-FNO introduces two key modules as architectural enhancements, namely (i) a local spectral convolution module and (ii) a high-frequency propagation module. Moreover, we propose a frequency-aware spectral space loss function based on radially binned spectral errors (see Appendix D.2 for the full algorithm and pseudocode), using which we conduct our experiments. In addition, we explore an additional loss, namely, spectral patch high-frequency emphasized residual energy (see Appendix D.1), in line with the idea of patching the spatial domain to model local features. Furthermore, we employ attention-based fusion strategies in LOGLO-FNO to seamlessly integrate multi-scale features, and assess the effect of such a feature (Appendix G.3).

**LOGLO-FNO.** Our proposed LOGLO-FNO model modifies Eqn. 1 to Eqn. 2: the LOGLO Fourier layer takes three input tensors. (i) $\mathbf{Z} \in \mathbb{R}^{\cdots \times N_c \times N_x \times N_y}$, (ii) $\hat{\mathbf{Z}} \in \mathbb{R}^{\cdots \times N_c \times \hat{N}_x \times \hat{N}_y}$ being the outputs of the *lifting* operation $\mathcal{P}$ for the first global and local Fourier layers, respectively, and the previous LOGLO Fourier layer outputs thereafter, and (iii) $\mathbf{Z}' \in \mathbb{R}^{\cdots \times N_c \times N_x \times N_y}$ is the *lifted* features of the extracted high frequency structures $\mathbf{X}_H$ (cf. Eqn. 4) for the first LOGLO Fourier layer and the previous channel MLP outputs thereafter.

$$
\begin{aligned}
\mathbf{Y}_{global} &:= \sigma\Big(\mathcal{K}_g(\mathbf{Z}) + \text{Conv1D}^1(\mathbf{Z})\Big), \qquad \mathbf{Y}_{local} := \sigma\Big(\mathcal{K}_l(\hat{\mathbf{Z}}) + \text{Conv1D}^1(\hat{\mathbf{Z}})\Big), \\
\mathcal{L}(\mathbf{Z}, \hat{\mathbf{Z}}, \mathbf{Z}') &:= \sigma\Big(\text{ChannelMLP}^1(\mathbf{Y}_{global}) + \text{SoftGating}^1(\mathbf{Z}) + \\
&\qquad \text{ChannelMLP}^1(\mathbf{Y}_{local}) + \text{SoftGating}^1(\hat{\mathbf{Z}}) + \text{ChannelMLP}^1(\mathbf{Z}')\Big),
\end{aligned}
\tag{2}
$$

where the suffixes $g$ and $l$ denote global and local Fourier kernels, respectively, and $\hat{N}_x \times \hat{N}_y$ the spatial resolution of a 2D patch. The notations of the input tensors naturally extend to 3D spatial data.

The LOGLO-FNO architecture, depicted in Figure 1, maintains all the essential properties of NOs, such as discretization invariance. In addition to the main *global* branch, it consists of two auxiliary branches: (i) one parallel *local* branch that retains all Fourier modes and (ii) another parallel high-frequency feature propagation branch.

## 3.1 Architectural Improvements for FNO

**Local Spectral Convolution Branch.** We propose employing auxiliary local branches to the global Fourier layer to enable the model to learn rich local features by performing convolutions restricted to local regions (e.g., patches). Unlike the global branch, the local one operates on patches of the input signal. First, we create non-overlapping patches similar to the input of prototypical vision transformers (Dosovitskiy et al., 2021). These are then fed to the *lifting* layer $\mathcal{P}$ before being passed on to the local spectral convolution layer, which retains all Fourier modes. Akin to a CNN, the local branch operating on patches performs local convolution since the spatial domain is now limited to the patch size, capturing local and small-scale patterns, whereas the main branch performs global convolution, modeling high-level phenomena (e.g., overall fluid flow direction from left to right or bottom to top).

We consider a partition of the domain $D$ into non-overlapping hypercubes $P_1, \ldots, P_M$ called patches, such that $\bigcup_{m=1}^{M} P_m = D$. Then, the local kernel integral operator $\mathcal{K}_l(\cdot)$ with a learnable $\phi$ can be defined as

$$
(\mathcal{K}_l(\phi)\hat{Z}_{\hat{\mathcal{L}}^n})(x) = \int_{P_{m(x)}} \kappa_\phi(x - y)\hat{Z}_{\hat{\mathcal{L}}^n}(y)\,dy,
\tag{3}
$$

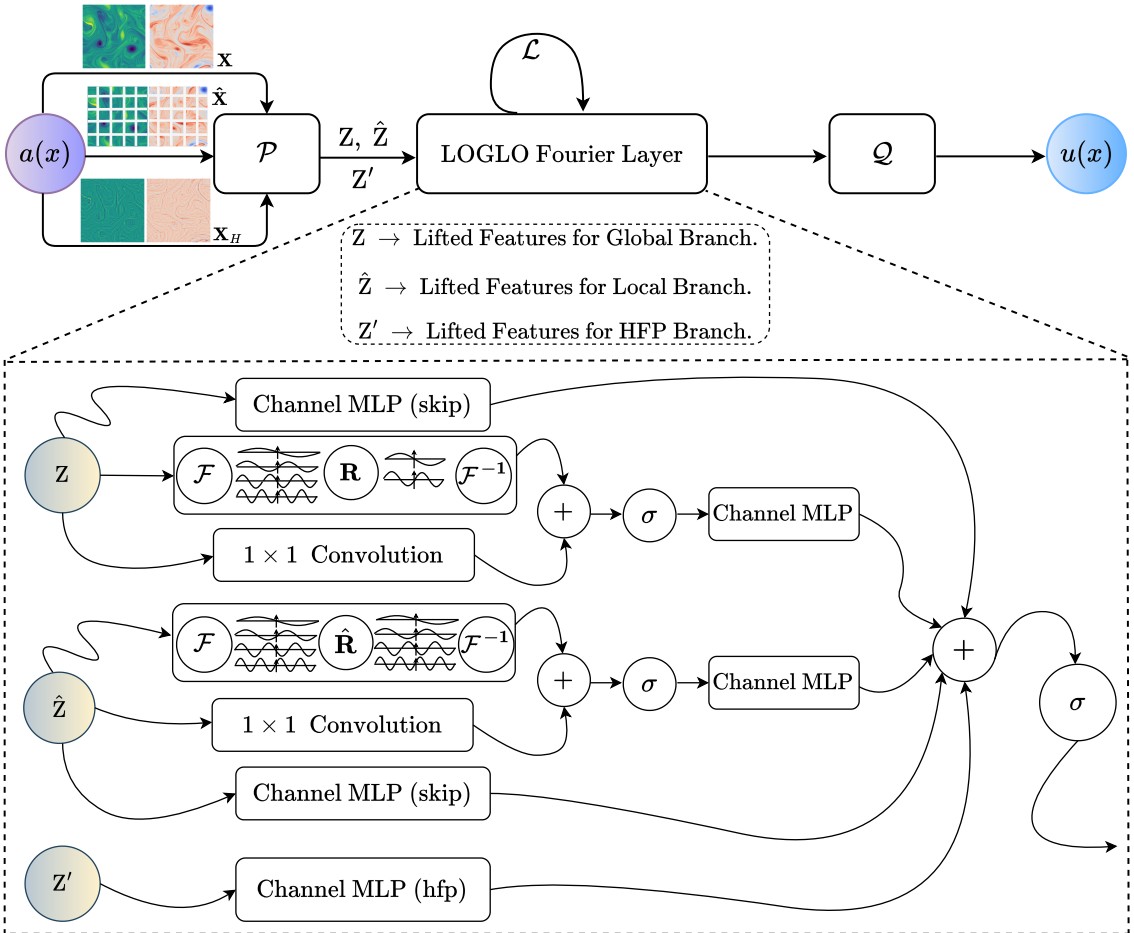

Figure 1: The overall architecture of our proposed LOGLO-FNO model. $\mathbf{X}$ is the discretization of $a(x)$, $\hat{\mathbf{X}}$ the patched version, and $\mathbf{X}_H$ the extracted high frequencies (cf. Eqn. 4). The full network has $\mathcal{L}$ repetitions of identical LOGLO Fourier layers, and the final activation function is applied on all but the last LOGLO Fourier layer. The features from the three branches and the skip connections are fused by a simple summation.

where $P_{m(x)}$ is the m$^{th}$ patch containing $x$. Unlike the global integral operator $\mathcal{K}_g(\cdot)$ in Eqn. 1, the kernel integral operator $\mathcal{K}_l(\cdot)$ in Eqn. 3 is restricted to a local region $P_m$ with a customizable spatial extent. The suffix $\hat{\mathcal{L}}^n$ denotes the $n^{th}$ instance of the local Fourier layer and $m$ indexes a specific patch out of the total M patches. Therefore, $\hat{Z}_{\hat{\mathcal{L}}^n}$ stands for the features fed as input to the n$^{th}$ local Fourier layer.

**High-Frequency Propagation (HFP) Branch.** In order to provide a stronger inductive bias of high-frequency features, we propose an HFP module as the third branch, placed in parallel, that encourages the accurate reconstruction of high-frequencies (Liu et al., 2020) that are otherwise subdued in the Fourier layers of the global branch due to explicit truncation of high-frequency modes. We employ one level of downsampling to the input signal using average pooling and an upsampling block using interpolation. The blurriness of the downsampled signal (e.g., a field variable) is directly proportional to the degree of spatial distortion, which, in the case of average pooling, is controlled by the kernel size and stride. The upsampling operation undoes this effect to reconstruct the original signal resolution. The extracted high frequencies $\mathbf{X}_H$ (cf. Figure 1) are *lifted* using $\mathcal{P}$ with shared parameters for a tight coupling of the high-pass filtered features with that of the full signal features, propagated using a channel MLP, and added to the output features of the local and global Fourier layers. Overall, the HFP block serves as a high-pass filter and can be written compactly as

$$\mathbf{X}_H = \mathbf{X} - \text{Interpolate}(\text{AvgPool}(\mathbf{X}; \text{kernel\_size}, \text{stride}); \; \text{size} = (\dots, N_c, N_x, N_y, [N_z])). \tag{4}$$

## 3.2 Frequency-aware Loss based on Radially Binned Spectral Energy Errors

Since our aim is to direct the optimization process to enable the model to faithfully reconstruct non-dominant frequencies in the predictions, we propose a frequency-sensitive loss and penalize the band-classified (mid- and high-frequency) spectral errors as an additional weighted term in the loss. Let $\tilde{\mathbf{Y}}$ and $\mathbf{Y}$ be the model prediction and target, respectively. The pointwise error in physical space is $\Delta u = \tilde{\mathbf{Y}} - \mathbf{Y}$, and the spectral error $\Delta \hat{u} = \mathcal{F}(\Delta u)$ is obtained by applying the FFT on the spatial dimensions. Then, we define the radially binned, temporal and channel-wise spectral error on 2D spatial data as,

$$C_{\text{freq}}^{t,c}(r) = \frac{L_x}{N_x} \cdot \frac{L_y}{N_y} \sqrt{\frac{1}{N_b} \sum_{b=1}^{N_b} \left( \sum_{\substack{(k_x, k_y) \text{ s.t.} \\ \mathcal{R}(k_x, k_y) = r}} |\Delta \hat{u}_{b,c,t}(k_x, k_y)|^2 \right)}, \tag{5}$$

where the inner summation aggregates the energy of spectral errors within each radial bin $r$, $L_x$ and $L_y$ are the spatial extents of the domain, whereas $N_x$ and $N_y$ are the corresponding total number of observation points along those spatial axes. Subsequently, the binned errors can be band-classified as low, mid, and high frequencies by considering suitable radial cutoff values (e.g., low=[1,4], mid=[5,12], and high=[13, $\mathcal{M}$]; see Appendix K.6) and summed or averaged over the physical variables and temporal dimensions.

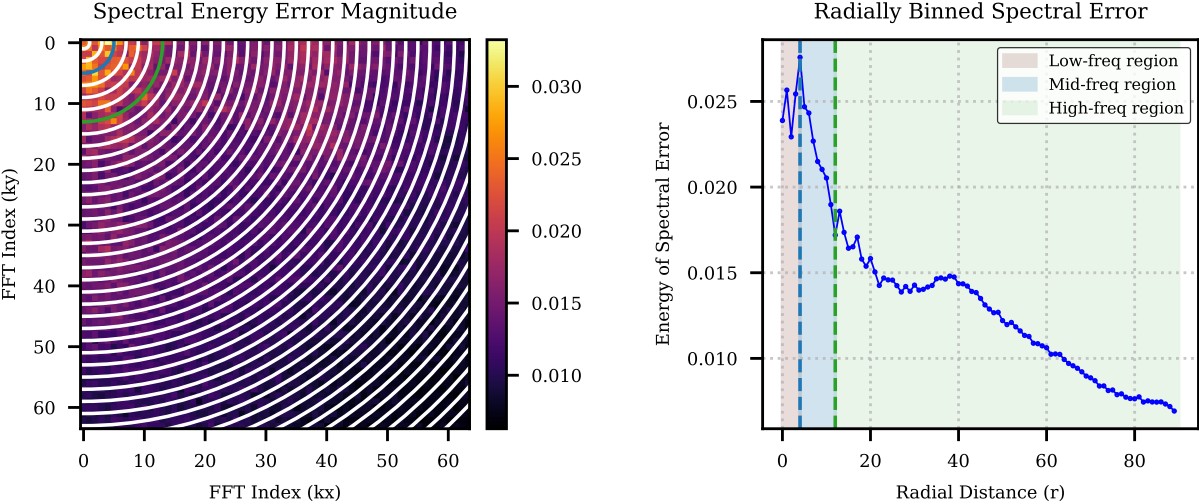

Figure 2: Representative illustration of radially binned spectral error. Left: the spectral energy of error magnitude with overlaid radial bins (blue – low frequency, green – mid frequency cutoff, the rest – high frequency band); Right: the energy of the radially binned spectral error across radial distances as a line plot.

A detailed step-by-step procedure (Algorithm 1) and a PyTorch-style pseudocode to compute the frequency loss for 2D spatial data are provided in Appendix D.2, and the visualizations are placed in Appendix D.3.

## 3.3 Fourier Layer Parameter Budget for 2D Spatial Data

The majority of the parameter mass in FNO is concentrated on the Fourier layers (specifically, the spectral convolution modules) and their cardinality. It is dependent on three factors: (i) the spatial dimensionality $s$

of the problem, (ii) the number of selected Fourier modes $K$, and (iii) the channel[†] dimension $d_c$. Denote the number of Fourier layers as $\mathcal{L}$. Then, for 2D spatial data ($s = 2$), the parameter complexity of the global spectral convolutional layer is $\mathcal{O}\big((d_c \cdot d_c) \cdot K^s \cdot \mathcal{L}\big)$. Note the quadratic growth in parameters as the number of chosen modes $K$ increases. This results in a higher number of trainable parameters for inputs of reasonably high resolutions (e.g., 512×512, or above). Considering this inefficiency and the lack of local convolutions, we consider an approach to distribute the apriori allotted parameter budget across a series of branches, each processing the inputs at different resolutions, mimicking the multi-scale modeling paradigm (Chen et al., 2021; Rahman et al., 2023; Gupta & Brandstetter, 2023). However, we differ from the prevalent approaches in that we do not downsample the incoming spatial resolution in the local branch but decompose it into smaller domains (i.e., patches) to process each independently. Therefore, the parameter complexity of the local Fourier layers performing local spectral convolution on the patches of 2D spatial data would be $\mathcal{O}\big((d_c \cdot d_c) \cdot (n_x \cdot (p_{\text{size}}//2 + 1)) \cdot \mathcal{L}\big)$, where $p_{\text{size}}$ denotes the patch size, $+1$ is for the Nyquist frequency, and $n_x$ is the x-axis resolution of the patches. Note that we retain all Fourier modes in the local branch.

### 3.4 Fourier Layer Parameter Budget for 3D Spatial Data

Extending the analysis of the dependence of trainable parameters on the spectral filter size $K$ for 2D spatial data in §3.3 to 3D spatial data, we obtain the parameter complexity of the global Fourier layer performing global spectral convolutions as $\mathcal{O}\big((d_c \cdot d_c) \cdot K^3 \cdot \mathcal{L}\big)$, while noting the cubic growth in trainable parameters as we increase the number of chosen Fourier modes $K$ for more accurate reconstruction of predictions and $d_c$ indicating the hidden channel dimension. In contrast, the parameter complexity of the local Fourier layers performing local spectral convolution on the patches of 3D spatial data would be $\mathcal{O}\big((d_c \cdot d_c) \cdot (n_x \cdot n_y \cdot (p_{\text{size}}//2 + 1)) \cdot \mathcal{L}\big)$, where $p_{\text{size}}$ denotes the patch size, $+1$ for the Nyquist frequency, and $n_x$ and $n_y$ are the x- and y-axes resolutions of the patches. Note that, as in the case of 2D spatial data, we always retain all Fourier modes in the local branch in all layers.

### 3.5 Distributing Parameters across Local and Global Branches

Following the observations made in §3.3, we consider a scenario where the goal is to train a neural operator given a parameter budget. A baseline FNO can be trained with this budget cost to reach a certain accuracy. In contrast, we argue that we can reach the same accuracy using fewer parameters with LOGLO-FNO by making the allotted parameters less concentrated on the global branch (thereby controlling the quadratic parameter growth in terms of selected modes) and distributing them to the local branch, which has a more subdued parameter growth (for modes) depending on the patch size. If, on the other hand, we use the entire parameter budget, better accuracy than the baseline FNO model can be achieved (see Figures 3, 14, 15, 16, and 20).

## 4 Experiments and Results

In this section, we explain the different training setups we use and present the quantitative results of the baselines and LOGLO-FNO on the six challenging time-dependent PDE problems, i.e., five 2D PDEs (Kolmogorov Flow, CNS Turb, Wave-Gauss, Compressible Euler four-quadrant Riemann problem, and Diffusion-Reaction) and a 3D PDE (Turbulent Radiative Mixing Layer from the Well benchmark), introduced and described in Appendix B.

**Baseline Models.** We consider the following neural operators as relevant and strong baselines in our work. The FNO-related baselines include base FNO (Li et al., 2021; Kossaifi et al., 2024a), U-FNO (Wen et al., 2022) that is well-suited for multiscale modeling, F-FNO (Tran et al., 2023) that factorizes the Fourier transform over the spatial dimensions, and variants of NO-LIDK (Liu-Schiaffini et al., 2024) that achieve local convolutions. Moreover, we include a modern version of U-Net from PDEArena (Gupta & Brandstetter, 2023), LSM (Wu et al., 2023) that applies spectral methods in learned latent space, and a physics-attention based transformer, namely, Transolver (Wu et al., 2024). Further specific details of baselines are in Appendix C.

---

[†]a.k.a. width or hidden channels in the Neural Operator framework: github.com/neuraloperator/neuraloperator

**Training Objective, Procedure, and Evaluation Metrics.** We train the baselines using the standard MSE loss ($\mathcal{C}_{\text{MSE}}$), whereas our frequency loss term is added on top for the LOGLO-FNO models. Consequently, the 1-step training loss for N trajectories, each comprising T timesteps, is,

$$\theta^* = \arg\min_\theta \sum_{n=1}^{N} \sum_{t=1}^{T-1} \mathcal{C}(\mathcal{N}_\theta(u^t), u^{t+1}), \qquad \mathcal{C} = \mathcal{C}_{\text{MSE}} + \lambda \cdot \mathcal{C}_{\text{freq}}, \quad 0 \le \lambda \le 1 \qquad (6)$$

where $(u^t, u^{t+1})$ are the input-output pairs constructed from trajectories and $\mathcal{C}$ is the weighted sum of cost functions, MSE ($\mathcal{C}_{\text{MSE}}$) and our band-classified frequency loss ($\mathcal{C}_{\text{freq}}$). Specifically, we minimize only the spectral errors corresponding to the mid-and high-frequency bands in our experiments. We use the Adam optimizer and halve the learning rate every 33 epochs in the case of Kolmogorov Flow and 100 epochs for Diffusion-Reaction 2D for a fair comparison with NO-LIDK (Liu-Schiaffini et al., 2024). However, the learning rate is halved every 10 epochs on the Turbulent Radiative Layer 3D dataset due to reduced training epochs. The full set of hyperparameters is listed in Appendix O.8. We evaluate metrics from PDEBench (Takamoto et al., 2022), which contain spectral, physics- and data-view-based error measures (fRMSE, cRMSE, nRMSE, MaxError). Additionally, we include metrics that measure the energy spectra deviations, viz. MELR and WLR (Wan et al., 2023). More elaborate details on the evaluation metrics can be found in Appendix G.1.

### 4.0.1 1-step Training and Evaluation

**2D Kolmogorov Flow.** We train using 1-step loss, a.k.a. teacher-forcing: $u(t-\Delta t, \cdot) \to u(t, \cdot)$. This training scheme mimics the functioning of classical numerical solvers and is commonly employed (Gupta & Brandstetter, 2023).

Gaussian noise is added in order to account for the distribution mismatch between this type of teacher-forcing-based training and autoregressive rollout for inference (Pfaff et al., 2021; Stachenfeld et al., 2022). In a similar spirit, we inject *adaptive* Gaussian noise, which is dependent on the high-frequency structures extracted by the HFP module (see Appendix E for full details). Further, we found it essential to employ gradient clipping for stabilized training when modeling high frequencies. The LOGLO-FNO models have been trained with a single local branch (patch size $8 \times 8$ or $16 \times 16$ for 2D spatial data) in addition to a single global branch operating on the full spatial resolution.

**1-step Evaluation.** Once trained, we evaluate the models for 1-step errors. A summary of results comparing proposed LOGLO-FNO with a range of strong baselines, including the FNO-based current state-of-the-art NO-LIDK (Liu-Schiaffini et al., 2024), is in Table 1.

Table 1: 1-step and 5-step AR evaluation of LOGLO-FNO compared with SOTA baselines on the test set of 2D Kolmogorov Flow (Li et al., 2022b). REL. % DIFF indicates improvement (-) or degradation (+) with respect to FNO. LOGLO-FNO uses 40 and (16, 9) modes in the global and local branches, respectively, whereas the width is set as 65. *NO-LIDK* denotes using only localized integral kernel, NO-LIDK$^\diamond$ means only differential kernel, and NO-LIDK$^\dagger$ means employing both. Transolver$^\star$ indicates a longer training time of the model for 500 epochs due to convergence issues at shorter training epochs of 136.

| Model | nRMSE ($\downarrow$) | fRMSE(L) | fRMSE(M) | fRMSE(H) ($\downarrow$) |
|---|---|---|---|---|
| **1-step Evaluation** | | | | |
| U-Net | $1.3 \cdot 10^{-1}$ | $2.24 \cdot 10^{-2}$ | $3.57 \cdot 10^{-2}$ | $4.39 \cdot 10^{-2}$ |
| Transolver$^\star$ | $1.39 \cdot 10^{-1}$ | $2.74 \cdot 10^{-2}$ | $4.48 \cdot 10^{-2}$ | $4.65 \cdot 10^{-2}$ |
| FNO | $1.47 \cdot 10^{-1}$ | $1.36 \cdot 10^{-2}$ | $2.05 \cdot 10^{-2}$ | $4.7 \cdot 10^{-2}$ |
| F-FNO | $1.37 \cdot 10^{-1}$ | $1.49 \cdot 10^{-2}$ | $2.15 \cdot 10^{-2}$ | $4.36 \cdot 10^{-2}$ |
| LSM | $1.36 \cdot 10^{-1}$ | $2.59 \cdot 10^{-2}$ | $4.42 \cdot 10^{-2}$ | $4.64 \cdot 10^{-2}$ |
| U-FNO | $1.12 \cdot 10^{-1}$ | $1.27 \cdot 10^{-2}$ | $2.22 \cdot 10^{-2}$ | $3.71 \cdot 10^{-2}$ |
| *NO-LIDK* | $1.33 \cdot 10^{-1}$ | $1.45 \cdot 10^{-2}$ | $2.69 \cdot 10^{-2}$ | $4.55 \cdot 10^{-2}$ |
| NO-LIDK$^\diamond$ | $1.11 \cdot 10^{-1}$ | $1.65 \cdot 10^{-2}$ | $2.29 \cdot 10^{-2}$ | $3.91 \cdot 10^{-2}$ |
| NO-LIDK$^\dagger$ | $1.07 \cdot 10^{-1}$ | $1.44 \cdot 10^{-2}$ | $2.46 \cdot 10^{-2}$ | $3.82 \cdot 10^{-2}$ |
| **LOGLO-FNO** | $\mathbf{1.07 \cdot 10^{-1}}$ | $\mathbf{1.21 \cdot 10^{-2}}$ | $\mathbf{1.81 \cdot 10^{-2}}$ | $\mathbf{3.54 \cdot 10^{-2}}$ |
| REL. % DIFF | -27.21 % | -11.22 % | -11.88 % | -24.64 % |
| **5-step Autoregressive Evaluation** | | | | |
| U-Net | $2.65 \cdot 10^{-1}$ | $6.53 \cdot 10^{-2}$ | $1.04 \cdot 10^{-1}$ | $9.38 \cdot 10^{-2}$ |
| Transolver$^\star$ | $3.36 \cdot 10^{-1}$ | $7.44 \cdot 10^{-2}$ | $1.51 \cdot 10^{-1}$ | $1.18 \cdot 10^{-1}$ |
| FNO | $2.35 \cdot 10^{-1}$ | $3.37 \cdot 10^{-2}$ | $5.80 \cdot 10^{-2}$ | $8.37 \cdot 10^{-2}$ |
| F-FNO | $2.28 \cdot 10^{-1}$ | $3.60 \cdot 10^{-2}$ | $5.52 \cdot 10^{-2}$ | $8.12 \cdot 10^{-2}$ |
| LSM | $3.18 \cdot 10^{-1}$ | $7.6 \cdot 10^{-2}$ | $1.44 \cdot 10^{-1}$ | $1.12 \cdot 10^{-1}$ |
| U-FNO | $2.03 \cdot 10^{-1}$ | $\mathbf{2.69 \cdot 10^{-2}}$ | $5.33 \cdot 10^{-2}$ | $7.35 \cdot 10^{-2}$ |
| *NO-LIDK* | $2.39 \cdot 10^{-1}$ | $3.14 \cdot 10^{-2}$ | $6.86 \cdot 10^{-2}$ | $8.91 \cdot 10^{-2}$ |
| NO-LIDK$^\diamond$ | $2.04 \cdot 10^{-1}$ | $3.71 \cdot 10^{-2}$ | $5.96 \cdot 10^{-2}$ | $7.6 \cdot 10^{-2}$ |
| NO-LIDK$^\dagger$ | $2.05 \cdot 10^{-1}$ | $3.31 \cdot 10^{-2}$ | $5.94 \cdot 10^{-2}$ | $7.74 \cdot 10^{-2}$ |
| **LOGLO-FNO** | $\mathbf{1.92 \cdot 10^{-1}}$ | $\underline{2.80 \cdot 10^{-2}}$ | $\mathbf{4.55 \cdot 10^{-2}}$ | $\mathbf{6.93 \cdot 10^{-2}}$ |
| REL. % DIFF | -18.39 % | -16.86 % | -21.62 % | -17.26 % |

Since our method also performs convolution using local spectral kernels, we compare our results with their local integral kernel (*NO-LIDK*) scores, whereas the other two variants (NO-LIDK$^\diamond$, NO-LIDK$^\dagger$) are provided for

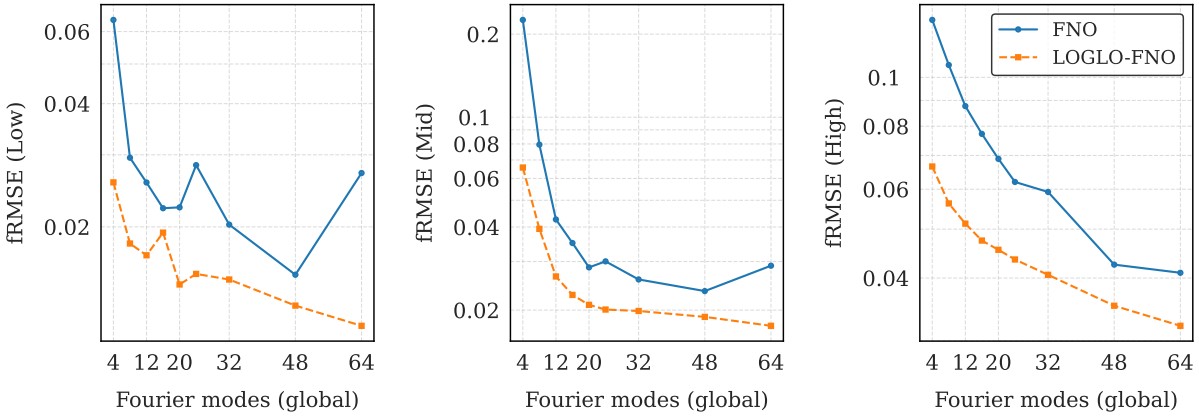

Figure 3: Comparison of FNO vs. LOGLO-FNO showing 1-step fRMSE (↓) on the test set of Kolmogorov Flow dataset ($Re = 5k$) (Li et al., 2022b) for a varying number of modes in the global branch and the full set of modes (i.e., $(16, 9)$ for patch size $(16 \times 16)$) in the local branch.

the sake of completeness. We observe that LOGLO-FNO outperforms the baselines on all metrics. Since we set out to model high-frequencies, we note that the mid- and high-frequency errors are close to 12% and 25% less compared to base FNO, indicating better preservation of high-frequency details, whereas MELR is down by over 75% (see Table 8). The results of a study analyzing the influence of an increasing number of global branch Fourier modes and full count of local branch Fourier modes (i.e., $(16, 9)$ for patch size $(16 \times 16)$) for LOGLO-FNO on the frequency and energy spectra errors are visualized in Figure 3 and Figure 16, respectively. More plots showing other error metrics are placed in Appendix G.4.

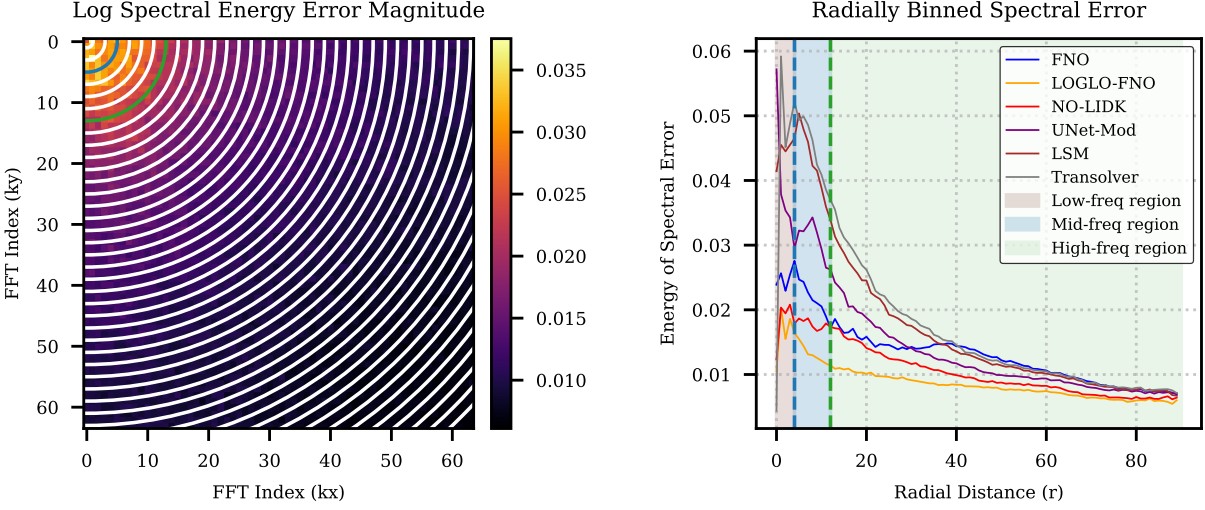

Figure 4: Comparison of radially binned spectral errors of baselines and LOGLO-FNO on the test set of Kolmogorov Flow ($Re = 5k$) (Li et al., 2022b).

**Autoregressive Evaluation.** In this setup, we evaluate the 1-step trained models on autoregressive rollouts for varying numbers of timesteps and compute the errors. The 5-step autoregressive rollout errors are shown in Table 1. We observe that LOGLO-FNO outperforms other models on seven out of the ten metrics and is superior to FNO on all metrics, achieving a noticeable reduction in mid- and high-frequency errors.

These 5-step and extended rollout results (see Figure 5) indicate that LOGLO-FNO is more robust to the autoregressive error accumulation problem than base FNO. Further metrics are visualized in Figures 17, 18, and 19 and the results with the full set of evaluation metrics are provided in Table 8 in Appendix G.2.1.

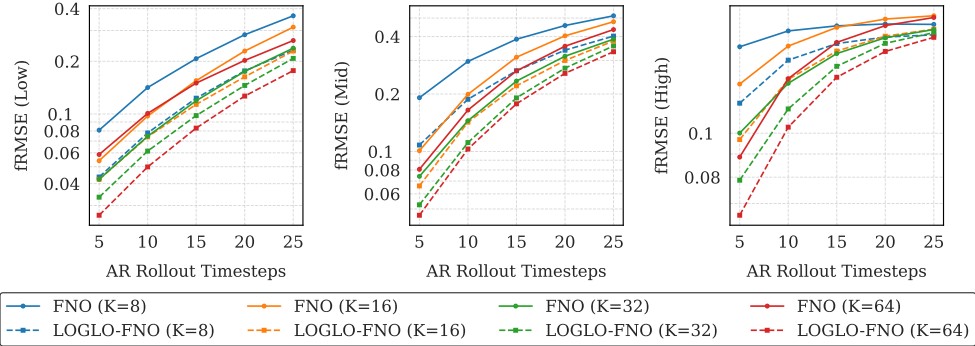

Figure 5: Comparison of FNO vs. LOGLO-FNO for autoregressive rollout showing fRMSE (($\downarrow$)) error growth over variable number of modes (K) in the global branch and varying timesteps in the trajectories on the test set of Kolmogorov Flow ($Re = 5k$) (Li et al., 2022b). The local branch uses a patch size of $16 \times 16$.

**CNS Turb 2D.** We extend our experiments to include compressible Navier-Stokes equations (see Eqn. 8 in Appendix B.2 for details on the PDE), specifically with turbulent initial conditions and sonic regime (Mach=1.0).

The dataset consists of 1k trajectories, of which we use 900 for training and 100 for testing, following PDEBench (Takamoto et al., 2022) split. Input-output data pairs are constructed by slicing successive frames across the timesteps, yielding 18k for training and 2k for testing. Each snapshot has two scalar fields (i.e., density and pressure) and one 2D vector field (i.e., velocity: velocity-x, velocity-y).

We replicate the 1-step training strategy of Kolmogorov Flow 2D problem, but instead use the AdamW optimizer for all models. We observe from the 1-step and 5-step AR errors listed in Table 2, that LOGLO-FNO yields consistent improvements across spatial and spectral metrics and outperforms all baselines in both evaluation settings.

The results on the full set of evaluation metrics are provided in Table 9 in Appendix G.2.1, and the hyperparameters are placed in Appendix O.

Table 2: **Top**: 1-step and 5-step AR evaluation of LOGLO-FNO compared with SOTA baselines on the test set of CNS Turb 2D. LOGLO-FNO uses 40 and (16, 9) modes in the global and local branches, respectively, whereas the width is set as 65. *NO-LIDK** denotes using only localized integral kernel, NO-LIDK$^\diamond$ means only differential kernel, and NO-LIDK$^\dagger$ means employing both. REL. % DIFF indicates improvement (-) or degradation (+) with respect to FNO.

| Model | nRMSE ($\downarrow$) | fRMSE(L) | fRMSE(M) | fRMSE(H) ($\downarrow$) |
|---|---|---|---|---|
| **CNS Turb 2D: 1-step Evaluation** | | | | |
| U-Net | $4.9 \cdot 10^{-2}$ | $2.47 \cdot 10^{-3}$ | $2.9 \cdot 10^{-3}$ | $1.48 \cdot 10^{-3}$ |
| Transolver* | $4.22 \cdot 10^{-1}$ | $4.77 \cdot 10^{-2}$ | $4.11 \cdot 10^{-2}$ | $4.47 \cdot 10^{-3}$ |
| FNO | $4.82 \cdot 10^{-2}$ | $1.28 \cdot 10^{-3}$ | $1.78 \cdot 10^{-3}$ | $1.57 \cdot 10^{-3}$ |
| LSM | $8.08 \cdot 10^{-2}$ | $6.17 \cdot 10^{-3}$ | $6.55 \cdot 10^{-3}$ | $2.01 \cdot 10^{-3}$ |
| U-FNO | $4.49 \cdot 10^{-2}$ | $2.24 \cdot 10^{-3}$ | $2.01 \cdot 10^{-3}$ | $1.43 \cdot 10^{-3}$ |
| *NO-LIDK** | $6.4 \cdot 10^{-2}$ | $1.5 \cdot 10^{-3}$ | $2.14 \cdot 10^{-3}$ | $1.98 \cdot 10^{-3}$ |
| NO-LIDK$^\diamond$ | $5.54 \cdot 10^{-2}$ | $1.35 \cdot 10^{-3}$ | $1.88 \cdot 10^{-3}$ | $1.76 \cdot 10^{-3}$ |
| NO-LIDK$^\dagger$ | $5.57 \cdot 10^{-2}$ | $1.37 \cdot 10^{-3}$ | $1.89 \cdot 10^{-3}$ | $1.77 \cdot 10^{-3}$ |
| **LOGLO-FNO** | $\mathbf{4.24 \cdot 10^{-2}}$ | $\mathbf{9.01 \cdot 10^{-4}}$ | $\mathbf{1.1 \cdot 10^{-3}}$ | $\mathbf{1.37 \cdot 10^{-3}}$ |
| REL. % DIFF | -11.91 % | -29.57% | -38.1 % | -12.38 % |
| **CNS Turb 2D: 5-step Autoregressive Evaluation** | | | | |
| U-Net | $1.44 \cdot 10^{-1}$ | $1.42 \cdot 10^{-2}$ | $1.3 \cdot 10^{-2}$ | $3.08 \cdot 10^{-3}$ |
| Transolver* | $6.69 \cdot 10^{-1}$ | $9.21 \cdot 10^{-2}$ | $6.0 \cdot 10^{-2}$ | $4.7 \cdot 10^{-3}$ |
| FNO | $7.78 \cdot 10^{-2}$ | $3.76 \cdot 10^{-3}$ | $5.37 \cdot 10^{-3}$ | $2.4 \cdot 10^{-3}$ |
| LSM | $2.67 \cdot 10^{-1}$ | $3.4 \cdot 10^{-2}$ | $2.5 \cdot 10^{-2}$ | $4.16 \cdot 10^{-3}$ |
| U-FNO | $8.42 \cdot 10^{-2}$ | $5.38 \cdot 10^{-3}$ | $5.9 \cdot 10^{-3}$ | $2.54 \cdot 10^{-3}$ |
| *NO-LIDK** | $8.55 \cdot 10^{-2}$ | $4.14 \cdot 10^{-3}$ | $5.92 \cdot 10^{-3}$ | $2.62 \cdot 10^{-3}$ |
| NO-LIDK$^\diamond$ | $7.88 \cdot 10^{-2}$ | $3.82 \cdot 10^{-3}$ | $5.35 \cdot 10^{-3}$ | $2.44 \cdot 10^{-3}$ |
| NO-LIDK$^\dagger$ | $7.91 \cdot 10^{-2}$ | $3.82 \cdot 10^{-3}$ | $5.38 \cdot 10^{-3}$ | $2.47 \cdot 10^{-3}$ |
| **LOGLO-FNO** | $\mathbf{6.67 \cdot 10^{-2}}$ | $\mathbf{2.76 \cdot 10^{-3}}$ | $\mathbf{3.97 \cdot 10^{-3}}$ | $\mathbf{2.12 \cdot 10^{-3}}$ |
| REL. % DIFF | -14.31 % | -26.7 % | -26.09 % | -11.42% |

**Wave-Gauss 2D.** As an additional problem exhibiting a different physical phenomenon of wave propagation, we include the 2D wave equation, specifically Wave-Gauss, from Poseidon (Herde et al., 2024). We refer the readers to Eqn. 9 in Appendix B.3 for details on the PDE and the initial and boundary conditions. The spatially dependent propagation speed is generated as a sum of Gaussians.

As with the previous problems in this section, we employ 1-step training strategy with the objective in Eqn. 6 and evaluate on 1-step and 5-step rollouts. The training data contains about 150k input-output pairs, 900 for validation, and 3.6k is reserved for testing.

Similar to the input data formatting setup of Herde et al. (2024), we concatenate the static propagation field $c$ to the time-dependent displacement field $u$ in the input and let the model learn to perform the identity mapping (i.e., copying) of the propagation field in the output.

As we observe from Table 3, although LOGLO-FNO achieves the lowest mid

Table 3: 1-step and 5-step AR evaluation of LOGLO-FNO on the test set of Wave-Gauss 2D. LOGLO-FNO uses 20 and (16, 9) modes in the global and local branches, respectively, whereas the width is set to 32. *NO-LIDK\** denotes using only localized integral kernel, NO-LIDK$^\diamond$ means only differential kernel, and NO-LIDK$^\dagger$ means employing both. REL. % DIFF indicates improvement (-) or degradation (+) with respect to FNO.

| Model | nRMSE ($\downarrow$) | fRMSE(L) | fRMSE(M) | fRMSE(H) ($\downarrow$) |
|---|---|---|---|---|
| **Wave-Gauss 2D: 1-step Evaluation** | | | | |
| Transolver* | $2.29 \cdot 10^{-1}$ | $3.51 \cdot 10^{-2}$ | $2.76 \cdot 10^{-2}$ | $2.2 \cdot 10^{-3}$ |
| FNO | $4.03 \cdot 10^{-2}$ | $7.5 \cdot 10^{-3}$ | $5.45 \cdot 10^{-3}$ | $6.22 \cdot 10^{-4}$ |
| LSM | $\mathbf{3.08} \cdot 10^{-2}$ | $\mathbf{4.91} \cdot 10^{-3}$ | $3.86 \cdot 10^{-3}$ | $8.54 \cdot 10^{-4}$ |
| U-FNO | $3.49 \cdot 10^{-2}$ | $6.62 \cdot 10^{-3}$ | $4.52 \cdot 10^{-3}$ | $7.24 \cdot 10^{-4}$ |
| *NO-LIDK** | $3.93 \cdot 10^{-2}$ | $6.95 \cdot 10^{-3}$ | $5.1 \cdot 10^{-3}$ | $8.22 \cdot 10^{-4}$ |
| NO-LIDK$^\diamond$ | $3.58 \cdot 10^{-2}$ | $6.49 \cdot 10^{-3}$ | $4.78 \cdot 10^{-3}$ | $7.56 \cdot 10^{-4}$ |
| NO-LIDK$^\dagger$ | $3.5 \cdot 10^{-2}$ | $6.31 \cdot 10^{-3}$ | $4.61 \cdot 10^{-3}$ | $7.4 \cdot 10^{-4}$ |
| **LOGLO-FNO** | $\underline{3.11} \cdot 10^{-2}$ | $6.43 \cdot 10^{-3}$ | $\mathbf{3.84} \cdot 10^{-3}$ | $\mathbf{3.44} \cdot 10^{-4}$ |
| REL. % DIFF | -22.79 % | -14.27% | -29.54% | -44.65% |
| **Wave-Gauss 2D: 5-step Autoregressive Evaluation** | | | | |
| Transolver* | $3.62 \cdot 10^{-1}$ | $7.73 \cdot 10^{-2}$ | $3.47 \cdot 10^{-2}$ | $2.93 \cdot 10^{-3}$ |
| FNO | $6.75 \cdot 10^{-2}$ | $1.6 \cdot 10^{-2}$ | $8.35 \cdot 10^{-3}$ | $8.57 \cdot 10^{-4}$ |
| LSM | $\mathbf{4.34} \cdot 10^{-2}$ | $\mathbf{8.64} \cdot 10^{-3}$ | $5.68 \cdot 10^{-3}$ | $1.1 \cdot 10^{-3}$ |
| U-FNO | $6.26 \cdot 10^{-2}$ | $1.6 \cdot 10^{-2}$ | $7.64 \cdot 10^{-3}$ | $9.27 \cdot 10^{-4}$ |
| *NO-LIDK** | $5.96 \cdot 10^{-2}$ | $1.34 \cdot 10^{-2}$ | $7.92 \cdot 10^{-3}$ | $1.19 \cdot 10^{-3}$ |
| NO-LIDK$^\diamond$ | $1.21 \cdot 10^{-1}$ | $2.43 \cdot 10^{-2}$ | $1.12 \cdot 10^{-2}$ | $4.07 \cdot 10^{-3}$ |
| NO-LIDK$^\dagger$ | $1.06 \cdot 10^{-1}$ | $2.1 \cdot 10^{-2}$ | $1.02 \cdot 10^{-2}$ | $3.85 \cdot 10^{-3}$ |
| **LOGLO-FNO** | $\underline{4.81} \cdot 10^{-2}$ | $\underline{1.29} \cdot 10^{-2}$ | $\mathbf{5.43} \cdot 10^{-3}$ | $\mathbf{4.57} \cdot 10^{-4}$ |
| REL. % DIFF | -28.76% | -19.52% | -34.91% | -46.6% |

and high frequency errors, the LSM baseline model outperforms all other models on the spatial and low frequency error metrics, and this trend is maintained on the complete trajectory rollouts (see Table 10).

The results on the full set of evaluation metrics and extended full trajectory rollout results are provided in Table 10 in the Appendix G.2.1, and the hyperparameters for all the models are in Appendix O.

**Compressible Euler four-quadrant Riemann problem 2D.** A challenging problem for neural operators is modeling shock waves and contact discontinuities. Towards this end, we conduct experiments on the Compressible Euler four-quadrant Riemann problem 2D from Poseidon (Herde et al., 2024). This PDE has four variables, viz. density, horizontal and vertical velocity fields, and pressure. See Appendix B.4 for further details. As with other problems in this section, we optimize using 1-step loss (cf. Eqn. 6), where the models are trained to predict all four field variables for a future timestep. We then evaluate the performance of LOGLO-FNO and compare against Base FNO and NO-LIDK variants. The quantitative results (i.e., 1-step and 5-step rollouts) are shown in Table 4, whereas the qualitative visualization of predictions along with the errors for the density, velocity-x, and pressure fields are shown in Figures 6, 7, and 8, respectively.

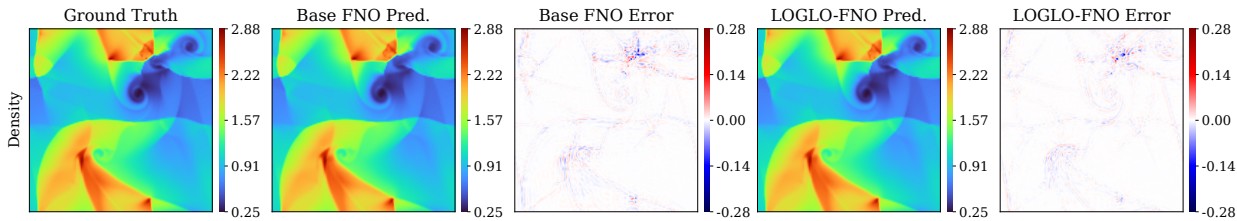

Figure 6: Qualitative results comparing the 1-step density field predictions of Base FNO and LOGLO-FNO on a random sample from the test set of compressible Euler four-quadrant Riemann problem 2D.

Table 4: 1-step and 5-step autoregressive rollout evaluations of LOGLO-FNO compared against SOTA baselines on the test set of Compressible Euler four-quadrant Riemann problem 2D (Herde et al., 2024). LOGLO-FNO uses 40 and (16, 9) modes in the global and local branches, respectively, whereas the width is set to 65. NO-LIDK* denotes using only the localized integral kernel with a radius cutoff value of 0.0078125, and NO-LIDK$^\diamond$ means only the differential kernel is enabled.

| Model | nRMSE ($\downarrow$) | fRMSE(L) | fRMSE(M) | fRMSE(H) ($\downarrow$) |
|---|---|---|---|---|
| **1-step Evaluation** | | | | |
| FNO | $2.08 \cdot 10^{-2}$ | $9.42 \cdot 10^{-4}$ | $1.03 \cdot 10^{-3}$ | $1.41 \cdot 10^{-3}$ |
| NO-LIDK* | $3.58 \cdot 10^{-2}$ | $1.5 \cdot 10^{-3}$ | $2.35 \cdot 10^{-3}$ | $2.47 \cdot 10^{-3}$ |
| NO-LIDK$^\diamond$ | $1.66 \cdot 10^{-2}$ | $1.05 \cdot 10^{-3}$ | $8.54 \cdot 10^{-4}$ | $1.11 \cdot 10^{-3}$ |
| **LOGLO-FNO** | $\mathbf{1.5} \cdot 10^{-2}$ | $\mathbf{5.79} \cdot 10^{-4}$ | $\mathbf{5.09} \cdot 10^{-4}$ | $\mathbf{9.67} \cdot 10^{-4}$ |
| REL. % DIFF | -27.9% | -38.56% | -50.71% | -31.58% |
| **5-step Autoregressive Evaluation** | | | | |
| FNO | $3.43 \cdot 10^{-2}$ | $2.58 \cdot 10^{-3}$ | $2.7 \cdot 10^{-3}$ | $2.35 \cdot 10^{-3}$ |
| NO-LIDK* | $4.02 \cdot 10^{-2}$ | $3.19 \cdot 10^{-3}$ | $3.58 \cdot 10^{-3}$ | $2.47 \cdot 10^{-3}$ |
| NO-LIDK$^\diamond$ | $3.13 \cdot 10^{-2}$ | $2.5 \cdot 10^{-3}$ | $2.38 \cdot 10^{-3}$ | $2.12 \cdot 10^{-3}$ |
| **LOGLO-FNO** | $\mathbf{2.34} \cdot 10^{-2}$ | $\mathbf{1.64} \cdot 10^{-3}$ | $\mathbf{1.35} \cdot 10^{-3}$ | $\mathbf{1.56} \cdot 10^{-3}$ |
| REL. % DIFF | -31.75% | -36.7% | -50% | -33.4 % |

We observe from Table 4 that LOGLO-FNO achieves better results than the baselines on all metrics, while reducing the mid- and high-frequency errors by over 50% and 30%, respectively. The full set of evaluation metrics and extended full trajectory rollouts are placed in Table 11 in the Appendix G.2.1, and the hyperparameters for all models studied are in Appendix O.

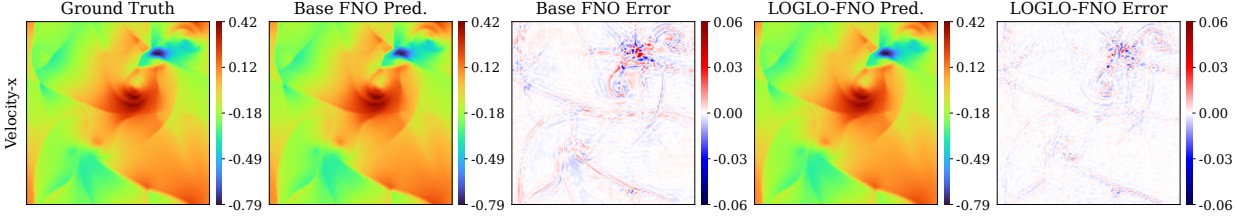

Figure 7: Qualitative results comparing 1-step horizontal velocity field predictions of Base FNO and LOGLO-FNO on a random sample from the test set of compressible Euler four-quadrant Riemann problem 2D.

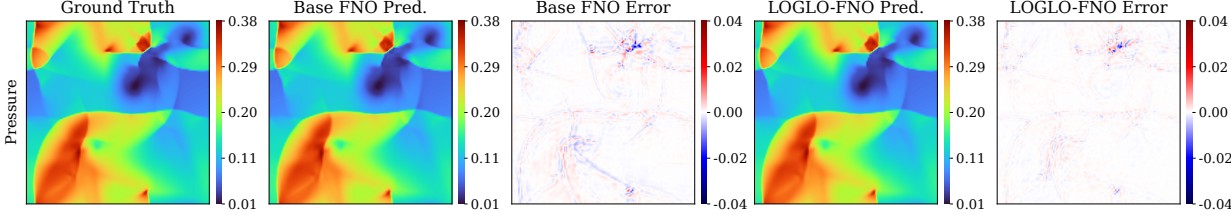

Figure 8: Qualitative results comparing 1-step pressure field predictions of Base FNO and LOGLO-FNO models on a random sample from the test set of compressible Euler four-quadrant Riemann problem 2D.

The above qualitative visualizations, specifically the error maps, reveal that LOGLO-FNO mitigates the errors to a considerable extent in the regions of high gradients. However, the model still struggles to eliminate them entirely, signifying the challenge of modeling discontinuities and sharp gradients that arise from shocks.

**Turbulent Radiative Layer 3D (TRL3D).** Noting the absence of support for local convolutions in FNO for 3D spatial data, we extend the LOGLO-FNO implementation to 3D and conduct experiments on a challenging version of a time-dependent turbulence simulation, viz., Turbulent Radiative Layer 3D (Fielding et al., 2020; Ohana et al., 2024) from the Well benchmark.

Similar to the Kolmogorov Flow 2D setup, the models are trained using 1-step training loss (cf. Eqn. 6), however, with the AdamW optimizer and only for 50 epochs by halving the learning rate every 10 epochs, maintaining the proportion of learning rate decays as with other problems. We chose this setup since we observed overfitting when training for longer epochs, such as 136 or even further. We attribute this behavior to the limited amount of available training data. Following Ohana et al. (2024), we consider the full data covering the entire range of $t_{cool}$ parameters, where $t_{cool} \in \{0.03, 0.06, 0.1, 0.18, 0.32, 0.56, 1.00, 1.78, 3.16\}$. For each $t_{cool}$ value, ten trajectories, each spanning 101 timesteps and of spatial resolution $128 \times 128 \times 256$, are provided. The dataset also comes with an explicit train (8), validation (1), and test (1) splits. We then construct 7200, 900, and 900 1-step data pairs for training, validation, and testing, respectively.

Table 5: 1-step and autoregressive evaluation of LOGLO-FNO compared with state-of-the-art baselines on the test set of Turbulent Radiative Mixing Layer 3D dataset (Fielding et al., 2020; Ohana et al., 2024). Rel. % Diff indicates an improvement (-) or degradation (+) with respect to base FNO, which uses a spectral filter size of 12 and 48 hidden channels. LOGLO-FNO uses 12 and (16, 16, 17)* modes in the global and local branches, respectively, whereas the width is set to 52. (*This is because the Turbulent Radiative Layer 3D dataset has 2× more sampling points on the z-axis relative to the x and y axes resolutions – (128 × 128 × 256).) Base FNO‡ means our implementation of FNO (see Figure 1). [t:t+Δt] symbolizes the inclusion of the timesteps at both ends of the interval when computing time-averaged vRMSE.

| Model | 1-step Eval. | Autoregressive Evaluation | |
|---|---|---|---|
| | 1-step vRMSE (↓) | [6:12] Time-Avg. vRMSE (↓) | [13:30] Time-Avg. vRMSE (↓) |
| FNO | $5.28 \cdot 10^{-1}$ | $8.1 \cdot 10^{-1}$ | $9.4 \cdot 10^{-1}$ |
| TFNO | $5.19 \cdot 10^{-1}$ | $>10$ | $>10$ |
| U-Net | $3.73 \cdot 10^{-1}$ | $9.5 \cdot 10^{-1}$ | $1.09 \cdot 10^{0}$ |
| CNextU-Net | $3.67 \cdot 10^{-1}$ | $7.7 \cdot 10^{-1}$ | $8.6 \cdot 10^{-1}$ |
| LSM | $\mathbf{2.74} \cdot 10^{-1}$ | $7.67 \cdot 10^{-1}$ | $1.02 \cdot 10^{0}$ |
| NO-LIDK◇ | $4.05 \cdot 10^{0}$ | $6.0 \cdot 10^{0}$ | $6.52 \cdot 10^{0}$ |
| Base FNO‡ | $3.18 \cdot 10^{-1}$ | $7.49 \cdot 10^{-1}$ | $8.58 \cdot 10^{-1}$ |
| **LOGLO-FNO** | $\underline{2.76} \cdot 10^{-1}$ | $\mathbf{7.09} \cdot 10^{-1}$ | $\mathbf{7.72} \cdot 10^{-1}$ |
| Rel. % Diff | -13.22% | -5.3% | -9.98% |

| Model | nRMSE (↓) | fRMSE(L) | fRMSE(M) | fRMSE(H) (↓) |
|---|---|---|---|---|
| | 1-step Evaluation | | | |
| LSM | $8.39 \cdot 10^{-1}$ | $2.4 \cdot 10^{-1}$ | $9.7 \cdot 10^{-2}$ | $3.43 \cdot 10^{-2}$ |
| NO-LIDK◇ | $4.34 \cdot 10^{-1}$ | $6.87 \cdot 10^{-2}$ | $1.22 \cdot 10^{-1}$ | $1.23 \cdot 10^{-1}$ |
| Base FNO‡ | $3.06 \cdot 10^{-1}$ | $4.87 \cdot 10^{-2}$ | $4.58 \cdot 10^{-2}$ | $2.89 \cdot 10^{-2}$ |
| **LOGLO-FNO** | $\mathbf{2.65} \cdot 10^{-1}$ | $\mathbf{4.45} \cdot 10^{-2}$ | $\mathbf{3.68} \cdot 10^{-2}$ | $\mathbf{2.48} \cdot 10^{-2}$ |
| Rel. % Diff | -13.27% | -8.61% | -19.57% | -14.01% |

Following recent investigations for conditioning the neural operators to generalize to unseen PDE parameters (Takamoto et al., 2023), we append the $t_{cool}$ coefficients as additional channels. The goal of the so-called *forward problem* for this PDE is then to predict the scalar (i.e., density and pressure) and vector (i.e., velocity-x, velocity-y, and velocity-z) physical quantities of $u^{t+1}(.)$ given the solution at the previous timestep $u^t(.)$. Further details on the PDE are provided in Appendix B.6.

Note that we have improved the 1-step vRMSE score (on the TRL3D test set) of the baseline FNO model by 41.4% compared to results reported by Ohana et al. (2024) (cf. Table 2: Model Performance Comparison) and, hence, our version of FNO is a strong baseline. We further note that the local integral kernel (DISCO conv layers) of NO-LIDK (Liu-Schiaffini et al., 2024) is not implemented for 3D spatial data and, hence, is not compared against in our experimental setup. We observe that LOGLO-FNO achieves the best results on all metrics and outperforms CNextU-Net, the best results of Ohana et al. (2024) on the Turbulent Radiative Layer 3D test dataset, on the vRMSE metric by 24.5%. The qualitative visualizations of predictions are provided in Appendix H.0.2 and the table with the full set of metrics is in Table 12 in the Appendix.

### 4.0.2 Autoregressive Training and Evaluation

We evaluate LOGLO-FNO on a fully autoregressive training setup on the Diffusion-Reaction 2D PDE from Takamoto et al. (2022). The model is trained with an initial context of 10 frames (i.e., $\Delta t = 10$), predicting one future state at a time until reaching the fully evolved state: $u(t - \Delta t, \cdot) \to u(t, \cdot)$. In other

words, since each simulation trajectory consists of 101 timesteps, AR rollout is performed for 91 timesteps both during training and evaluation to be consistent and for a fair comparison with the training setup of NO-LIDK (Liu-Schiaffini et al., 2024).

**2D Diffusion-Reaction.** We observe from Table 6 that LOGLO-FNO achieves a significant reduction in frequency errors across all bands (55%, 43%, 21%, resp.) compared to baseline FNO and superior to NO-LIDK on low- and mid-frequencies (30%, 30.4%, resp.). The tabulation of results with all metrics is provided in Table 13 in the Appendix G.2.2.

## 5 Ablation Studies

In order to investigate the effect of each of the proposed modules in LOGLO-FNO or the radially binned frequency loss, we conduct an ablation study on the Turbulent Radiative Layer 3D dataset, isolating the components. The LOGLO-FNO model (cf. Fig. 1) by default uses the local spectral convolution and the HFP branches, and is trained with the radially binned spectral loss. This configuration corresponds to the result on the second row in Table 7 and achieves the best result, emphasizing the need to retain all the proposed contributions.

Afterwards, to obtain the result in the next row, we disable the frequency loss in the training objective of Eqn. 6 while retaining all other hyperparameters as before, quantifying the effect of the absence of radially binned frequency loss.

In the next setup, we additionally disable the HFP branch to yield a configuration of training LOGLO-FNO with only the local branch and MSE loss. The result for this is on the penultimate row of Table 7.

To isolate the effect of removing the HFP branch, we disable it and train LOGLO-FNO with the training objective in Eqn. 6. The result of this model configuration is presented in the last row of Table 7.

To summarize, although just using any one of the proposed contributions is sufficient to obtain an improved result over baseline FNO, both local spectral convolution and HFP branches, and the radially binned frequency loss are necessary to achieve the best result.

Table 6: Fully autoregressive evaluation of LOGLO-FNO compared with SOTA baselines on the test set of challenging 2D Diffusion-Reaction coupled problem from PDEBench. We also report the REL. % DIFF to indicate the error improvement (-) with respect to baseline FNO.

| Model | nRMSE ($\downarrow$) | fRMSE(L) | fRMSE(M) | fRMSE(H) ($\downarrow$) |
|---|---|---|---|---|
| U-Net | $8.4 \cdot 10^{-1}$ | $1.7 \cdot 10^{-2}$ | $8.2 \cdot 10^{-4}$ | $5.7 \cdot 10^{-2}$ |
| Transolver | $2.63 \cdot 10^{-1}$ | $2.99 \cdot 10^{-3}$ | $2.03 \cdot 10^{-3}$ | $5.51 \cdot 10^{-4}$ |
| U-FNO | $2.6 \cdot 10^{-1}$ | $3.4 \cdot 10^{-3}$ | $1.6 \cdot 10^{-3}$ | $2.6 \cdot 10^{-4}$ |
| FNO | $8.3 \cdot 10^{-2}$ | $6.2 \cdot 10^{-4}$ | $5.6 \cdot 10^{-4}$ | $2.4 \cdot 10^{-4}$ |
| F-FNO | $7.0 \cdot 10^{-2}$ | $9.6 \cdot 10^{-4}$ | $4.7 \cdot 10^{-4}$ | $\mathbf{1.3} \cdot 10^{-4}$ |
| LSM | $4.47 \cdot 10^{-1}$ | $7.17 \cdot 10^{-3}$ | $2.4 \cdot 10^{-3}$ | $3.67 \cdot 10^{-4}$ |
| NO-LIDK (loc. int) | $\mathbf{6.3} \cdot 10^{-2}$ | $4.0 \cdot 10^{-4}$ | $4.6 \cdot 10^{-4}$ | $1.5 \cdot 10^{-4}$ |
| **LOGLO-FNO** | $6.4 \cdot 10^{-2}$ | $\mathbf{2.8} \cdot 10^{-4}$ | $\mathbf{3.2} \cdot 10^{-4}$ | $1.9 \cdot 10^{-4}$ |
| REL. % DIFF | -22.75 % | -54.84 % | -42.86 % | -20.83 % |

Table 7: 1-step results of LOGLO-FNO on the test set of TRL3D (Fielding et al., 2020; Ohana et al., 2024). Base FNO uses a spectral filter size of 18 and 48 hidden channels. LOGLO-FNO uses 16 and $(16, 16, 17)^*$ Fourier modes in the global and local branches, respectively, whereas the width is 52. (*Turbulent Radiative Layer 3D dataset has $2\times$ more sampling points on the z-axis relative to the x- and y-axes resolutions – $(128 \times 128 \times 256)$.)

| Model | nRMSE ($\downarrow$) | fRMSE(L) | fRMSE(M) | fRMSE(H) | vRMSE ($\downarrow$) |
|---|---|---|---|---|---|
| **1-step Evaluation** | | | | | |
| Base FNO | $2.97 \cdot 10^{-1}$ | $5.17 \cdot 10^{-2}$ | $4.45 \cdot 10^{-2}$ | $2.76 \cdot 10^{-2}$ | $3.09 \cdot 10^{-1}$ |
| LOGLO-FNO (+freq loss, +HFP) | $\mathbf{2.58} \cdot 10^{-1}$ | $\mathbf{4.1} \cdot 10^{-2}$ | $\mathbf{3.62} \cdot 10^{-2}$ | $\mathbf{2.45} \cdot 10^{-2}$ | $\mathbf{2.68} \cdot 10^{-1}$ |
| LOGLO-FNO (-freq loss, +HFP) | $2.75 \cdot 10^{-1}$ | $4.4 \cdot 10^{-2}$ | $4.09 \cdot 10^{-2}$ | $2.63 \cdot 10^{-2}$ | $2.86 \cdot 10^{-1}$ |
| LOGLO-FNO (-freq loss, -HFP) | $2.76 \cdot 10^{-1}$ | $4.4 \cdot 10^{-2}$ | $4.13 \cdot 10^{-2}$ | $2.64 \cdot 10^{-2}$ | $2.87 \cdot 10^{-1}$ |
| LOGLO-FNO (+freq loss, -HFP) | $2.66 \cdot 10^{-1}$ | $4.53 \cdot 10^{-2}$ | $3.76 \cdot 10^{-2}$ | $2.48 \cdot 10^{-2}$ | $2.76 \cdot 10^{-1}$ |

## 6 Conclusions, Limitations, and Future Work

In this paper, we have proposed an enhancement to the FNO architecture (Li et al., 2021; Kossaifi et al., 2024b;a) through auxiliary parallel branches for local spectral convolution and high-frequency feature propagation. Through our experiments on six relevant and highly challenging PDE problems, namely, Kolmogorov Flow 2D, CNS Turb 2D, Wave-Gauss 2D, Compressible Euler four-quadrant Riemann problem 2D, Turbulent Radiative Layer 3D, and non-linearly coupled Diffusion-Reaction 2D problems that give rise to turbulence or high frequencies, we have demonstrated the promise to significantly reduce both the

spatial and spectral errors, showing the importance of local convolutions in mitigating the spectral bias to a considerable extent. In addition, our proposed radially binned spectral loss has also been found to be useful in this aspect. As a result of incorporating a local spectral convolutional branch in the architecture, our model has paved a way for reducing the number of trainable parameters to reach the baseline model accuracy. The current work, however, in order to allow a fair comparison to Liu-Schiaffini et al. (2024), has not considered any sophisticated parameter-efficient spectral convolution methods (e.g., tensor factorization techniques introduced in Kossaifi et al. (2024b)), which we see as a limitation. Future work could look into improving the scalability of LOGLO-FNO, provide a theoretical analysis of its benefits, or explore optimized implementation to accelerate the training and inference speed.

### Broader Impact Statement

Accelerated and efficient Neural PDE Solvers help reduce the cost of running cost-prohibitive simulations such as weather forecasting, cyclone predictions, and extreme weather events. As a consequence, disaster preparedness, the design, and manufacturing timelines can be accelerated, resulting in life and cost savings as well as decreased $CO_2$ emissions. As a negative side effect, we cannot rule out the possibility of misuse by bad actors, as computational fluid dynamics and turbulent flow simulations are also used to design military equipment within the military-industrial complex.

### Acknowledgments

The authors are thankful for the suggestions and questions of the anonymous reviewers on OpenReview, which have been helpful in improving the manuscript. We thank Miguel Liu-Schiaffini for the correspondence on NO-LIDK and Julius Berner for the initial discussions on local convolutions. Furthermore, we thank Artur Petrov Toshev for the insightful discussions on operator learning and brainstorming ideas in general. Moreover, we thank Julius Herb for the discussions and feedback on the initial draft. In addition, we thank Makoto Takamoto for the fruitful discussions during the rebuttal phase. Marimuthu Kalimuthu and Mathias Niepert are funded by Deutsche Forschungsgemeinschaft (DFG, German Research Foundation) under Germany's Excellence Strategy - EXC 2075 – 390740016. We acknowledge the support of the Stuttgart Center for Simulation Science (SimTech). The authors thank the International Max Planck Research School for Intelligent Systems (IMPRS-IS) for supporting Marimuthu Kalimuthu and Mathias Niepert. Last but not least, the authors gratefully acknowledge the computing time provided to them at the NHR Center NHR4CES at RWTH Aachen University (project number p0021158). This is funded by the Federal Ministry of Education and Research, and the state governments participating on the basis of the resolutions of the GWK for national high-performance computing at universities (http://www.nhr-verein.de/unsere-partner).

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

# LOGLO-FNO: Efficient Learning of Local and Global Features in Fourier Neural Operators
## Supplementary Material To The Main Paper
### Project Page: HTTPS://KMARIO23.GITHUB.IO/LOGLO-FNO/

# A Related Work

It has been a well-established fact that deep neural networks, trained with first-order gradient descent optimization methods, have a spectral bias for learning functions with low frequencies before picking up high-frequency details, explaining their superior generalization capabilities on downstream tasks (Xu et al., 2019; Rahaman et al., 2019; Cao et al., 2021; Schwarz et al., 2021; Molina et al., 2024; Yu et al., 2024). This means that deep networks require longer training times (Basri et al., 2019), necessitate multi-stage residue-based training (Wang & Lai, 2024; Ng et al., 2024; Kong et al., 2025), or experiences difficulty altogether, in learning complex functions with high-frequency components. This phenomenon is apropos of the neural PDE solving community (Li et al., 2022b; Fu et al., 2023; Liu et al., 2024) where the high-frequencies are more pronounced in multiscale, multiphysics, time-dependent PDEs and can evolve in time. This limitation of deep neural nets affects not only an accurate reconstruction of the spectral components but also rollouts (Worrall et al., 2025) and causes hallucinations (Sun et al., 2024b).

In the field of computer graphics and vision, several approaches have been proposed to solve this problem. Sun et al. (2024a) explore learning high-frequencies for generative tasks in computer graphics and propose a sinusoidal positional encoding method to learn high-frequency features effectively. Pan et al. (2022) propose a faster variant of ViT with HiLo attention by splitting the attention heads to learn high and low frequencies in a disentangled fashion.

In the research area of neural PDE solving and SciML applications, Liu et al. (2024) study the limitations of existing neural operators for multiscale PDEs and propose a hierarchical attention method and $H^1$ loss to direct the emulator to learn high-frequencies. Other approaches in improving the capability of neural operators to learn high frequencies include (i) an adaptive selection of Fourier modes and resolution (George et al., 2024) since using a suboptimal choice for the number of modes may be detrimental on many PDE problems (Lanthaler et al., 2024), (ii) a two-staged approach of first predicting the future states, given some historical temporal context, using a neural operator and feeding that as an input (i.e., conditioning factor) to a diffusion model in stage 2 (Oommen et al., 2024). Similarly, Fan et al. (2024) employ a two-stage pipeline of first generating low-resolution Kolmogorov flow turbulence simulations using a hybrid DNS neural solver on coarse grids and leveraging a conditional diffusion model to super-resolve for high frequencies and fine-grained structures in the second stage. Poli et al. (2022) study frequency-domain models for PDE solving with an aim to accelerate the training process. They propose a method to learn solution operators directly in the frequency domain, which is made possible with a variance-preserving weight initialization scheme. Closest to our work in terms of achieving local convolution and modeling high-frequency features in FNO is NO-LIDK (Liu-Schiaffini et al., 2024), which designs two different local layers for learning the differential and integral operators that capture local and high-frequency features. Hence, we consider this as the relevant and competitive baseline. Although LOGLO-FNO has a high-level similarity to NO-LIDK by introducing architectural modifications to FNO with an additional parallel branch for local convolutions, it differs from NO-LIDK in the manner it achieves local convolutions. It does so by decomposing the input domain into sub-domains and then individually applying spectral convolutions on these sub-domains, thereby effectively capturing localized features. Moreover, an additional parallel branch specifically designed for modeling and providing an inductive bias of high-frequency features is introduced.

Chen et al. (2021) develop a multi-scale vision transformer for image classification tasks and explore different fusion methods aimed at an effective fusion of multi-scale features from the respective branches. Dai et al. (2021) explore attention mechanisms for aggregating the local and global context features and demonstrate their effectiveness on image classification and object localization tasks in computer vision. Specifically, they propose a multi-scale channel attention module and attentional feature fusion blocks for fusing multi-scale features in deep neural networks.

# B  Benchmark PDEs and Datasets

We experiment with 2D and 3D transient PDEs that exhibit high frequencies, turbulence behaviour, and rich local patterns due to their inherent nature (e.g., coupled variables), controllable parameters (i.e., PDE coefficients, Reynolds number, forcing terms), chaotic dynamics, shocks, discontinuities, porosity of the medium, and higher-order derivatives.

## B.1  2D Incompressible Navier-Stokes (INS)

**Kolmogorov Flow.**  Following prior literature (Li et al., 2022b; George et al., 2024; Liu-Schiaffini et al., 2024; Lanthaler et al., 2024), we adopt 2D Kolmogorov Flow, which is a form of Navier-Stokes PDE, as one of the test cases.

$$\frac{\partial u}{\partial t} + u \cdot \nabla u - \underbrace{\frac{1}{Re}}_{\text{Reynolds number}} \Delta u = -\nabla p + \boxed{\sin(ny)\hat{x}}, \qquad \nabla \cdot u = 0, \qquad \text{on } [0, 2\pi]^2 \times (0, \infty) \qquad (7)$$

forcing term; (n=4)

We use the dataset configuration of Li et al. (2022a) for $Re5000$ that exhibits high-frequency structures due to the flow being fully turbulent. It consists of 100 samples, each with a spatial and temporal resolution of 128×128 and 500 snapshots, respectively. As with George et al. (2024), we exclude the first 100 timesteps to let the trajectories reach the attractor, obtaining a total of 40,000 pairs of samples, out of which we use 36K input-output pairs for training and 4K for testing.

## B.2  2D Compressible Navier-Stokes (CNS, PDEBench)

In order to study a problem exhibiting turbulence and also capable of describing shock wave formation and propagation, we experiment with the 2D CNS equations (Eq. 8a to 8c) from PDEBench (Takamoto et al., 2022). More specifically, we use the sonic regime dataset (Mach=1.0) where the initial conditions comprise turbulent velocity with uniform mass density and pressure, and the boundary conditions are periodic.

mass density (mass per unit volume)

shear viscosity

$$\partial_t \rho + \nabla \cdot (\boxed{\rho} \mathbf{v}) = 0, \qquad (8a)$$

$$\rho(\partial_t \mathbf{v} + \mathbf{v} \cdot \nabla \mathbf{v}) = -\nabla p + \eta \triangle \mathbf{v} + (\zeta + \boxed{\frac{\eta}{3}}) \nabla(\nabla \cdot \mathbf{v}), \qquad (8b)$$

bulk viscosity

pressure

$$\partial_t(\epsilon + \frac{\rho v^2}{2}) + \nabla \cdot [(\boxed{p} + \boxed{\epsilon} + \frac{\rho v^2}{2})\mathbf{v} - \mathbf{v} \cdot \boxed{\sigma'}] = \mathbf{0}, \qquad (8c)$$

internal energy

viscous stress tensor

The dataset (Mach=1.0) contains 1000 trajectories, each comprising 21 timesteps. Following the data split of the PDEBench[*] repository, we use 900 samples for training and the rest for testing. To facilitate the 1-step training and testing tasks, we construct the input-output pairs by slicing the solution fields (i.e., density, pressure, velocity-x, and velocity-y) at successive timesteps along the length of the trajectory, yielding 18K samples for training and 2K for testing. Due to the high spatial resolution of the data of $512 \times 512$, we subsample it to $256 \times 256$ throughout our experiments. However, we use the full spatial resolution to evaluate spatial ZSSR capabilities.

---

[*]https://github.com/pdebench/PDEBench

### B.3 2D Wave Equation (Wave-Gauss, Poseidon)

As an additional challenging PDE problem modeling a different physical phenomenon of the propagation of acoustic waves through a spatially varying medium, we consider the time-dependent 2D Wave Equation, specifically the so-called Wave-Gauss, from the PDEgym[††] benchmark of Poseidon (Herde et al., 2024).

$$\frac{\partial^2 u(x,y,t)}{\partial t^2} - (\ c(x,y)\ )^2 \left( \frac{\partial^2 u(x,y,t)}{\partial x^2} + \frac{\partial^2 u(x,y,t)}{\partial y^2} \right) = 0, \qquad \text{in } D \times (0,T) \tag{9}$$

spatially varying propagation field

The initial condition $u^0(x,y)$ of the displacement $u$, and the propagation field $c(x,y)$ are both generated using a sum of several Gaussians, and hence the name.

$$u_0(x,y) = \sum_{i=1}^{n} g_i(x,y), \quad x,y \in (0,1)$$

$$g_i(x,y) = \exp\left( -\frac{(x_{c,i} - x)^2 + (y_{c,i} - y)^2}{2s_i^2} \right), \quad x,y \in (0,1)$$

where $s_i$ is the standard deviation of the $i^{th}$ Gaussian in the propagation field $c$.

$$c(x,y) = c_0 + \sum_{i=1}^{4} f_i(x,y), \quad x,y \in (0,1)$$

$$f_i(x,y) = v_i \cdot \exp\left( -\frac{(x_i + dx_i - x)^2 + (y_i + dy_i - y)^2}{2\sigma_i^2} \right), \quad x,y \in (0,1)$$

where $v_i$ and $\sigma_i$ are the amplitude and standard deviation of the $i^{th}$ Gaussian, respectively.

The dataset has a total of 10,512 trajectories of 16 timesteps each, with a spatial resolution of $128 \times 128$, of which we use 10,212 trajectories for training, 60 trajectories for validation, and 240 trajectories for testing, following the data split provided by the authors. Since we perform 1-step training and evaluation, we construct input-output pairs by sampling subsequent timesteps in the length of the trajectory, yielding us 153.18K training, 900 validation, and 3.6K testing sample pairs. The autoregressive rollout is evaluated against the full trajectory length by predicting one future timestep at a time.

### B.4 Compressible Euler four-quadrant Riemann problem 2D (CE-RP, Poseidon)

In order to test the capabilities of LOGLO-FNO in modeling shock waves and discontinuities, we consider the highly challenging Compressible Euler four-quadrant Riemann problem 2D from Poseidon (Herde et al., 2024). The four-quadrant Riemann problem is a generalization of the Sod shock tube to 2D space.

$$D_{i,j} = \left\{ (x,y) \in \mathbb{T}^2 \mid \frac{i-1}{p} \le x < \frac{i}{p}, \frac{j-1}{p} \le y < \frac{j}{p} \right\},$$

where $\mathbb{T}^2$ is the 2D torus, and considering a unit square $D = [0,1]^2$, it can be divided into $2 \times 2$ square sub-domains when $p = 2$. The underlying solution operator $S(t, \rho_0, v_{x,y}^0, p_0) = [\rho(t), v_{x,y}(t), p(t)]$, solving the

---

[††] https://camlab-ethz.github.io/poseidon/#Time-dependent

compressible Euler equations with periodic boundary conditions, has four fields, viz. density, pressure, and velocity.

The dataset consists of 10k trajectories, with each sample containing 21 snapshots at a spatial resolution of $128 \times 128$. Following the pre-defined train/val/test split of Herde et al. (2024), we use the initial 9640 trajectories for training, the next 120 for validation, and the ultimate 240 for testing. As with other PDE problems we consider, we slice successive timesteps as input-output pairs starting from the initial condition of a trajectory. This yields us 192.8k training, 2.4k validation, and 4.8k testing data pairs.

### B.5 2D Diffusion-Reaction (PDEBench)

2D Diffusion-Reaction (Takamoto et al., 2022) is a challenging SciML benchmark problem, modeling biological pattern formation. It is challenging because of the nonlinear coupling of the solution variables, viz. *activator u(x,y,t)* and *inhibitor v(x,y,t)* from the time-dependent PDE,

$$\frac{\partial u}{\partial t} = D_u \partial_{xx} u + D_u \partial_{yy} u + R_u(u, v), \quad \frac{\partial v}{\partial t} = D_v \partial_{xx} v + D_v \partial_{yy} v + R_v(u, v), \quad (-1,1)^2 \times (0, 5] \quad (10)$$

where $D_u$ and $D_v$ are diffusion coefficients for the activator and inhibitor, respectively, and the reaction functions $R_u$ and $R_v$ in Eqn. 10 are defined by the Fitzhugh-Nagumo equations as,

$$R_u(u, v) = u - u^3 - k - v, \qquad R_v(u, v) = u - v, \quad (11)$$

The simulation data is generated by setting $D_u = 1 \cdot 10^{-3}$, $D_v = 5 \cdot 10^{-3}$, and $k = 5 \cdot 10^{-3}$ with no-flow Neumann boundary condition. The training and test sets consist of 900 and 100 trajectories, respectively, where each simulation is of shape $(128 \times 128 \times 101 \times 2)$, indicating the solutions for 101 timesteps and a channel each for the activator ($u$) and inhibitor ($v$).

### B.6 3D Turbulent Radiative Mixing Layers (TRL3D, The Well)

In three-dimensional astrophysical environments, dense cold gas clouds (the "cold phase") move through a hotter, diffuse medium (the "hot phase"), generating turbulent mixing at their interfaces. This 3D turbulent mixing produces a multiscale intermediate-temperature layer, where gas rapidly cools radiatively due to the enhanced cooling efficiency at intermediate temperatures. As energy is lost via photon emission, the mixed gas condenses and accretes onto the cold phase.

- *Growth vs. Dissolution of Cold Clumps*
  - When radiative cooling dominates turbulent mixing ($t_{\text{cool}} \ll t_{\text{mix}}$), the cold phase grows as mixed gas rapidly cools and accretes onto cold clouds.
  - When mixing dominates ($t_{\text{cool}} \gg t_{\text{mix}}$), the cold phase evaporates into the hot medium.
- *Cooling-Mass Transfer Relation:* The volume-integrated cooling rate $\dot{E}_{\text{cool}}$ and mass transfer rate $\dot{M}$ from hot to cold phases scale as

$$\dot{E}_{\text{cool}} \propto \dot{M} \propto v_{\text{rel}}^{3/4} t_{\text{cool}}^{-1/4},$$

  where $v_{\text{rel}}$ is the *3D relative velocity* between phases, and $t_{\text{cool}}$ is the cooling timescale.

These 3D simulations explicitly model the competition between turbulent mixing (driven by shear, Kelvin-Helmholtz Instabilities (KHI), and turbulence) and radiative cooling, providing a more complete picture of multiphase gas dynamics in astrophysical environments (e.g., galactic halos, stellar winds, or the interstellar medium) (Fielding et al., 2020). The equations underlying these simulations are given by Ohana et al. (2024),

$$\frac{\partial \rho}{\partial t} + \nabla \cdot (\ \rho\ \ \mathbf{v}\ ) = 0$$

mass density

velocity field

$$\frac{\partial \rho \mathbf{v}}{\partial t} + \nabla \cdot (\ \rho\mathbf{v} \otimes \mathbf{v}\ +\ P\ ) = 0$$

momentum flux tensor

pressure tensor

$$\frac{\partial E}{\partial t} + \nabla \cdot \big((E+P)\mathbf{v}\big) = -\frac{E}{t_{\text{cool}}}$$

cooling timescale

$$E = P/(\ \gamma - 1) \qquad \text{where} \qquad \gamma = \frac{5}{3}$$

adiabatic index
($\frac{5}{3}$ for monatomic gas)

## C Baseline Neural Operator Models

We choose the following list of seven diverse, competitive, and recent state-of-the-art baselines.

### C.1 Modern U-Net

We benchmark the modern version of U-Net (https://github.com/pdearena/pdearena/tree/main/pdearena/modules) from PDEArena (Gupta & Brandstetter, 2023). Considering that we do not focus on testing the generalization capabilities of neural operators on a diverse set of PDE parameters, the parameter conditioning is skipped. The U-Net architecture models multi-scale spatio-temporal phenomena through a series of downsampling and an equal number of upsampling blocks, with the skip-connections passing information from the downsampling to the upsampling pass. We employ four levels of downsampling and set the channel multipliers to $(1, 2, 2, 3, 4)$ to maintain parameter parity with other baselines. Following prior studies (Lippe et al., 2023), the network directly predicts the residual to the next step instead of the actual solution, and then we down weight this output with the factor $\frac{3}{10}$. Note that this setting improves the results of the modern U-Net model on Kolmogorov Flow significantly (1-step nRMSE REL. % DIFF of 22%) compared to the reported results of NO-LIDK (Liu-Schiaffini et al., 2024) and hence is a strong baseline. The hyperparameters are listed in Appendix O.1.

### C.2 FNO: Fourier Neural Operators

As the original FNO architecture (Li et al., 2021) has been improved and optimized over the years, we utilize the current state-of-the-art implementation of FNO (https://github.com/neuraloperator/neuraloperator) from Kossaifi et al. (2024a) for Kolmogorov Flow 2D, CNS Turb 2D, Wave-Gauss 2D, and Compressible Euler four-quadrant Riemann problem 2D PDEs. However, we omitted tensor factorization techniques in our experiments so as to be able to directly compare our results with NO-LIDK (Liu-Schiaffini et al., 2024), which was also devoid of any such data compression strategies. We use the implementation of Takamoto et al. (2022) for the Diffusion-Reaction 2D problem following Liu-Schiaffini et al. (2024) for a fair comparison. The hyperparameters are provided in Appendix O.2.

### C.3 F-FNO: Factorized Fourier Neural Operators

Tran et al. (2023) propose to improve the original FNO architecture of Li et al. (2021) by factorizing the FFT computation for data with spatial dimensions of two or greater, among other contributions such as skip connections, Markov assumption, Gaussian noise, cosine learning rate schedule, and hence the model Factorized FNO. The factorization leads to a massive drop in the number of trainable parameters, resulting in a lightweight and efficient model. Therefore, we use this as one of the baselines. Having borrowed their open-source implementation (`https://github.com/alasdairtran/fourierflow`) and adapted to our PDE problems, we provide the hyperparameters we use in Appendix O.3.

### C.4 U-FNO: An Enhanced FNO for Multiphase Flow

Wen et al. (2022) propose U-FNO as an improved version of FNO specifically targeted for modeling multiphase flows. The network consists of six layers in total, three of which are standard Fourier layers from Li et al. (2021) and three UNet layers added to the Fourier layers. We adapt the open-source implementation (`https://github.com/gegewen/ufno`) provided by Wen et al. (2022) for 3D spatial data to our experiments on 2D spatial data. We tune the model hyperparameters, viz., learning rate, modes, and channel dimension, to approximately match the number of trainable parameters as the other baselines while also reaching a competitive accuracy. The full set of hyperparameters is provided in Appendix O.6.

### C.5 LSM: Latent Spectral Models

Wu et al. (2023) introduce a multi-scale neural operator architecture with the use of a hierarchical projection network mapping the high-dimensional input space into latent dimensions and propose to model the dynamics in this low-dimensional latent space. The essential component of LSM is the so-called Neural Spectral Block, which consists of an encode-process-decode style of processing. Considering the robustness of MSE to LSM's hyperparameters (cf. Appendix G in Wu et al. (2023)), such as scales, number of latent tokens, and number of basis operators, we use 5 scales, 4 latent tokens, and 12 basis operators in the hierarchical projection network. This yields 64, 128, 256, 512, and 512 channels for the five scales in question, starting with the full spatial resolution and initial channel dimensions of 64 for the 2D Kolmogorov Flow, CNS Turb 2D, and Wave-Gauss 2D problems. As for the Diffusion-Reaction 2D PDE, we use the initial channel dimensions of 32. Therefore, the channel dimensions for the five scales starting with the full incoming spatial resolution would be 32, 64, 128, 256, and 256, respectively. The other hyperparameters are provided in Appendix O.4.

### C.6 Transolver: Fast Transformer Solver for PDEs on General Geometries

Wu et al. (2024) design a transformer-based neural operator model, namely Transolver, to solve PDEs on both regular and irregular domains. The idea stems from the observation that self-attention over individual mesh points is unnecessary, expensive, and not scalable because it is not uncommon to have real-world data spanning tens of millions of points or higher, resulting from the discretization of the continuous physical fields. Towards this end, Transolver introduces a so-called 'Physics Attention' mechanism where the mesh points belonging to similar physical states are grouped into *slices* and the self-attention is computed over these *slices*. The resultant model achieves a near-linear complexity and scales to reasonably sized mesh points (e.g., 16,384 – 128×128). Therefore, to include a transformer-based model in the baselines, we benchmark this model on the 2D Kolmogorov Flow, CNS Turb 2D, Wave-Gauss 2D, and Diffusion-Reaction 2D PDEs by adapting the implementation provided in `https://github.com/thuml/Transolver/tree/main/PDE-Solving-StandardBenchmark`. However, we note that Transolver needs an extended training time of 500 epochs for convergence on the challenging Kolmogorov Flow problem, while the other baselines converge to an optimal loss value in just 136 epochs. The model architecture and training hyperparameters are listed in Appendix O.5.

### C.7 NO-LIDK: Neural Operators with Localized Integral and Differential Kernels

Noticing the deficiencies of FNO in learning local and finer scale features due to the absence of support for local receptive fields, Liu-Schiaffini et al. (2024) propose two additional discretization-invariant convolutional

layers (one for the differential operator and another for the local integral kernel operator) that can be placed in parallel to the (global) Fourier layers of (spherical) FNO. These additional pathways enrich the architecture with capabilities for local convolutions, which are realized through localized integral and differential kernels. The authors test the efficacy of these layers by conducting experiments on a wide range of two-dimensional PDE problems, such as Darcy Flow, turbulent Navier-Stokes, and Diffusion-Reaction on the 2D planar domain, and the Shallow Water Equations on the spherical computational domain. Their results significantly improve over the baseline FNO, ranging from 34% to 87% in terms of nRMSE, demonstrating the effectiveness and importance of these local operations for operator learning. Since our work also achieves local convolutions, albeit by decomposing the input spatial resolution into sub-domains (e.g., patches), we consider the NO-LIDK model as a closely related and competitive baseline. Considering that we report additional evaluation metrics, we train three variants of NO-LIDK (namely, (i) NO-LIDK$^*$ denoting only the presence of local integral kernel layers, (ii) NO-LIDK$^\diamond$ representing only the existence of differential kernel layers, and (iii) NO-LIDK$^\dagger$ indicating the use of both local integral kernel layers and differential kernel layers, in addition to the global FNO branch) on the INS Kolmogorov Flow 2D dataset and reproduce the reported results. Following the authors' suggestion, we use a radius cutoff of $0.05\pi$ for the local integral kernel layers and set the domain length to $[2\pi, 2\pi]$ (cf. Eqn 7). We borrow the other hyperparameters, such as the number of Fourier modes, hidden channels, epochs, learning rate, scheduler, and scheduler steps, for each of the model variants from their paper. The specific values used for each of these are provided in Appendix O.7, which also lists the training and model configuration settings, including the radius cutoff values for the CNS Turb 2D, Wave-Gauss 2D, and Compressible Euler four-quadrant Riemann problem 2D PDEs..

## D  Spectral Space Loss Functions

In this section, we focus on two loss functions that operate entirely in the frequency domain.

### D.1  Spectral Patch High-frequency Emphasized Residual Energy ($\mathbb{SPHERE}$) Loss

In congruence with the local branch of LOGLO-FNO, we propose a spectral loss localized to each of the sub-domains $P_m$ of the domain $D$ (§3.1).

Let $u_{pred}$ and $u_{gt}$ be the predictions and ground truths of non-overlapping patches of the solution in the domain, respectively. First, we compute the residual in the physical space: $\Delta u = u_{pred} - u_{gt}$. Second, these residuals of patches are transformed into the frequency domain using 2D DFT realized through FFT: $\Delta \hat{u}(k_x, k_y) = \mathcal{F}(\Delta u)$.

The $\mathbb{SPHERE}$ loss is then computed as the weighted squared magnitude of Fourier coefficients as

$$\mathbb{SPHERE} = \mathcal{W}(k_x, k_y) \odot |\Delta \hat{u}(k_x, k_y)|^2 \tag{12}$$

where the weight function $\mathcal{W}(k_x, k_y)$ emphasizing high-frequencies is defined as,

$$\mathcal{W}(k_x, k_y) = 1 + \alpha \left( \frac{\mathbb{FM}}{\max(\mathbb{FM}) + \epsilon} \right)^p, \qquad \mathbb{FM} = \sqrt{k_x^2 + k_y^2} \tag{13}$$

whereby $\alpha$ and $p$ control the emphasis on high-frequencies. For our experiments, we use $p = 2$.

The loss is finally *mean* or *sum* reduced over the channel dimensions and the number of patches, and added to the MSE loss during training.

```python
def compute_frequency_weights(alpha, p, psize):
    kx = torch.fft.fftfreq(psize, d=1.0).reshape(1, 1, 1, psize)
    ky = torch.fft.fftfreq(psize, d=1.0).reshape(1, 1, psize, 1)
    freq_magnitude = torch.sqrt(kx**2 + ky**2)

    # Normalize and define frequency weighting function
    freq_magnitude /= (freq_magnitude.max() + 1e-8)
    return 1 + alpha * (freq_magnitude ** p)

def extract_patches(x, psize):
    # Reshape to (nb*nt*nc, 1, nx, ny) to apply unfold over spatial dims
    nb, nx, ny, nt, nc = x.shape
    x_reshp = x.permute(0, 3, 4, 1, 2).reshape(nb * nt * nc, 1, nx, ny)

    # Extract patches using unfold
    patches = F.unfold(x_reshp, kernel_size=psize, stride=psize)

    # Reshape back to (nb, nt, nc, num_patches, patch_size, patch_size)
    num_ptch = patches.shape[-1]
    patches = patches.view(nb, nt, nc, num_ptch, psize, psize)

    return patches

def SPHERELoss(preds, target, alpha, p, patch_size, reduction="mean"):
    # extract non-overlapping patches
    pred_patches = extract_patches(pred, patch_size)
    target_patches = extract_patches(target, patch_size)

    # Compute difference (i.e., error) in real space once
    diff_patches = pred_patches - target_patches

    # Use vmap to apply FFT over temporal and channel dims
    def fft_and_weight(diff):
        diff_fft = torch.fft.fft2(diff, norm="ortho")
        weight = compute_frequency_weights(alpha, p, patch_size)
        return weight * (diff_fft.real ** 2 + diff_fft.imag ** 2)

    # Apply vmap once along both (nt, nc) dimensions
    fft_loss = torch.vmap(
                        torch.vmap(fft_and_weight, in_dims=1),
                        in_dims=2
                      )(diff_patches)

    # Aggregate spectral loss over all patches, time, and channels
    spect_loss = fft_loss.mean(dim=(1, 2, 3))

    if reduction == "mean":
        return spect_loss.mean()
    else:
        return spect_loss.sum()
```

Unlike the function *compute_frequency_weights*(...) shown here for expediency, in practice, we compute the *weight* only once per patch size and cache it for efficiency reasons.

## D.2   Radially Binned Spectral Energy Errors as a Frequency-aware Loss

We elaborate on the frequency-aware loss term based on radially binning of energy spectra errors introduced in §3.2 by considering a 2D spatiotemporal domain.

The detailed procedure to compute the frequency loss for 2D spatial data is shown in the algorithm below.

---

**Algorithm 1** Frequency-aware Loss based on Radially Binned Spectral Errors

---

1: **Input:** Prediction $\tilde{\mathbf{Y}} \in \mathbb{R}^{N_b \times N_c \times N_x \times N_y \times N_t}$ and Target $\mathbf{Y} \in \mathbb{R}^{N_b \times N_c \times N_x \times N_y \times N_t}$, $N_x$ and $N_y$ be the spatial resolutions, $N_b$ the batch size, $N_t$ the total timesteps, $N_c$ the number of physical fields, $L_x$ and $L_y$ the length of the spatial domain along the respective axes of the 2D domain, and $\mathbf{x}$ and $\mathbf{y}$ be the 2D mesh grid of frequency indices. Let $iL$ and $iH$ be the radial bin cutoff indices, i.e., $iL$ is the count of low-frequency bins, and $iH$ is the count of low+mid-frequency bins.

       // $\mathbf{k_x}, \mathbf{k_y}$ *below are frequency domain indices (e.g., $k_x \in [0, N_x/2 - 1]$, $k_y \in [0, N_y/2 - 1]$)*

2: $\boldsymbol{E}_F(b, c, k_x, k_y, t) \leftarrow \mid \mathcal{F}(\mathbf{Y}_{b,c,t} - \tilde{\mathbf{Y}}_{b,c,t})(k_x, k_y) \mid^2$       `{Energy Spectra Error (ESE)}`

3: $\mathcal{R}(k_x, k_y) \leftarrow \left\lfloor \sqrt{\mathbf{x}^2 + \mathbf{y}^2} \right\rfloor$                                `{Binned radial distances}`

4: $\mathcal{M} \leftarrow \max_{k_x, k_y}(\mathcal{R}(k_x, k_y))$                                `{Max radius}`

5: $\boldsymbol{S}_{rad}(b, c, r, t) \leftarrow \sum_{\substack{(k_x, k_y) \text{ s.t.} \\ \mathcal{R}(k_x, k_y) = r}} \boldsymbol{E}_F(b, c, k_x, k_y, t)$ for $r = 0 \dots \mathcal{M}$     `{Sum ESE into bin `$r$`, per `$b, c, t$`}`

6: $\overline{\boldsymbol{S}}_{rad}(c, r, t) \leftarrow \text{mean}_b(\boldsymbol{S}_{rad}(b, c, r, t))$            `{Average summed ESE per bin over batch}`

7: $\overline{\boldsymbol{E}}_F(c, r, t) \leftarrow \sqrt{\overline{\boldsymbol{S}}_{rad}(c, r, t)} \cdot \left(\frac{L_x}{N_x}\right) \cdot \left(\frac{L_y}{N_y}\right)$       `{Normalize 1D radial error spectrum}`

8: $\text{err}_F^{\text{low}}(c, t) \leftarrow \text{mean}_{r \in [0, iL-1]}(\overline{\boldsymbol{E}}_F(c, r, t))$          `{Low-frequency band errors}`

9: $\text{err}_F^{\text{mid}}(c, t) \leftarrow \text{mean}_{r \in [iL, iH-1]}(\overline{\boldsymbol{E}}_F(c, r, t))$        `{Mid-frequency band errors}`

10: $\text{err}_F^{\text{high}}(c, t) \leftarrow \text{mean}_{r \in [iH, \mathcal{M}]}(\overline{\boldsymbol{E}}_F(c, r, t))$        `{High-frequency band errors}`

---

The concrete steps are listed in the PyTorch-style pseudocode below. Since we use `torch.fft.fftn` in the '*backward*' normalization mode, no normalization term is included. Let *preds* and *target* be the predictions and ground truth, respectively, and the 2D spatial data has a single channel and timestep. The computation for multi-channel and multi-timestep data is straightforward since they are obtained independently for each physical variable and timestep in the input.

```python
def RadialBinnedSpectralLoss(preds, target):
    # input data shape and params
    nb, nc, nx, ny, nt = target.size()
    iLow, iHigh = 4, 12
    Lx, Ly = 1.0, 1.0

    # Compute error in Fourier space
    err_phys = preds - target
    err_fft = torch.fft.fftn(err_phys, dim=[2, 3])
    err_fft_sq = torch.abs(err_fft)**2
    err_fft_sq_h = err_fft_sq[Ellipsis, :nx//2, :ny//2, :]

    # Create radial indices
    x = torch.arange(nx//2)
    y = torch.arange(ny//2)
    X, Y = torch.meshgrid(x, y, indexing="ij")
    radii = torch.sqrt(X**2 + Y**2).floor().to(torch.int) # Radial dist.
    max_radius = int(torch.max(radii))

    # flatten radii for binary mask
    radii_flat = radii.flatten()  # (nx//2 * ny//2)

    # Spatially flatten Fourier space error; (nb, nc, nx//2 * ny//2, nt)
    err_fft_sq_flat = err_fft_sq_h.contiguous().reshape(nb, nc, -1, nt)

    # initialize output tensor to hold the Fourier error
    # for each radial bin at distance r from the origin
    err_F_vect_full = torch.zeros(nb, nc, max_radius + 1, nt)

    # Apply 'index_add_' for all radii and accumulate the errors
    valid_r = radii_flat <= max_radius  # binary mask to find valid radii
```

```
33    # Sum for all valid radial indices
34    err_F_vect_full.index_add_(2,
35                               radii_flat[valid_r],
36                               err_fft_sq_flat[:, :, valid_r]
37                               )
38
39    # Normalize & compute mean over batch; => (nc, min(nx//2, ny//2), nt)
40    nrm = (Lx/nx) * (Ly/ny)
41    _err_F = torch.sqrt(torch.mean(err_F_vect_full, dim=0)) * nrm
42
43    # Classify Fourier space error into three bands
44    err_F = torch.zeros([nc, 3, nt])
45    err_F[:, 0] += torch.mean(_err_F[:, :iLow], dim=1)      # low freqs
46    err_F[:, 1] += torch.mean(_err_F[:, iLow:iHigh], dim=1) # mid freqs
47    err_F[:, 2] += torch.mean(_err_F[:, iHigh:], dim=1)     # high freqs
48
49    # mean or sum over channels and time dimensions
50    if reduction == "mean":
51        freq_loss = torch.mean(err_F, dim=[0, -1])
52    elif reduction == "sum":
53        freq_loss = torch.sum(err_F, dim=[0, -1])
```

## D.3  Visualizing Radially Binned Energy of Spectral Errors of Baselines and LOGLO-FNO

In this section, we provide a visualization of the proposed radially binned energy of the spectral errors for 2D spatial data. In the interest of providing an uncluttered representation, we only show every $2^{nd}$ radial bin in the Figures 9, 10, 11, and 12.

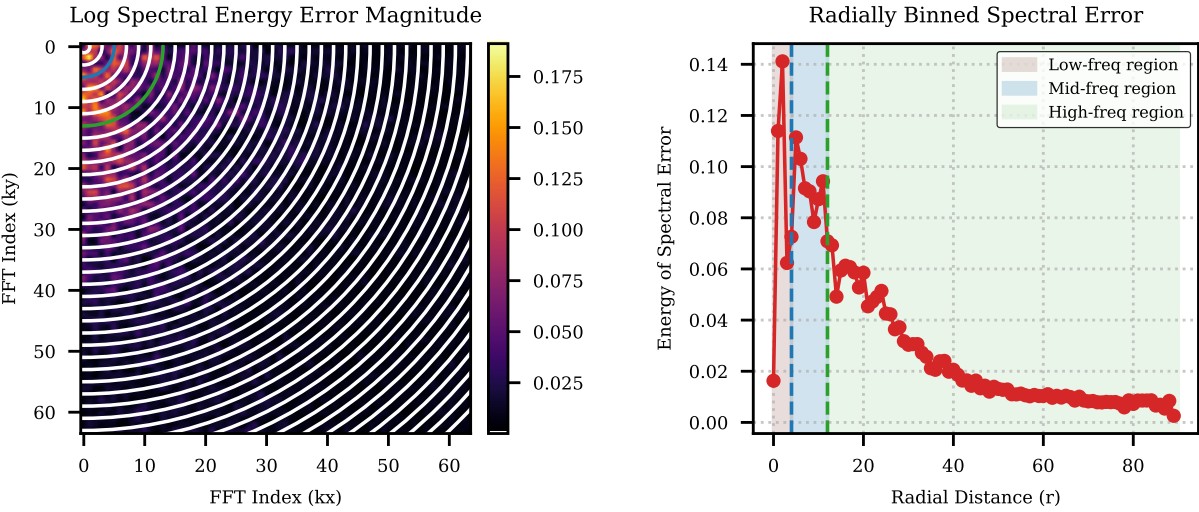

Figure 9: Visualization of the radially binned energy of spectral errors of baseline FNO predictions on a random trajectory and timestep from the test set of Kolmogorov Flow ($Re = 5k$) (Li et al., 2022b). The left plot depicts the radial bins and the location of the boundaries of mid and high-frequency groups. Note that we show only every other radial bin. The right plot visualizes the radially binned spectral error as a line plot over the radial distance from the DC component. The dashed blue and green vertical lines indicate the starting locations of the mid and high-frequency regions, respectively.

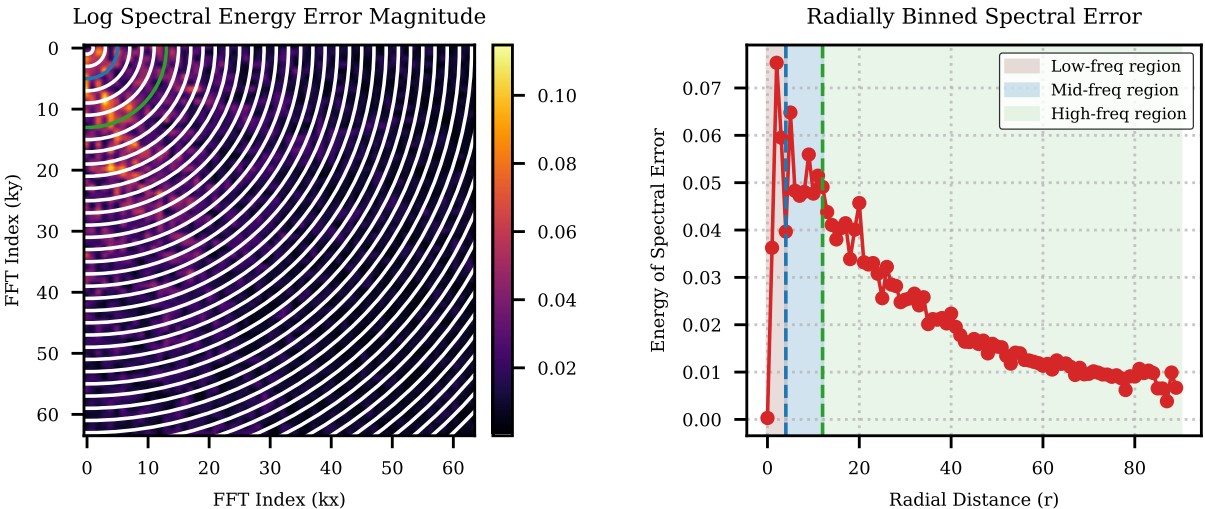

Figure 10: Visualization of the radially binned energy of spectral errors of LOGLO-FNO predictions on the same random trajectory and timestep (as that of the one for baseline FNO) from the test set of Kolmogorov Flow ($Re = 5k$) (Li et al., 2022b). The left plot depicts the radial bins and the location of the boundaries of mid and high-frequency groups. Note that we show only every other radial bin. The right plot visualizes the radially binned spectral error as a line plot over the radial distance from the DC component. The dashed blue and green vertical lines indicate the starting locations of the mid and high-frequency regions, respectively.

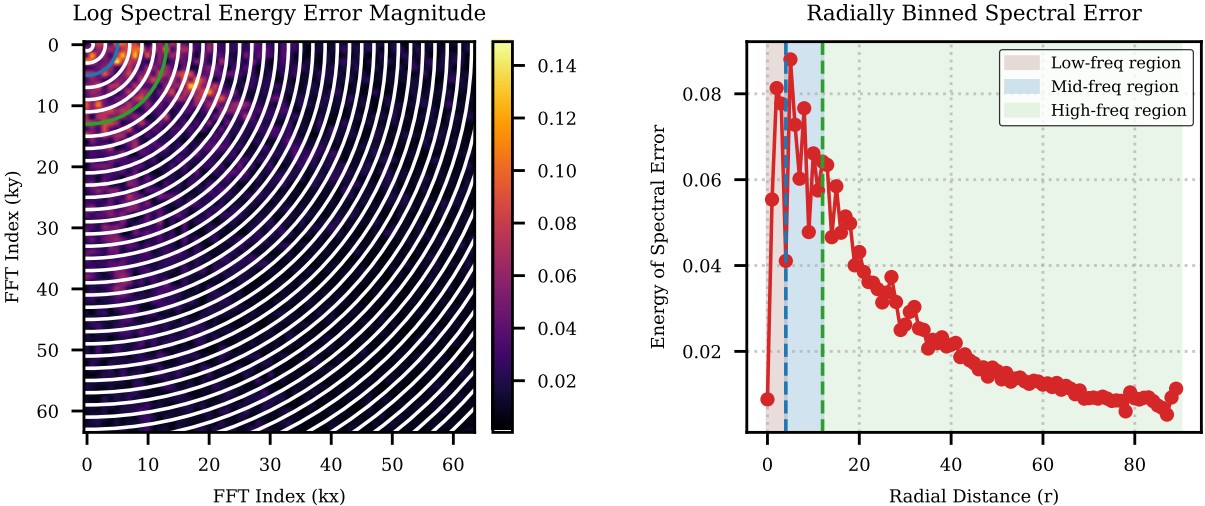

Figure 11: Visualization of the radially binned energy of spectral errors of NO-LIDK predictions on the same random trajectory and timestep (as that of the one for baseline FNO) from the test set of Kolmogorov Flow ($Re = 5k$) (Li et al., 2022b). The left plot depicts the radial bins and the location of the boundaries of mid and high-frequency groups. Note that we show only every other radial bin. The right plot visualizes the radially binned spectral error as a line plot over the radial distance from the DC component. The dashed blue and green vertical lines indicate the starting locations of the mid and high-frequency regions, respectively.

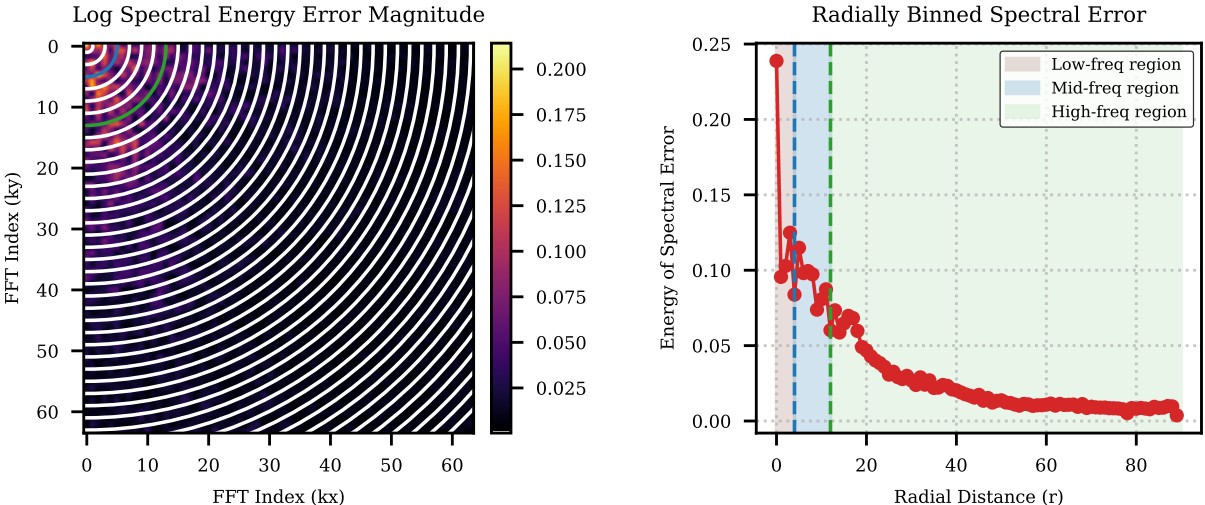

Figure 12: Visualization of the radially binned energy of spectral errors of modern U-Net predictions on the same random trajectory and timestep (as that of the one for baseline FNO) from the test set of Kolmogorov Flow ($Re = 5k$) (Li et al., 2022b). The left plot depicts the radial bins and the location of the boundaries of mid and high-frequency groups. Note that we show only every other radial bin. The right plot visualizes the radially binned spectral error as a line plot over the radial distance from the DC component. The dashed blue and green vertical lines indicate the starting locations of the mid and high-frequency regions, respectively.

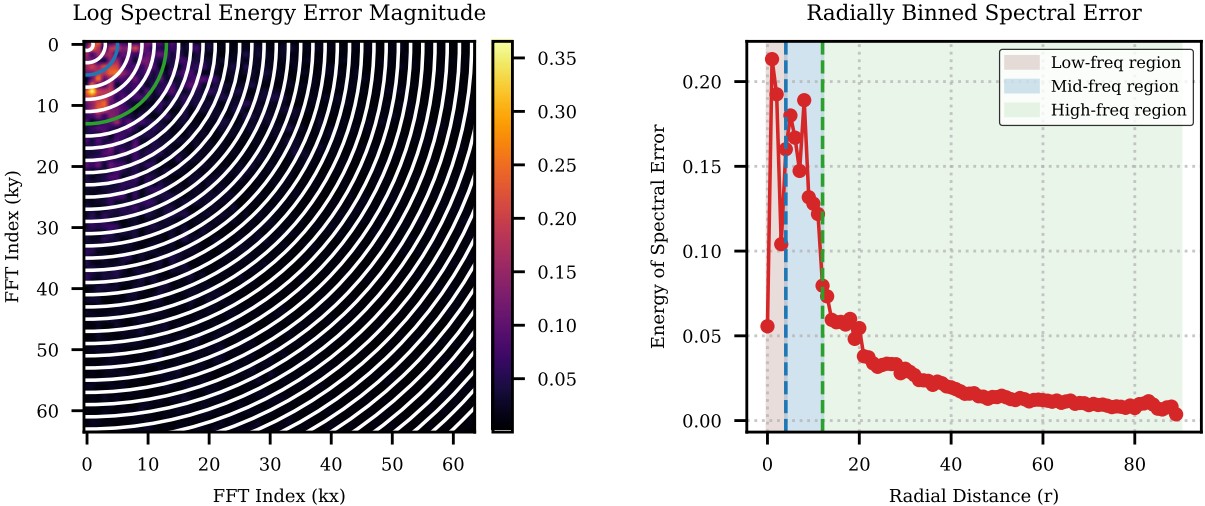

Figure 13: Visualization of the radially binned energy of spectral errors of LSM predictions on the same random trajectory and timestep (as that of the one for baseline FNO) from the test set of Kolmogorov Flow ($Re = 5k$) (Li et al., 2022b). The left plot depicts the radial bins and the location of the boundaries of mid and high-frequency groups. Note that we show only every other radial bin. The right plot visualizes the radially binned spectral error as a line plot over the radial distance from the DC component. The dashed blue and green vertical lines indicate the starting locations of the mid and high-frequency regions, respectively.

# E  High-Frequency Feature Adaptive Gaussian Noise

In many machine learning and signal processing tasks, high-frequency (HF) features often capture fine-grained details or noise-like patterns in the data. Introducing controlled noise based on these features during training can improve model robustness, prevent overfitting, and enhance generalization. It has been shown in prior surrogate modeling investigations that adding a small amount of Gaussian noise helps with long rollouts (Pfaff et al., 2021; Stachenfeld et al., 2022) and stabilized training (Tran et al., 2023). However, traditional Gaussian noise, which is static and independent of the input data, may not adequately capture the variability inherent in turbulent PDEs and physics simulation data rich in high frequencies. To address this shortcoming, we propose a novel method to generate dynamic Gaussian noise that adapts to the statistical properties of the input HF features. Specifically, we obtain the high-frequency feature adaptive Gaussian noise by scaling ($\alpha$) the standard Gaussian noise $\mathcal{N}(0,1)$ based on the mean ($\mu$) and standard deviation ($\sigma$) of the HF features.

$$\mathbf{N}_{dynamic} = \mu + \alpha \cdot \sigma \cdot \mathbf{N}, \qquad \mathbf{N} \sim \mathcal{N}(0,1) \tag{14}$$

**High-Frequency Feature Adaptive Gaussian Noise**

Let $\mathbf{X}^{hf} \in \mathbb{R}^{N_b \times N_x \times N_y \times (N_t \cdot N_c)}$ denote the input tensor of HF features, where:

- $N_b$ is the batch size,

- $N_x$ and $N_y$ are the resolutions of the spatial dimensions, and

- $N_t \cdot N_c$ represents the combined temporal and channel dimensions.

The high-frequency feature adaptive Gaussian noise $\mathbf{N}_{dynamic}$ is then computed as follows:

1. *Compute (per sample) Mean ($\mu_b$) and Standard Deviation ($\sigma_b$) of High-Frequency Features*

$$\mu_b = \frac{1}{N_x \cdot N_y \cdot N_t \cdot N_c} \sum_{i=1}^{N_x} \sum_{j=1}^{N_y} \sum_{k=1}^{N_t N_c} \mathbf{X}^{\text{hf}}_{b,i,j,k}$$

$$\sigma_b = \sqrt{\frac{1}{N_x \cdot N_y \cdot N_t \cdot N_c} \sum_{i=1}^{N_x} \sum_{j=1}^{N_y} \sum_{k=1}^{N_t N_c} (\mathbf{X}^{\text{hf}}_{b,i,j,k} - \mu)^2 + \epsilon}$$

where $\epsilon$ is a small constant added for numerical stability, and $\mu$ and $\sigma$ are obtained by stacking the per sample statistics along the batch dimension.

2. *Generate Standard Gaussian Noise*
$$\mathbf{N} \sim \mathcal{N}(0,1)$$

3. *Scale Noise Dynamically*
$$\mathbf{N}_{dynamic} = \mu + \alpha \cdot \sigma \cdot \mathbf{N}$$

where $\alpha$ is a small value such as 0.025 and $\mathbf{N}_{dynamic}$ has the same shape as the input $\mathbf{X}^{hf}$.

$\mathbf{N}_{dynamic}$ can now be added to the batch of inputs to the global and local branches during the training phase of LOGLO-FNO.

# F    Complexity Analysis of Local and Global Spectral Convolutions

## F.1    2D Spectral Convolutions

**Global branch.**    FLOPs for FFT and IFFT computations (Cooley & Tukey, 1965) on the full 2D spatial resolution.

---

**Global Branch FLOPs Calculation for FFT & IFFT in Global Spectral Convolution 2D**

Let $\mathbf{X} \in \mathbb{R}^{N_b \times N_c \times N_x \times N_y}$ denote the input tensor for the 2D global spectral convolution module, where:

- $N_b$ is the batch size,

- $N_x$ and $N_y$ are the resolutions of the spatial dimensions,

- $N_c$ represents the total number of hidden channels.

$$\text{FLOPs}_{\text{FFT}} = 5 \cdot N_b \cdot C_{\text{in}} \cdot N_x \cdot N_y \cdot \log_2(N_x \cdot N_y)$$
$$\text{FLOPs}_{\text{IFFT}} = 5 \cdot N_b \cdot C_{\text{out}} \cdot N_x \cdot N_y \cdot \log_2(N_x \cdot N_y)$$

where $c_{\text{in}}$ and $c_{\text{out}}$ are typically of the same dimension and are referred to as either width or embedding/hidden channel dimension.

---

Therefore, FFT computation on the global branch with the full 2D spatial resolution has a complexity of $\mathcal{O}\Big(N_x \cdot N_y \cdot \log_2(N_x \cdot N_y)\Big)$ per channel, making it expensive for large $N_x$ and $N_y$ such as $2048 \times 2048$ or higher spatial resolutions.

**Local branch.**    FLOPs for FFT and IFFT computations (Cooley & Tukey, 1965) on the domain decomposed 2D spatial resolution, for instance, with non-overlapping patches.

---

**Local Branch FLOPs Calculation for FFT & IFFT in Local Spectral Convolution 2D**

Let $\hat{\mathbf{X}} \in \mathbb{R}^{N_b \times N_c \times N_p \times P_{size} \times P_{size}}$ denote the input tensor for the 2D local spectral convolution module, where:

- $N_p$ is the number of patches obtained by $\frac{N_x \cdot N_y}{P_{size}^2}$,

- $N_x$ and $N_y$ are the resolutions of the spatial dimensions,

- $P_{size} \times P_{size}$ is the patch size (e.g., $16 \times 16$ ),

- $N_c$ represents the width or the number of hidden channels.

$$\begin{aligned}\text{FLOPs}_{\text{FFT}} &= 5 \cdot N_b \cdot C_{\text{in}} \cdot N_p \cdot P_{size} \cdot P_{size} \cdot \log_2(P_{size} \cdot P_{size}) \\ &= 5 \cdot N_b \cdot C_{\text{in}} \cdot N_x \cdot N_y \cdot \log_2(P_{size} \cdot P_{size})\end{aligned}$$

$$\begin{aligned}\text{FLOPs}_{\text{IFFT}} &= 5 \cdot N_b \cdot C_{\text{out}} \cdot N_p \cdot P_{size} \cdot P_{size} \cdot \log_2(P_{size} \cdot P_{size}) \\ &= 5 \cdot N_b \cdot C_{\text{out}} \cdot N_x \cdot N_y \cdot \log_2(P_{size} \cdot P_{size})\end{aligned}$$

where $c_{\text{in}}$ and $c_{\text{out}}$ are typically set to be of the same dimension (e.g., 128).

---

Thus, the FFT and IFFT computations (Cooley & Tukey, 1965) on the local branch operating on the patches has a computational complexity of $\mathcal{O}\Big(N_x \cdot N_y \cdot \log_2(P_{size} \cdot P_{size})\Big)$ per channel. Since $\mathcal{O}\Big(N_x \cdot N_y \cdot \log_2(P_{size} \cdot$

$P_{size})\Big) \ll \mathcal{O}\Big(N_x \cdot N_y \cdot \log_2(N_x \cdot N_y)\Big)$ in practice, the FFT computation is significantly cheaper in the local branch. Moreover, it is highly parallelizable when computing FFTs since each patch can be processed independently, leveraging modern accelerators such as GPUs and TPUs.

### F.2 3D Spectral Convolutions

**Global branch.** FLOPs for FFT and IFFT computations (Cooley & Tukey, 1965) on the full 3D spatial resolution.

---

**Global Branch FLOPs Calculation for FFT & IFFT in Global Spectral Convolution 3D**

Let $\mathbf{X} \in \mathbb{R}^{N_b \times N_c \times N_x \times N_y \times N_z}$ denote the input tensor for the 3D global spectral convolution module, where:

- $N_b$ is the batch size,

- $N_x$, $N_y$, and $N_z$ are the resolutions of the spatial dimensions,

- $N_c$ represents the total number of hidden channels.

$$\text{FLOPs}_{\text{FFT}} = 5 \cdot N_b \cdot C_{\text{in}} \cdot N_x \cdot N_y \cdot N_z \cdot \log_2(N_x \cdot N_y \cdot N_z)$$

$$\text{FLOPs}_{\text{IFFT}} = 5 \cdot N_b \cdot C_{\text{out}} \cdot N_x \cdot N_y \cdot N_z \cdot \log_2(N_x \cdot N_y \cdot N_z)$$

where $c_{\text{in}}$ and $c_{\text{out}}$ are typically of the same dimension and are referred to as either width or embedding/hidden channel dimension.

---

Therefore, FFT computation on the global branch with the full 3D spatial resolution has a complexity of $\mathcal{O}\Big(N_x \cdot N_y \cdot N_z \cdot \log_2(N_x \cdot N_y \cdot N_z)\Big)$ per channel, making it highly expensive for large values of $N_x$, $N_y$, and $N_z$ such as $512 \times 512 \times 512$ or further higher spatial resolutions.

**Local branch.** FLOPs for FFT and IFFT computations (Cooley & Tukey, 1965) on the domain decomposed 3D spatial resolution, for instance, with non-overlapping patches.

---

**Local Branch FLOPs Calculation for FFT & IFFT in Local Spectral Convolution 3D**

Let $\hat{\mathbf{X}} \in \mathbb{R}^{N_b \times N_c \times N_p \times P_{size} \times P_{size} \times P_{size}}$ denote the input tensor for the 3D local spectral convolution module, where:

- $N_p$ is the number of patches obtained by $\frac{N_x \cdot N_y \cdot N_z}{P_{size}^3}$,

- $N_x$, $N_y$, and $N_z$ are the resolutions of the spatial dimensions,

- $P_{size} \times P_{size} \times P_{size}$ is the patch size (e.g., $16 \times 16 \times 16$, $32 \times 32 \times 32$, etc.),

- $N_c$ represents the width or number of hidden channels.

$$\begin{aligned} \text{FLOPs}_{\text{FFT}} &= 5 \cdot N_b \cdot C_{\text{in}} \cdot N_p \cdot P_{size} \cdot P_{size} \cdot P_{size} \cdot \log_2(P_{size} \cdot P_{size} \cdot P_{size}) \\ &= 5 \cdot N_b \cdot C_{\text{in}} \cdot N_x \cdot N_y \cdot N_z \cdot \log_2(P_{size} \cdot P_{size} \cdot P_{size}) \end{aligned}$$

$$\begin{aligned} \text{FLOPs}_{\text{IFFT}} &= 5 \cdot N_b \cdot C_{\text{out}} \cdot N_p \cdot P_{size} \cdot P_{size} \cdot P_{size} \cdot \log_2(P_{size} \cdot P_{size} \cdot P_{size}) \\ &= 5 \cdot N_b \cdot C_{\text{out}} \cdot N_x \cdot N_y \cdot N_z \cdot \log_2(P_{size} \cdot P_{size} \cdot P_{size}) \end{aligned}$$

where $c_{\text{in}}$ and $c_{\text{out}}$ are typically set to be of the same dimension (e.g., 64, 128, etc.).

---

Thus, the FFT and IFFT computations (Cooley & Tukey, 1965) on the local branch operating on 3D patches has a computational complexity of $\mathcal{O}\left(N_x \cdot N_y \cdot N_z \cdot \log_2(P_{size} \cdot P_{size} \cdot P_{size})\right)$ per channel. Since $\mathcal{O}\left(N_x \cdot N_y \cdot N_z \cdot \log_2(P_{size} \cdot P_{size} \cdot P_{size})\right) \ll \mathcal{O}\left(N_x \cdot N_y \cdot N_z \cdot \log_2(N_x \cdot N_y \cdot N_z)\right)$ in practice, the FFT computation is significantly cheaper in the local branch. Moreover, it is highly parallelizable when computing FFTs since each patch can be processed independently, leveraging modern accelerators such as GPUs and TPUs.

## G Detailed Quantitative Results and Analysis

### G.1 Evaluation Metrics

Following prior research efforts (Takamoto et al., 2022; Wan et al., 2023), we evaluate our models on a wide range of metrics, viz. RMSE, nRMSE, fRMSE (low, mid, high), MaxError, MELR, and WLR. In addition, we introduce REL. % DIFF (RPD) to report *relative* performance improvement w.r.t a baseline. For the sake of completeness, we explain the computation of the metrics briefly.

#### G.1.1 Normalized Root Mean Square Error (nRMSE) a.k.a. Normalized $L_2$ a.k.a. Relative $L_2$ Error

The normalized $L_2$ norm of a vector $\mathbf{e} = (e_1, e_2, \ldots, e_n)^T$ is given by:

$$\text{Normalized } L_2 = \frac{\|\mathbf{e}\|_2}{\|\mathbf{y}\|_2} = \frac{\sqrt{\sum_{i=1}^{n} e_i^2}}{\sqrt{\sum_{i=1}^{n} y_i^2}},$$

where $\mathbf{y} = (y_1, y_2, \ldots, y_n)^T$ is the reference vector.

The normalized RMSE is defined as:

$$\text{Normalized RMSE} = \frac{\sqrt{\frac{1}{n} \sum_{i=1}^{n} e_i^2}}{\sqrt{\frac{1}{n} \sum_{i=1}^{n} y_i^2}} = \frac{\sqrt{\sum_{i=1}^{n} e_i^2}}{\sqrt{\sum_{i=1}^{n} y_i^2}}$$

The factor $\frac{1}{\sqrt{n}}$ cancels out in the numerator and denominator, leading to:

$$\text{Normalized RMSE} = \text{Normalized } L_2.$$

Given a prediction tensor $\hat{\mathbf{Y}} \in \mathbb{R}^{N_b \times N_c \times N_x \times N_y \times N_t}$ and a target tensor $\mathbf{Y} \in \mathbb{R}^{N_b \times N_c \times N_x \times N_y \times N_t}$, where $b = 1, \ldots, N_b$, $c = 1, \ldots, N_c$, $x = 1, \ldots, N_x$, $y = 1, \ldots, N_y$, and $t = 1, \ldots, N_t$. The nRMSE is computed as follows:

1. Mean Squared Error over the spatial dimensions:

$$\text{MSE}_{b,c,t} = \frac{1}{N_x \cdot N_y} \sum_{x=1}^{N_x} \sum_{y=1}^{N_y} (\hat{\mathbf{Y}}_{b,c,x,y,t} - \mathbf{Y}_{b,c,x,y,t})^2$$

2. Root Mean Squared Error over the spatial dimensions:

$$\text{RMSE}_{b,c,t} = \sqrt{\frac{1}{N_x \cdot N_y} \sum_{x=1}^{N_x} \sum_{y=1}^{N_y} (\hat{\mathbf{Y}}_{b,c,x,y,t} - \mathbf{Y}_{b,c,x,y,t})^2}$$

3. Normalization Factor:

$$\text{Normalization Factor}_{b,c,t} = \sqrt{\frac{1}{N_x \cdot N_y} \sum_{x=1}^{N_x} \sum_{y=1}^{N_y} \mathbf{Y}_{b,c,x,y,t}^2}$$

4. Normalized RMSE (nRMSE):

$$\text{nRMSE}_{b,c,t} = \frac{\text{RMSE}_{b,c,t}}{\text{Normalization Factor}_{b,c,t}}$$

5. Mean nRMSE over the batch, channel, and temporal dimensions:

$$\text{nRMSE} = \frac{1}{N_b \cdot N_c \cdot N_t} \sum_{b=1}^{N_b} \sum_{c=1}^{N_c} \sum_{t=1}^{N_t} \text{nRMSE}_{b,c,t}$$

### G.1.2   Max Error

Considering 2D spatial data, we can deduce that Takamoto et al. (2022) define the maximum or the worst case error in the entire test set as,

$$\text{MaxError} = \frac{1}{N_c \cdot N_t} \sum_{c=1}^{N_c} \sum_{t=1}^{N_t} \left( \max_{b,x,y} \left| \hat{\mathbf{Y}}_{b,c,x,y,t} - \mathbf{Y}_{b,c,x,y,t} \right| \right)$$

### G.1.3   RMSE at the Boundaries of the Spatial Domain (bRMSE)

Again, taking 2D spatial data as an exemplary case, we write the formula for bRMSE from Takamoto et al. (2022) as,

1. Compute Sum of Squared Errors (SSE) on the Boundaries of the Spatial Domain:

$$
\begin{aligned}
&\overset{\text{\% left vertical boundary}}{} \qquad \overset{\text{\% right vertical boundary}}{}\\
\mathrm{SSE}_{b,c,t} = &\sum_{y=1}^{N_y}\left(\hat{\mathbf{Y}}_{b,c,1,y,t} - \mathbf{Y}_{b,c,1,y,t}\right)^2 + \sum_{y=1}^{N_y}\left(\hat{\mathbf{Y}}_{b,c,N_x,y,t} - \mathbf{Y}_{b,c,N_x,y,t}\right)^2 +\\
&\sum_{x=1}^{N_x}\left(\hat{\mathbf{Y}}_{b,c,x,1,t} - \mathbf{Y}_{b,c,x,1,t}\right)^2 + \sum_{x=1}^{N_x}\left(\hat{\mathbf{Y}}_{b,c,x,N_y,t} - \mathbf{Y}_{b,c,x,N_y,t}\right)^2\\
&\underset{\text{\% bottom horizontal boundary}}{} \quad \underset{\text{\% top horizontal boundary}}{}
\end{aligned}
$$

2. Compute Per-Sample Boundary RMSE:

$$
\mathrm{bRMSE}_{b,c,t} = \sqrt{\frac{\mathrm{SSE}_{b,c,t}}{2N_x + 2N_y}}
$$

3. Compute Final bRMSE:

$$
\mathrm{bRMSE} = \frac{1}{N_b \cdot N_c \cdot N_t}\sum_{b=1}^{N_b}\sum_{c=1}^{N_c}\sum_{t=1}^{N_t}\mathrm{bRMSE}_{b,c,t}
$$

Or more succinctly put,

$$
\mathrm{bRMSE} = \frac{1}{N_b N_c N_t}\sum_{b=1}^{N_b}\sum_{c=1}^{N_c}\sum_{t=1}^{N_t}\sqrt{\frac{1}{2N_x + 2N_y}\left[\sum_{y=1}^{N_y}(\Delta_{b,c,1,y,t}^2 + \Delta_{b,c,N_x,y,t}^2) + \sum_{x=1}^{N_x}(\Delta_{b,c,x,1,t}^2 + \Delta_{b,c,x,N_y,t}^2)\right]}
$$

$$
\text{where } \Delta_{b,c,x,y,t} = \left(\hat{\mathbf{Y}}_{b,c,x,y,t} - \mathbf{Y}_{b,c,x,y,t}\right)
$$

### G.1.4   RMSE of Conserved Physical Quantities (cRMSE)

We again consider 2D spatial data for illustrative purposes here since computing the same metric for 3D spatial data is straightforward. Takamoto et al. (2022) define cRMSE as follows.

1. Compute the Error of Spatial Sum ($\Delta_{\mathbf{S}}$):

For each sample $b$, channel $c$, and timestep $t$, calculate the difference between the spatial sum of the predicted field and the spatial sum of the target field.

$$
\Delta_{\mathbf{S},b,c,t} = \sum_{x=1}^{N_x}\sum_{y=1}^{N_y}\left(\hat{\mathbf{Y}}_{b,c,x,y,t} - \mathbf{Y}_{b,c,x,y,t}\right)
$$

2. Compute Normalized cRMSE:

$$\overline{\text{cRMSE}}_{c,t} = \frac{1}{N_x \cdot N_y} \sqrt{\frac{1}{N_b} \sum_{b=1}^{N_b} (\Delta_{\mathbf{S},b,c,t})^2}$$

3. Compute Final cRMSE:

$$\text{cRMSE} = \frac{1}{N_c \cdot N_t} \sum_{c=1}^{N_c} \sum_{t=1}^{N_t} \overline{\text{cRMSE}}_{c,t}$$

Alternatively, as a one-line expression:

$$\text{cRMSE} = \frac{1}{N_c \cdot N_t \cdot N_x \cdot N_y} \sum_{c=1}^{N_c} \sum_{t=1}^{N_t} \sqrt{\frac{1}{N_b} \sum_{b=1}^{N_b} \left( \sum_{x=1}^{N_x} \sum_{y=1}^{N_y} \left( \hat{\mathbf{Y}}_{b,c,x,y,t} - \mathbf{Y}_{b,c,x,y,t} \right) \right)^2}$$

### G.1.5 Variance-scaled Root Mean Square Error (vRMSE)

Ohana et al. (2024) introduce vRMSE, which is merely RMSE normalized by the variance of the ground truth. Considering 3D spatial data,

$$\text{MSE}_{b,c,t} = \frac{1}{N_x \cdot N_y \cdot N_z} \sum_{x=1}^{N_x} \sum_{y=1}^{N_y} \sum_{z=1}^{N_z} (\hat{\mathbf{Y}}_{b,c,x,y,z,t} - \mathbf{Y}_{b,c,x,y,z,t})^2$$

$$\text{RMSE}_{b,c,t} = \sqrt{\frac{1}{N_x \cdot N_y \cdot N_z} \sum_{x=1}^{N_x} \sum_{y=1}^{N_y} \sum_{z=1}^{N_z} (\hat{\mathbf{Y}}_{b,c,x,y,z,t} - \mathbf{Y}_{b,c,x,y,z,t})^2}$$

$$\text{Mean}_{b,c,t} = \frac{1}{N_x \cdot N_y \cdot N_z} \sum_{x=1}^{N_x} \sum_{y=1}^{N_y} \sum_{z=1}^{N_z} \mathbf{Y}_{b,c,x,y,z,t}$$

$$\text{Var}_{b,c,t} = \frac{1}{N_x \cdot N_y \cdot N_z} \sum_{x=1}^{N_x} \sum_{y=1}^{N_y} \sum_{z=1}^{N_z} (\mathbf{Y}_{b,c,x,y,z,t} - \text{Mean}_{b,c,t})^2$$

$$\text{vRMSE} = \frac{1}{N_b \cdot N_c \cdot N_t} \sum_{b=1}^{N_b} \sum_{c=1}^{N_c} \sum_{t=1}^{N_t} \left( \frac{\text{RMSE}_{b,c,t}}{\sqrt{\text{Var}_{b,c,t} + \epsilon}} \right)$$

$$\text{vRMSE} = \frac{1}{N_b \cdot N_c \cdot N_t} \sum_{b=1}^{N_b} \sum_{c=1}^{N_c} \sum_{t=1}^{N_t} \left( \sqrt{\frac{\text{MSE}_{b,c,t}}{\text{Var}_{b,c,t} + \epsilon}} \right)$$

where the mean and variance are computed over the spatial dimensions in $\text{Mean}_{b,c,t}$ and $\text{Var}_{b,c,t}$, respectively. In other words, the mean and variance are computed for each sample, timestep, and channel independently. The vRMSE score is finally computed by taking a mean over the batch, channel, and temporal dimensions, yielding a single scalar value.

### G.1.6 Mean Energy Log Ratio (MELR) and Weighted Log Ratio (WLR)

Let $K$ be the set of wavenumbers (frequency bins) being evaluated, $|K|$ the cardinality of $K$, and $E_{\text{pred}}(k)$ and $E_{\text{ref}}(k)$ the energy of the predicted and reference (i.e., ground truth) spectrum at wavenumber $k$.

Wan et al. (2023) defines the MELR metric as,

$$\text{MELR} = \sum_{k \in K} w_k \left| \log \left( \frac{E_{\text{pred}}(k)}{E_{\text{ref}}(k)} \right) \right|, \tag{15}$$

with the energy spectrum being defined as,

$$E(k) = \sum_{|\underline{k}|=k} |\hat{u}(\underline{k})|^2 = \sum_{|\underline{k}|=k} \left| \sum_i u(x_i) \exp(-j2\pi \underline{k} \cdot x_i/L) \right|^2 \tag{16}$$

where $L$ is the length of the physical domain, $u$ is the snapshot of the system state, $\hat{u}(\underline{k})$ is the Fourier coefficient at wave-vector $\underline{k}$, $k$ is the magnitude of the wave-vector $\underline{k}$ (i.e., for 2D spatial data, $k = |\underline{k}| = \sqrt{k_x^2 + k_y^2}$).

Treating each wavenumber's log ratio equally and assuming $N_b$ samples in the test set, we arrive at the formula for the unweighted MELR or simply MELR,

$$\text{MELR} = \frac{1}{N_b} \sum_{b=1}^{N_b} \left( \frac{1}{|K|} \sum_{k \in K} \left| \log \left( \frac{E_{\text{pred}}^{(b)}(k)}{E_{\text{ref}}^{(b)}(k)} \right) \right| \right) \tag{17}$$

WLR is then the weighted version of MELR with the weights $w_k$ computed using the reference energy spectra,

$$\text{WLR} = \frac{1}{N_b} \sum_{b=1}^{N_b} \left( \sum_{k \in K} \left( \frac{E_{\text{ref}}^{(b)}(k)}{\sum_{j \in K} E_{\text{ref}}^{(b)}(j)} \right) \left| \log \left( \frac{E_{\text{pred}}^{(b)}(k)}{E_{\text{ref}}^{(b)}(k)} \right) \right| \right) \tag{18}$$

### G.1.7 Relative Percentage Difference (Rel. % Diff)

We report REL. % DIFF, which indicates the relative improvement (negative value) or degradation (positive value) of errors over the baseline. Denote the proposed model's score of a chosen evaluation metric (e.g., fRMSE) as $\hat{p}$ and the baseline model's score as $\hat{b}$. Then, the REL. % DIFF is defined as,

$$\text{REL. \% DIFF} := \frac{(\hat{p} - \hat{b})}{\hat{b}} \times 100 \tag{19}$$

## G.2 Detailed Quantitative Results with Full Set of Metrics

### G.2.1 1-step Training and Evaluation

To expand on the abridged results presented in the main paper, we present the results with the full set of evaluation metrics for the setting of 1-step training of baselines and our proposed LOGLO-FNO model on four 2D PDE problems and one 3D PDE in the sections that follow.

**Kolmogorov Flow 2D.** Supplementing the results in the main paper, Table 8 shows the full set of evaluation metrics for 1-step and 5-step AR rollouts on Kolmogorov Flow test dataset.

Table 8: 1-step and 5-step AR evaluation of LOGLO-FNO compared with SOTA baselines on the test set of 2D Kolmogorov Flow (Li et al., 2022b). Rel. % Diff indicates improvement (-) or degradation (+) with respect to FNO. LOGLO-FNO uses 40 and (16, 9) modes in the global and local branches, respectively, whereas the width is set as 65. NO-LIDK* denotes using only localized integral kernel, NO-LIDK$^\diamond$ means only differential kernel, and NO-LIDK$^\dagger$ means employing both. Transolver$^\star$ indicates a longer training time of the model for 500 epochs due to convergence issues at shorter training epochs of 136.

| Model | RMSE ($\downarrow$) | nRMSE | bRMSE | cRMSE | fRMSE(L) | fRMSE(M) | fRMSE(H) | MaxError ($\downarrow$) | MELR ($\downarrow$) | WLR ($\downarrow$) |
|---|---|---|---|---|---|---|---|---|---|---|
| **1-step Evaluation** | | | | | | | | | | |
| U-Net | $7.17 \cdot 10^{-1}$ | $1.3 \cdot 10^{-1}$ | $1.47 \cdot 10^0$ | $1.74 \cdot 10^{-2}$ | $2.24 \cdot 10^{-2}$ | $3.57 \cdot 10^{-2}$ | $4.39 \cdot 10^{-2}$ | $2.01 \cdot 10^1$ | $1.64 \cdot 10^{-1}$ | $1.48 \cdot 10^{-2}$ |
| Transolver$^\star$ | $7.69 \cdot 10^{-1}$ | $1.39 \cdot 10^{-1}$ | $1.42 \cdot 10^0$ | $7.65 \cdot 10^{-3}$ | $2.74 \cdot 10^{-2}$ | $4.48 \cdot 10^{-2}$ | $4.65 \cdot 10^{-2}$ | $1.52 \cdot 10^1$ | $1.43 \cdot 10^{-1}$ | $1.83 \cdot 10^{-2}$ |
| FNO | $8.08 \cdot 10^{-1}$ | $1.47 \cdot 10^{-1}$ | $7.94 \cdot 10^{-1}$ | $1.1 \cdot 10^{-2}$ | $1.36 \cdot 10^{-2}$ | $2.05 \cdot 10^{-2}$ | $4.7 \cdot 10^{-2}$ | $1.46 \cdot 10^1$ | $5.2 \cdot 10^{-1}$ | $2.83 \cdot 10^{-2}$ |
| F-FNO | $7.53 \cdot 10^{-1}$ | $1.37 \cdot 10^{-1}$ | $7.41 \cdot 10^{-1}$ | $1.5 \cdot 10^{-2}$ | $1.49 \cdot 10^{-2}$ | $2.15 \cdot 10^{-2}$ | $4.36 \cdot 10^{-2}$ | $1.42 \cdot 10^1$ | $4.74 \cdot 10^{-1}$ | $2.28 \cdot 10^{-2}$ |
| LSM | $7.49 \cdot 10^{-1}$ | $1.36 \cdot 10^{-1}$ | $1.47 \cdot 10^0$ | $1.36 \cdot 10^{-2}$ | $2.59 \cdot 10^{-2}$ | $4.42 \cdot 10^{-2}$ | $4.64 \cdot 10^{-2}$ | $2.05 \cdot 10^1$ | $1.43 \cdot 10^{-1}$ | $1.68 \cdot 10^{-2}$ |
| U-FNO | $6.13 \cdot 10^{-1}$ | $1.12 \cdot 10^{-1}$ | $1.0 \cdot 10^0$ | $\mathbf{7.09} \cdot 10^{-3}$ | $1.27 \cdot 10^{-2}$ | $2.22 \cdot 10^{-2}$ | $3.71 \cdot 10^{-2}$ | $1.64 \cdot 10^1$ | $1.38 \cdot 10^{-1}$ | $1.09 \cdot 10^{-2}$ |
| NO-LIDK* | $7.25 \cdot 10^{-1}$ | $1.33 \cdot 10^{-1}$ | $1.12 \cdot 10^0$ | $9.05 \cdot 10^{-3}$ | $1.45 \cdot 10^{-2}$ | $2.69 \cdot 10^{-2}$ | $4.55 \cdot 10^{-2}$ | $1.65 \cdot 10^1$ | $1.85 \cdot 10^{-1}$ | $1.54 \cdot 10^{-2}$ |
| NO-LIDK$^\diamond$ | $6.13 \cdot 10^{-1}$ | $1.11 \cdot 10^{-1}$ | $5.95 \cdot 10^{-1}$ | $1.46 \cdot 10^{-2}$ | $1.65 \cdot 10^{-2}$ | $2.29 \cdot 10^{-2}$ | $3.91 \cdot 10^{-2}$ | $1.5 \cdot 10^1$ | $9.57 \cdot 10^{-2}$ | $1.1 \cdot 10^{-2}$ |
| NO-LIDK$^\dagger$ | $5.86 \cdot 10^{-1}$ | $1.07 \cdot 10^{-1}$ | $\mathbf{5.64} \cdot 10^{-1}$ | $1.05 \cdot 10^{-2}$ | $1.44 \cdot 10^{-2}$ | $2.46 \cdot 10^{-2}$ | $3.82 \cdot 10^{-2}$ | $1.47 \cdot 10^1$ | $\mathbf{7.11} \cdot 10^{-2}$ | $\mathbf{1.0} \cdot 10^{-2}$ |
| **LOGLO-FNO** | $\underline{5.89} \cdot 10^{-1}$ | $\mathbf{1.07} \cdot 10^{-1}$ | $\underline{6.74} \cdot 10^{-1}$ | $\underline{7.23} \cdot 10^{-3}$ | $\mathbf{1.21} \cdot 10^{-2}$ | $\mathbf{1.81} \cdot 10^{-2}$ | $\mathbf{3.54} \cdot 10^{-2}$ | $\mathbf{1.33} \cdot 10^1$ | $\underline{1.29} \cdot 10^{-1}$ | $\underline{1.06} \cdot 10^{-2}$ |
| Rel. % Diff | -27.1 % | -27.21 % | -15.12 % | -34.31 % | -11.22 % | -11.88 % | -24.64 % | -8.99 % | -75.12 % | -62.63 % |
| **5-step Autoregressive Evaluation** | | | | | | | | | | |
| U-Net | $1.51 \cdot 10^0$ | $2.65 \cdot 10^{-1}$ | $2.31 \cdot 10^0$ | $5.39 \cdot 10^{-2}$ | $6.53 \cdot 10^{-2}$ | $1.04 \cdot 10^{-1}$ | $9.38 \cdot 10^{-2}$ | $1.81 \cdot 10^1$ | $2.13 \cdot 10^{-1}$ | $3.52 \cdot 10^{-2}$ |
| Transolver$^\star$ | $1.91 \cdot 10^0$ | $3.36 \cdot 10^{-1}$ | $2.59 \cdot 10^0$ | $\mathbf{6.12} \cdot 10^{-3}$ | $7.44 \cdot 10^{-2}$ | $1.51 \cdot 10^{-1}$ | $1.18 \cdot 10^{-1}$ | $2.36 \cdot 10^1$ | $2.45 \cdot 10^{-1}$ | $4.92 \cdot 10^{-2}$ |
| FNO | $1.33 \cdot 10^0$ | $2.35 \cdot 10^{-1}$ | $1.34 \cdot 10^0$ | $1.46 \cdot 10^{-2}$ | $3.37 \cdot 10^{-2}$ | $5.80 \cdot 10^{-2}$ | $8.37 \cdot 10^{-2}$ | $1.60 \cdot 10^1$ | $6.18 \cdot 10^{-1}$ | $4.93 \cdot 10^{-2}$ |
| F-FNO | $1.29 \cdot 10^0$ | $2.28 \cdot 10^{-1}$ | $1.27 \cdot 10^0$ | $2.28 \cdot 10^{-2}$ | $3.60 \cdot 10^{-2}$ | $5.52 \cdot 10^{-2}$ | $8.12 \cdot 10^{-2}$ | $1.50 \cdot 10^1$ | $5.37 \cdot 10^{-1}$ | $3.99 \cdot 10^{-2}$ |
| LSM | $1.81 \cdot 10^0$ | $3.18 \cdot 10^{-1}$ | $2.76 \cdot 10^0$ | $3.87 \cdot 10^{-2}$ | $7.6 \cdot 10^{-2}$ | $1.44 \cdot 10^{-1}$ | $1.12 \cdot 10^{-1}$ | $2.08 \cdot 10^1$ | $2.04 \cdot 10^{-1}$ | $4.39 \cdot 10^{-2}$ |
| U-FNO | $1.15 \cdot 10^0$ | $2.03 \cdot 10^{-1}$ | $1.45 \cdot 10^0$ | $6.3 \cdot 10^{-3}$ | $\mathbf{2.69} \cdot 10^{-2}$ | $5.33 \cdot 10^{-2}$ | $7.35 \cdot 10^{-2}$ | $1.56 \cdot 10^1$ | $2.01 \cdot 10^{-1}$ | $2.51 \cdot 10^{-2}$ |
| NO-LIDK* | $1.36 \cdot 10^0$ | $2.39 \cdot 10^{-1}$ | $1.8 \cdot 10^0$ | $1.1 \cdot 10^{-2}$ | $3.14 \cdot 10^{-2}$ | $6.86 \cdot 10^{-2}$ | $8.91 \cdot 10^{-2}$ | $1.59 \cdot 10^1$ | $2.23 \cdot 10^{-1}$ | $2.93 \cdot 10^{-2}$ |
| NO-LIDK$^\diamond$ | $1.17 \cdot 10^0$ | $2.04 \cdot 10^{-1}$ | $1.12 \cdot 10^0$ | $1.92 \cdot 10^{-2}$ | $3.71 \cdot 10^{-2}$ | $5.96 \cdot 10^{-2}$ | $7.6 \cdot 10^{-2}$ | $1.61 \cdot 10^1$ | $1.42 \cdot 10^{-1}$ | $2.12 \cdot 10^{-2}$ |
| NO-LIDK$^\dagger$ | $1.17 \cdot 10^0$ | $2.05 \cdot 10^{-1}$ | $1.14 \cdot 10^0$ | $1.23 \cdot 10^{-2}$ | $3.31 \cdot 10^{-2}$ | $5.94 \cdot 10^{-2}$ | $7.74 \cdot 10^{-2}$ | $1.56 \cdot 10^1$ | $\mathbf{1.03} \cdot 10^{-1}$ | $2.18 \cdot 10^{-2}$ |
| **LOGLO-FNO** | $\mathbf{1.09} \cdot 10^0$ | $\mathbf{1.92} \cdot 10^{-1}$ | $\mathbf{1.12} \cdot 10^0$ | $8.99 \cdot 10^{-3}$ | $\underline{2.80} \cdot 10^{-2}$ | $\mathbf{4.55} \cdot 10^{-2}$ | $\mathbf{6.93} \cdot 10^{-2}$ | $\mathbf{1.26} \cdot 10^1$ | $\underline{1.67} \cdot 10^{-1}$ | $\mathbf{2.07} \cdot 10^{-2}$ |
| Rel. % Diff | -18.26 % | -18.39 % | -16.12 % | -38.21 % | -16.86 % | -21.62 % | -17.26 % | -21.31 % | -73.04 % | -58.02 % |

**CNS Turb 2D.** Table 9 presents the full set of evaluation metrics for 1-step and 5-step autoregressive rollouts on the test split of CNS Turb 2D dataset.

Table 9: 1-step and 5-step AR evaluation of LOGLO-FNO compared with SOTA baselines on the test set of CNS Turb 2D (Takamoto et al., 2022). REL. % DIFF indicates improvement (-) or degradation (+) with respect to FNO. LOGLO-FNO uses 40 and (16, 9) modes in the global and local branches, respectively, whereas the width is set as 65. NO-LIDK* denotes using only localized integral kernel, NO-LIDK$^\diamond$ means only differential kernel, and NO-LIDK$^\dagger$ means employing both.

| Model | RMSE ($\downarrow$) | nRMSE | bRMSE | cRMSE | fRMSE(L) | fRMSE(M) | fRMSE(H) | MaxError ($\downarrow$) | MELR ($\downarrow$) | WLR ($\downarrow$) |
|---|---|---|---|---|---|---|---|---|---|---|
| | | | | | **1-step Evaluation** | | | | | |
| U-Net | $3.73 \cdot 10^{-2}$ | $4.9 \cdot 10^{-2}$ | $9.27 \cdot 10^{-2}$ | $3.06 \cdot 10^{-3}$ | $2.47 \cdot 10^{-3}$ | $2.9 \cdot 10^{-3}$ | $1.48 \cdot 10^{-3}$ | $2.1 \cdot 10^{0}$ | $1.37 \cdot 10^{-1}$ | $7.39 \cdot 10^{-3}$ |
| Transolver$^\star$ | $3.13 \cdot 10^{-1}$ | $4.22 \cdot 10^{-1}$ | $3.08 \cdot 10^{-1}$ | $1.1 \cdot 10^{-2}$ | $4.77 \cdot 10^{-2}$ | $4.11 \cdot 10^{-2}$ | $4.47 \cdot 10^{-3}$ | $3.66 \cdot 10^{0}$ | $5.01 \cdot 10^{-1}$ | $3.33 \cdot 10^{-1}$ |
| FNO | $3.69 \cdot 10^{-2}$ | $4.82 \cdot 10^{-2}$ | $3.64 \cdot 10^{-2}$ | $1.42 \cdot 10^{-3}$ | $1.28 \cdot 10^{-3}$ | $1.78 \cdot 10^{-3}$ | $1.57 \cdot 10^{-3}$ | $1.96 \cdot 10^{0}$ | $2.78 \cdot 10^{-1}$ | $7.8 \cdot 10^{-3}$ |
| LSM | $6.11 \cdot 10^{-2}$ | $8.08 \cdot 10^{-2}$ | $1.67 \cdot 10^{-1}$ | $3.83 \cdot 10^{-3}$ | $6.17 \cdot 10^{-3}$ | $6.55 \cdot 10^{-3}$ | $2.01 \cdot 10^{-3}$ | $2.43 \cdot 10^{0}$ | $2.7 \cdot 10^{-1}$ | $1.66 \cdot 10^{-2}$ |
| U-FNO | $3.43 \cdot 10^{-2}$ | $4.49 \cdot 10^{-2}$ | $4.92 \cdot 10^{-2}$ | $3.42 \cdot 10^{-3}$ | $2.24 \cdot 10^{-3}$ | $2.01 \cdot 10^{-3}$ | $1.43 \cdot 10^{-3}$ | $2.0 \cdot 10^{0}$ | $\mathbf{1.32} \cdot 10^{-1}$ | $9.5 \cdot 10^{-3}$ |
| NO-LIDK* | $4.81 \cdot 10^{-2}$ | $6.4 \cdot 10^{-2}$ | $4.77 \cdot 10^{-2}$ | $1.72 \cdot 10^{-3}$ | $1.5 \cdot 10^{-3}$ | $2.14 \cdot 10^{-3}$ | $1.98 \cdot 10^{-3}$ | $2.35 \cdot 10^{0}$ | $2.98 \cdot 10^{-1}$ | $1.31 \cdot 10^{-2}$ |
| NO-LIDK$^\diamond$ | $4.18 \cdot 10^{-2}$ | $5.54 \cdot 10^{-2}$ | $4.13 \cdot 10^{-2}$ | $1.46 \cdot 10^{-3}$ | $1.35 \cdot 10^{-3}$ | $1.88 \cdot 10^{-3}$ | $1.76 \cdot 10^{-3}$ | $2.28 \cdot 10^{0}$ | $2.0 \cdot 10^{-1}$ | $9.49 \cdot 10^{-3}$ |
| NO-LIDK$^\dagger$ | $4.2 \cdot 10^{-2}$ | $5.57 \cdot 10^{-2}$ | $4.16 \cdot 10^{-2}$ | $1.56 \cdot 10^{-3}$ | $1.37 \cdot 10^{-3}$ | $1.89 \cdot 10^{-3}$ | $1.77 \cdot 10^{-3}$ | $2.26 \cdot 10^{0}$ | $2.08 \cdot 10^{-1}$ | $9.54 \cdot 10^{-3}$ |
| **LOGLO-FNO** | $\mathbf{3.23} \cdot 10^{-2}$ | $\mathbf{4.24} \cdot 10^{-2}$ | $\mathbf{3.34} \cdot 10^{-2}$ | $\mathbf{1.24} \cdot 10^{-3}$ | $\mathbf{9.01} \cdot 10^{-4}$ | $\mathbf{1.1} \cdot 10^{-3}$ | $\mathbf{1.37} \cdot 10^{-3}$ | $\mathbf{1.86} \cdot 10^{0}$ | $1.71 \cdot 10^{-1}$ | $\mathbf{5.5} \cdot 10^{-3}$ |
| REL. % DIFF | -12.31% | -11.91 % | -8.3% | -12.54 % | -29.57% | -38.1 % | -12.38 % | -4.79 % | -38.39% | -29.53% |
| | | | | | **5-step Autoregressive Evaluation** | | | | | |
| U-Net | $1.15 \cdot 10^{-1}$ | $1.44 \cdot 10^{-1}$ | $1.98 \cdot 10^{-1}$ | $7.13 \cdot 10^{-3}$ | $1.42 \cdot 10^{-2}$ | $1.3 \cdot 10^{-2}$ | $3.08 \cdot 10^{-3}$ | $2.87 \cdot 10^{0}$ | $1.37 \cdot 10^{-1}$ | $2.13 \cdot 10^{-2}$ |
| Transolver$^\star$ | $5.29 \cdot 10^{-1}$ | $6.69 \cdot 10^{-1}$ | $5.3 \cdot 10^{-1}$ | $2.7 \cdot 10^{-2}$ | $9.21 \cdot 10^{-2}$ | $6.0 \cdot 10^{-2}$ | $4.7 \cdot 10^{-3}$ | $4.23 \cdot 10^{0}$ | $3.54 \cdot 10^{0}$ | $1.25 \cdot 10^{0}$ |
| FNO | $6.34 \cdot 10^{-2}$ | $7.78 \cdot 10^{-2}$ | $6.21 \cdot 10^{-2}$ | $2.45 \cdot 10^{-3}$ | $3.76 \cdot 10^{-3}$ | $5.37 \cdot 10^{-3}$ | $2.4 \cdot 10^{-3}$ | $2.45 \cdot 10^{0}$ | $2.22 \cdot 10^{-1}$ | $9.36 \cdot 10^{-3}$ |
| LSM | $2.12 \cdot 10^{-1}$ | $2.67 \cdot 10^{-1}$ | $3.74 \cdot 10^{-1}$ | $1.91 \cdot 10^{-2}$ | $3.4 \cdot 10^{-2}$ | $2.5 \cdot 10^{-2}$ | $4.16 \cdot 10^{-3}$ | $3.16 \cdot 10^{0}$ | $2.68 \cdot 10^{-1}$ | $5.34 \cdot 10^{-2}$ |
| U-FNO | $6.84 \cdot 10^{-2}$ | $8.42 \cdot 10^{-2}$ | $7.91 \cdot 10^{-2}$ | $5.46 \cdot 10^{-3}$ | $5.38 \cdot 10^{-3}$ | $5.9 \cdot 10^{-3}$ | $2.54 \cdot 10^{-3}$ | $2.57 \cdot 10^{0}$ | $\mathbf{1.13} \cdot 10^{-1}$ | $1.49 \cdot 10^{-2}$ |
| NO-LIDK* | $6.95 \cdot 10^{-2}$ | $8.55 \cdot 10^{-2}$ | $6.86 \cdot 10^{-2}$ | $2.83 \cdot 10^{-3}$ | $4.14 \cdot 10^{-3}$ | $5.92 \cdot 10^{-3}$ | $2.62 \cdot 10^{-3}$ | $2.83 \cdot 10^{0}$ | $2.06 \cdot 10^{-1}$ | $1.15 \cdot 10^{-2}$ |
| NO-LIDK$^\diamond$ | $6.41 \cdot 10^{-2}$ | $7.88 \cdot 10^{-2}$ | $6.32 \cdot 10^{-2}$ | $2.64 \cdot 10^{-3}$ | $3.82 \cdot 10^{-3}$ | $5.35 \cdot 10^{-3}$ | $2.44 \cdot 10^{-3}$ | $2.49 \cdot 10^{0}$ | $1.37 \cdot 10^{-1}$ | $9.57 \cdot 10^{-3}$ |
| NO-LIDK$^\dagger$ | $6.44 \cdot 10^{-2}$ | $7.91 \cdot 10^{-2}$ | $6.35 \cdot 10^{-2}$ | $2.63 \cdot 10^{-3}$ | $3.82 \cdot 10^{-3}$ | $5.38 \cdot 10^{-3}$ | $2.47 \cdot 10^{-3}$ | $2.57 \cdot 10^{0}$ | $1.45 \cdot 10^{-1}$ | $1.01 \cdot 10^{-2}$ |
| **LOGLO-FNO** | $\mathbf{5.43} \cdot 10^{-2}$ | $\mathbf{6.67} \cdot 10^{-2}$ | $\mathbf{5.48} \cdot 10^{-2}$ | $\mathbf{2.1} \cdot 10^{-3}$ | $\mathbf{2.76} \cdot 10^{-3}$ | $\mathbf{3.97} \cdot 10^{-3}$ | $\mathbf{2.12} \cdot 10^{-3}$ | $\mathbf{2.25} \cdot 10^{0}$ | $\underline{1.29} \cdot 10^{-1}$ | $\mathbf{6.67} \cdot 10^{-3}$ |
| REL. % DIFF | -14.35% | -14.31 % | -11.8% | -14.23 % | -26.7 % | -26.09 % | -11.42% | -8.09% | -41.8% | -28.75% |

**Wave-Gauss 2D.** In Table 10, we present results with the full set of evaluation metrics on the test set of Wave-Gauss 2D dataset.

Table 10: 1-step, 5-step, and 15-step AR evaluation of LOGLO-FNO compared against SOTA baselines on the test set of Wave-Gauss 2D (Herde et al., 2024). REL. % DIFF indicates improvement (-) or degradation (+) with respect to FNO. LOGLO-FNO uses 20 and (16, 9) modes in the global and local branches, respectively, whereas the width is set to 32. NO-LIDK* denotes using only localized integral kernel, NO-LIDK$^\diamond$ means only differential kernel, and NO-LIDK$^\dagger$ means employing both.

| Model | RMSE ($\downarrow$) | nRMSE | bRMSE | cRMSE | fRMSE(L) | fRMSE(M) | fRMSE(H) | MaxError ($\downarrow$) | MELR ($\downarrow$) | WLR ($\downarrow$) |
|---|---|---|---|---|---|---|---|---|---|---|
| | | | | **1-step Evaluation** | | | | | | |
| Transolver* | $2.01 \cdot 10^{-1}$ | $2.29 \cdot 10^{-1}$ | $7.29 \cdot 10^{-2}$ | $1.97 \cdot 10^{-2}$ | $3.51 \cdot 10^{-2}$ | $2.76 \cdot 10^{-2}$ | $2.2 \cdot 10^{-3}$ | $2.94 \cdot 10^{0}$ | $1.29 \cdot 10^{0}$ | $1.1 \cdot 10^{-1}$ |
| FNO | $3.76 \cdot 10^{-2}$ | $4.03 \cdot 10^{-2}$ | $1.91 \cdot 10^{-2}$ | $2.82 \cdot 10^{-3}$ | $7.5 \cdot 10^{-3}$ | $5.45 \cdot 10^{-3}$ | $6.22 \cdot 10^{-4}$ | $1.31 \cdot 10^{0}$ | $9.66 \cdot 10^{-1}$ | $9.01 \cdot 10^{-3}$ |
| LSM | $\mathbf{2.83} \cdot 10^{-2}$ | $\mathbf{3.08} \cdot 10^{-2}$ | $3.04 \cdot 10^{-2}$ | $2.69 \cdot 10^{-3}$ | $\mathbf{4.91} \cdot 10^{-3}$ | $3.86 \cdot 10^{-3}$ | $8.54 \cdot 10^{-4}$ | $\mathbf{9.99} \cdot 10^{-1}$ | $1.2 \cdot 10^{0}$ | $\mathbf{6.2} \cdot 10^{-3}$ |
| U-FNO | $3.23 \cdot 10^{-2}$ | $3.49 \cdot 10^{-2}$ | $2.09 \cdot 10^{-2}$ | $3.89 \cdot 10^{-3}$ | $6.62 \cdot 10^{-3}$ | $4.52 \cdot 10^{-3}$ | $7.24 \cdot 10^{-4}$ | $1.2 \cdot 10^{0}$ | $1.41 \cdot 10^{0}$ | $6.82 \cdot 10^{-3}$ |
| NO-LIDK* | $3.62 \cdot 10^{-2}$ | $3.93 \cdot 10^{-2}$ | $1.91 \cdot 10^{-2}$ | $2.94 \cdot 10^{-3}$ | $6.95 \cdot 10^{-3}$ | $5.1 \cdot 10^{-3}$ | $8.22 \cdot 10^{-4}$ | $1.2 \cdot 10^{0}$ | $1.27 \cdot 10^{0}$ | $8.28 \cdot 10^{-3}$ |
| NO-LIDK$^\diamond$ | $3.33 \cdot 10^{-2}$ | $3.58 \cdot 10^{-2}$ | $1.68 \cdot 10^{-2}$ | $\mathbf{2.57} \cdot 10^{-3}$ | $6.49 \cdot 10^{-3}$ | $4.78 \cdot 10^{-3}$ | $7.56 \cdot 10^{-4}$ | $1.3 \cdot 10^{0}$ | $1.31 \cdot 10^{0}$ | $7.32 \cdot 10^{-3}$ |
| NO-LIDK$^\dagger$ | $3.25 \cdot 10^{-2}$ | $3.5 \cdot 10^{-2}$ | $1.68 \cdot 10^{-2}$ | $2.7 \cdot 10^{-3}$ | $6.31 \cdot 10^{-3}$ | $4.61 \cdot 10^{-3}$ | $7.4 \cdot 10^{-4}$ | $1.19 \cdot 10^{0}$ | $1.28 \cdot 10^{0}$ | $7.04 \cdot 10^{-3}$ |
| **LOGLO-FNO** | $\underline{2.9} \cdot 10^{-2}$ | $\underline{3.11} \cdot 10^{-2}$ | $\mathbf{1.17} \cdot 10^{-2}$ | $2.75 \cdot 10^{-3}$ | $\underline{6.43} \cdot 10^{-3}$ | $\mathbf{3.84} \cdot 10^{-3}$ | $\mathbf{3.44} \cdot 10^{-4}$ | $1.12 \cdot 10^{0}$ | $\mathbf{9.29} \cdot 10^{-1}$ | $6.82 \cdot 10^{-3}$ |
| REL. % DIFF | -22.97% | -22.79 % | -38.59% | -2.55% | -14.27% | -29.54% | -44.65% | -14.79% | -3.77% | -24.33% |
| | | | | **5-step Autoregressive Evaluation** | | | | | | |
| Transolver* | $3.45 \cdot 10^{-1}$ | $3.62 \cdot 10^{-1}$ | $1.31 \cdot 10^{-1}$ | $6.09 \cdot 10^{-2}$ | $7.73 \cdot 10^{-2}$ | $3.47 \cdot 10^{-2}$ | $2.93 \cdot 10^{-3}$ | $3.66 \cdot 10^{0}$ | $2.07 \cdot 10^{0}$ | $1.79 \cdot 10^{-1}$ |
| FNO | $6.91 \cdot 10^{-2}$ | $6.75 \cdot 10^{-2}$ | $2.06 \cdot 10^{-2}$ | $6.5 \cdot 10^{-3}$ | $1.6 \cdot 10^{-2}$ | $8.35 \cdot 10^{-3}$ | $8.57 \cdot 10^{-4}$ | $2.23 \cdot 10^{0}$ | $1.5 \cdot 10^{0}$ | $1.93 \cdot 10^{-2}$ |
| LSM | $\mathbf{4.33} \cdot 10^{-2}$ | $\mathbf{4.34} \cdot 10^{-2}$ | $3.63 \cdot 10^{-2}$ | $\mathbf{5.48} \cdot 10^{-3}$ | $\mathbf{8.64} \cdot 10^{-3}$ | $5.68 \cdot 10^{-3}$ | $1.1 \cdot 10^{-3}$ | $2.12 \cdot 10^{0}$ | $1.72 \cdot 10^{0}$ | $\mathbf{1.24} \cdot 10^{-2}$ |
| U-FNO | $6.37 \cdot 10^{-2}$ | $6.26 \cdot 10^{-2}$ | $2.55 \cdot 10^{-2}$ | $1.44 \cdot 10^{-2}$ | $1.6 \cdot 10^{-2}$ | $7.64 \cdot 10^{-3}$ | $9.27 \cdot 10^{-4}$ | $2.54 \cdot 10^{0}$ | $2.0 \cdot 10^{0}$ | $1.82 \cdot 10^{-2}$ |
| NO-LIDK* | $6.05 \cdot 10^{-2}$ | $5.96 \cdot 10^{-2}$ | $1.91 \cdot 10^{-2}$ | $7.76 \cdot 10^{-3}$ | $1.34 \cdot 10^{-2}$ | $7.92 \cdot 10^{-3}$ | $1.19 \cdot 10^{-3}$ | $2.65 \cdot 10^{0}$ | $1.99 \cdot 10^{0}$ | $1.81 \cdot 10^{-2}$ |
| NO-LIDK$^\diamond$ | $1.17 \cdot 10^{-1}$ | $1.21 \cdot 10^{-1}$ | $2.23 \cdot 10^{-2}$ | $1.14 \cdot 10^{-2}$ | $2.43 \cdot 10^{-2}$ | $1.12 \cdot 10^{-2}$ | $4.07 \cdot 10^{-3}$ | $2.96 \cdot 10^{0}$ | $2.42 \cdot 10^{0}$ | $4.02 \cdot 10^{-2}$ |
| NO-LIDK$^\dagger$ | $1.02 \cdot 10^{-1}$ | $1.06 \cdot 10^{-1}$ | $2.16 \cdot 10^{-2}$ | $1.38 \cdot 10^{-2}$ | $2.1 \cdot 10^{-2}$ | $1.02 \cdot 10^{-2}$ | $3.85 \cdot 10^{-3}$ | $2.95 \cdot 10^{0}$ | $2.41 \cdot 10^{0}$ | $3.02 \cdot 10^{-2}$ |
| **LOGLO-FNO** | $\underline{5.1} \cdot 10^{-2}$ | $\underline{4.81} \cdot 10^{-2}$ | $\mathbf{1.12} \cdot 10^{-2}$ | $\underline{6.35} \cdot 10^{-3}$ | $\underline{1.29} \cdot 10^{-2}$ | $\mathbf{5.43} \cdot 10^{-3}$ | $\mathbf{4.57} \cdot 10^{-4}$ | $\mathbf{1.62} \cdot 10^{0}$ | $\mathbf{1.48} \cdot 10^{0}$ | $\underline{1.39} \cdot 10^{-2}$ |
| REL. % DIFF | -26.2% | -28.76% | -45.43% | -2.41% | -19.52% | -34.91% | -46.6% | -27.16% | -1.74% | -27.85% |
| | | | | **15-step Autoregressive Evaluation** | | | | | | |
| Transolver* | $4.33 \cdot 10^{-1}$ | $4.96 \cdot 10^{-1}$ | $2.11 \cdot 10^{-1}$ | $1.2 \cdot 10^{-1}$ | $1.15 \cdot 10^{-1}$ | $3.64 \cdot 10^{-2}$ | $3.5 \cdot 10^{-3}$ | $3.84 \cdot 10^{0}$ | $1.73 \cdot 10^{0}$ | $2.36 \cdot 10^{-1}$ |
| FNO | $1.21 \cdot 10^{-1}$ | $1.34 \cdot 10^{-1}$ | $3.86 \cdot 10^{-2}$ | $1.76 \cdot 10^{-2}$ | $3.13 \cdot 10^{-2}$ | $1.32 \cdot 10^{-2}$ | $1.39 \cdot 10^{-3}$ | $2.42 \cdot 10^{0}$ | $1.16 \cdot 10^{0}$ | $3.53 \cdot 10^{-2}$ |
| LSM | $\mathbf{7.66} \cdot 10^{-2}$ | $\mathbf{8.49} \cdot 10^{-2}$ | $5.72 \cdot 10^{-2}$ | $\mathbf{1.29} \cdot 10^{-2}$ | $\mathbf{1.79} \cdot 10^{-2}$ | $9.33 \cdot 10^{-3}$ | $1.61 \cdot 10^{-3}$ | $1.98 \cdot 10^{0}$ | $1.29 \cdot 10^{0}$ | $2.63 \cdot 10^{-2}$ |
| U-FNO | $1.15 \cdot 10^{-1}$ | $1.27 \cdot 10^{-1}$ | $4.4 \cdot 10^{-2}$ | $3.08 \cdot 10^{-2}$ | $3.14 \cdot 10^{-2}$ | $1.24 \cdot 10^{-2}$ | $1.17 \cdot 10^{-3}$ | $2.67 \cdot 10^{0}$ | $1.54 \cdot 10^{0}$ | $3.32 \cdot 10^{-2}$ |
| NO-LIDK* | $9.98 \cdot 10^{-2}$ | $1.13 \cdot 10^{-1}$ | $3.14 \cdot 10^{-2}$ | $1.81 \cdot 10^{-2}$ | $2.61 \cdot 10^{-2}$ | $1.21 \cdot 10^{-2}$ | $1.38 \cdot 10^{-3}$ | $3.09 \cdot 10^{0}$ | $1.57 \cdot 10^{0}$ | $3.22 \cdot 10^{-2}$ |
| NO-LIDK$^\diamond$ | $1.14 \cdot 10^{0}$ | $1.48 \cdot 10^{0}$ | $4.61 \cdot 10^{-1}$ | $4.2 \cdot 10^{-1}$ | $2.88 \cdot 10^{-1}$ | $5.88 \cdot 10^{-2}$ | $4.24 \cdot 10^{-2}$ | $1.65 \cdot 10^{1}$ | $3.2 \cdot 10^{0}$ | $3.0 \cdot 10^{-1}$ |
| NO-LIDK$^\dagger$ | $7.89 \cdot 10^{-1}$ | $1.03 \cdot 10^{0}$ | $3.1 \cdot 10^{-1}$ | $1.77 \cdot 10^{-1}$ | $1.75 \cdot 10^{-1}$ | $6.09 \cdot 10^{-2}$ | $2.93 \cdot 10^{-2}$ | $1.94 \cdot 10^{1}$ | $3.09 \cdot 10^{0}$ | $2.06 \cdot 10^{-1}$ |
| **LOGLO-FNO** | $\underline{9.19} \cdot 10^{-2}$ | $\underline{1.02} \cdot 10^{-1}$ | $\mathbf{2.25} \cdot 10^{-2}$ | $\underline{1.58} \cdot 10^{-2}$ | $\underline{2.61} \cdot 10^{-2}$ | $\mathbf{9.24} \cdot 10^{-3}$ | $\mathbf{7.38} \cdot 10^{-4}$ | $\mathbf{1.86} \cdot 10^{0}$ | $1.04 \cdot 10^{0}$ | $\mathbf{2.54} \cdot 10^{-2}$ |
| REL. % DIFF | -23.83% | -24.5% | -41.61% | -10.45% | -16.84% | -30.19% | -46.99% | -23.46% | -10.16% | -28.0% |

**Compressible Euler four-quadrant Riemann problem 2D.** In Table 11, we present the preliminary 1-step and varying timesteps of rollout evaluation results from the experiments we conducted on the highly challenging Compressible Euler four-quadrant Riemann problem 2D problem, which exhibits shock waves and discontinuities.

Table 11: 1-step and varying timesteps (5, 10, 15, 20) autoregressive rollout evaluations of LOGLO-FNO compared against SOTA baselines on the test set of Compressible Euler four-quadrant Riemann problem 2D (Herde et al., 2024). REL. % DIFF indicates improvement (-) or degradation (+) with respect to FNO. LOGLO-FNO uses 40 and (16, 9) modes in the global and local branches, respectively, whereas the width is set to 65. NO-LIDK* denotes using only the localized integral kernel with a radius cutoff value of 0.0078125, and NO-LIDK$^\diamond$ means only the differential kernel.

| Model | RMSE (↓) | nRMSE | bRMSE | cRMSE | fRMSE(L) | fRMSE(M) | fRMSE(H) | MaxError (↓) | MELR (↓) | WLR (↓) |
|---|---|---|---|---|---|---|---|---|---|---|
| | | | | **1-step Evaluation** | | | | | | |
| FNO | $2.02 \cdot 10^{-2}$ | $2.08 \cdot 10^{-2}$ | $2.38 \cdot 10^{-2}$ | $8.46 \cdot 10^{-4}$ | $9.42 \cdot 10^{-4}$ | $1.03 \cdot 10^{-3}$ | $1.41 \cdot 10^{-3}$ | $1.58 \cdot 10^{0}$ | $8.8 \cdot 10^{-2}$ | $3.09 \cdot 10^{-3}$ |
| NO-LIDK* | $3.56 \cdot 10^{-2}$ | $3.58 \cdot 10^{-2}$ | $3.94 \cdot 10^{-2}$ | $1.14 \cdot 10^{-3}$ | $1.5 \cdot 10^{-3}$ | $2.35 \cdot 10^{-3}$ | $2.47 \cdot 10^{-3}$ | $2.35 \cdot 10^{0}$ | $7.26 \cdot 10^{-2}$ | $5.67 \cdot 10^{-3}$ |
| NO-LIDK$^\diamond$ | $1.59 \cdot 10^{-2}$ | $1.66 \cdot 10^{-2}$ | $1.83 \cdot 10^{-2}$ | $1.47 \cdot 10^{-3}$ | $1.05 \cdot 10^{-3}$ | $8.54 \cdot 10^{-4}$ | $1.11 \cdot 10^{-3}$ | $1.4 \cdot 10^{0}$ | $4.34 \cdot 10^{-2}$ | $2.54 \cdot 10^{-3}$ |
| **LOGLO-FNO** | $\mathbf{1.48 \cdot 10^{-2}}$ | $\mathbf{1.5 \cdot 10^{-2}}$ | $\mathbf{1.87 \cdot 10^{-2}}$ | $\mathbf{6.36 \cdot 10^{-4}}$ | $\mathbf{5.79 \cdot 10^{-4}}$ | $\mathbf{5.09 \cdot 10^{-4}}$ | $\mathbf{9.67 \cdot 10^{-4}}$ | $\mathbf{1.31 \cdot 10^{0}}$ | $\mathbf{5.42 \cdot 10^{-2}}$ | $\mathbf{1.75 \cdot 10^{-3}}$ |
| REL. % DIFF | -26.65% | -27.9% | -21.29% | -24.82% | -38.56% | -50.71% | -31.58% | -16.9% | -38.45% | -43.26% |
| | | | | **5-step Autoregressive Evaluation** | | | | | | |
| FNO | $3.39 \cdot 10^{-2}$ | $3.43 \cdot 10^{-2}$ | $5.62 \cdot 10^{-2}$ | $2.69 \cdot 10^{-3}$ | $2.58 \cdot 10^{-3}$ | $2.7 \cdot 10^{-3}$ | $2.35 \cdot 10^{-3}$ | $2.49 \cdot 10^{0}$ | $1.18 \cdot 10^{-1}$ | $8.3 \cdot 10^{-3}$ |
| NO-LIDK* | $3.91 \cdot 10^{-2}$ | $4.02 \cdot 10^{-2}$ | $6.5 \cdot 10^{-2}$ | $2.79 \cdot 10^{-3}$ | $3.19 \cdot 10^{-3}$ | $3.58 \cdot 10^{-3}$ | $2.47 \cdot 10^{-3}$ | $2.7 \cdot 10^{0}$ | $1.08 \cdot 10^{-1}$ | $9.75 \cdot 10^{-3}$ |
| NO-LIDK$^\diamond$ | $3.08 \cdot 10^{-2}$ | $3.13 \cdot 10^{-2}$ | $4.89 \cdot 10^{-2}$ | $2.66 \cdot 10^{-3}$ | $2.5 \cdot 10^{-3}$ | $2.38 \cdot 10^{-3}$ | $2.12 \cdot 10^{-3}$ | $2.31 \cdot 10^{0}$ | $8.17 \cdot 10^{-2}$ | $7.52 \cdot 10^{-3}$ |
| **LOGLO-FNO** | $\mathbf{2.31 \cdot 10^{-2}}$ | $\mathbf{2.34 \cdot 10^{-2}}$ | $\mathbf{4.07 \cdot 10^{-2}}$ | $\mathbf{1.95 \cdot 10^{-3}}$ | $\mathbf{1.64 \cdot 10^{-3}}$ | $\mathbf{1.35 \cdot 10^{-3}}$ | $\mathbf{1.56 \cdot 10^{-3}}$ | $\mathbf{1.99 \cdot 10^{0}}$ | $\mathbf{8.11 \cdot 10^{-2}}$ | $\mathbf{4.85 \cdot 10^{-3}}$ |
| REL. % DIFF | -31.82% | -31.75% | -27.61% | -27.55% | -36.7% | -50% | -33.4 % | -20.26 % | -31.44% | -41.58% |
| | | | | **10-step Autoregressive Evaluation** | | | | | | |
| FNO | $5.06 \cdot 10^{-2}$ | $4.76 \cdot 10^{-2}$ | $6.33 \cdot 10^{-2}$ | $4.14 \cdot 10^{-3}$ | $4.58 \cdot 10^{-3}$ | $4.79 \cdot 10^{-3}$ | $3.22 \cdot 10^{-3}$ | $3.48 \cdot 10^{0}$ | $1.11 \cdot 10^{-1}$ | $1.04 \cdot 10^{-2}$ |
| NO-LIDK* | $6.15 \cdot 10^{-2}$ | $5.87 \cdot 10^{-2}$ | $7.68 \cdot 10^{-2}$ | $4.63 \cdot 10^{-3}$ | $5.88 \cdot 10^{-3}$ | $6.35 \cdot 10^{-3}$ | $3.59 \cdot 10^{-3}$ | $3.67 \cdot 10^{0}$ | $9.65 \cdot 10^{-2}$ | $1.35 \cdot 10^{-2}$ |
| NO-LIDK$^\diamond$ | $4.72 \cdot 10^{-2}$ | $4.52 \cdot 10^{-2}$ | $5.6 \cdot 10^{-2}$ | $4.39 \cdot 10^{-3}$ | $4.55 \cdot 10^{-3}$ | $4.37 \cdot 10^{-3}$ | $3.02 \cdot 10^{-3}$ | $3.55 \cdot 10^{0}$ | $7.95 \cdot 10^{-2}$ | $9.75 \cdot 10^{-3}$ |
| **LOGLO-FNO** | $\mathbf{3.43 \cdot 10^{-2}}$ | $\mathbf{3.25 \cdot 10^{-2}}$ | $\mathbf{4.44 \cdot 10^{-2}}$ | $\mathbf{3.16 \cdot 10^{-3}}$ | $\mathbf{2.9 \cdot 10^{-3}}$ | $\mathbf{2.74 \cdot 10^{-3}}$ | $\mathbf{2.19 \cdot 10^{-3}}$ | $\mathbf{2.67 \cdot 10^{0}}$ | $\mathbf{7.47 \cdot 10^{-2}}$ | $\mathbf{6.28 \cdot 10^{-3}}$ |
| REL. % DIFF | -32.33% | -31.75% | -29.77% | -23.71% | -36.74% | -42.7% | -31.99% | -23.2% | -32.65% | -39.84% |
| | | | | **15-step Autoregressive Evaluation** | | | | | | |
| FNO | $6.87 \cdot 10^{-2}$ | $6.37 \cdot 10^{-2}$ | $7.35 \cdot 10^{-2}$ | $5.55 \cdot 10^{-3}$ | $6.91 \cdot 10^{-3}$ | $7.19 \cdot 10^{-3}$ | $4.11 \cdot 10^{-3}$ | $3.98 \cdot 10^{0}$ | $1.07 \cdot 10^{-1}$ | $1.31 \cdot 10^{-2}$ |
| NO-LIDK* | $8.44 \cdot 10^{-2}$ | $7.92 \cdot 10^{-2}$ | $9.08 \cdot 10^{-2}$ | $6.68 \cdot 10^{-3}$ | $9.04 \cdot 10^{-3}$ | $9.36 \cdot 10^{-3}$ | $4.61 \cdot 10^{-3}$ | $4.09 \cdot 10^{0}$ | $9.57 \cdot 10^{-2}$ | $1.79 \cdot 10^{-2}$ |
| NO-LIDK$^\diamond$ | $6.81 \cdot 10^{-2}$ | $6.49 \cdot 10^{-2}$ | $6.91 \cdot 10^{-2}$ | $6.51 \cdot 10^{-3}$ | $7.78 \cdot 10^{-3}$ | $7.24 \cdot 10^{-3}$ | $4.08 \cdot 10^{-3}$ | $4.4 \cdot 10^{0}$ | $8.97 \cdot 10^{-2}$ | $1.35 \cdot 10^{-2}$ |
| **LOGLO-FNO** | $\mathbf{4.76 \cdot 10^{-2}}$ | $\mathbf{4.45 \cdot 10^{-2}}$ | $\mathbf{5.16 \cdot 10^{-2}}$ | $\mathbf{4.49 \cdot 10^{-3}}$ | $\mathbf{4.49 \cdot 10^{-3}}$ | $\mathbf{4.53 \cdot 10^{-3}}$ | $\mathbf{2.91 \cdot 10^{-3}}$ | $\mathbf{3.49 \cdot 10^{0}}$ | $\mathbf{7.4 \cdot 10^{-2}}$ | $\mathbf{8.02 \cdot 10^{-3}}$ |
| REL. % DIFF | -30.73% | -30.12% | -29.86% | -19.17% | -35.04% | -36.95% | -29.04% | -12.33% | -30.94% | -38.74% |
| | | | | **20-step (full trajectory) Autoregressive Evaluation** | | | | | | |
| FNO | $8.41 \cdot 10^{-2}$ | $7.86 \cdot 10^{-2}$ | $8.54 \cdot 10^{-2}$ | $6.68 \cdot 10^{-3}$ | $8.94 \cdot 10^{-3}$ | $9.24 \cdot 10^{-3}$ | $4.8 \cdot 10^{-3}$ | $4.46 \cdot 10^{0}$ | $1.08 \cdot 10^{-1}$ | $1.54 \cdot 10^{-2}$ |
| NO-LIDK* | $1.03 \cdot 10^{-1}$ | $9.75 \cdot 10^{-2}$ | $1.06 \cdot 10^{-1}$ | $8.5 \cdot 10^{-3}$ | $1.18 \cdot 10^{-2}$ | $1.18 \cdot 10^{-2}$ | $5.36 \cdot 10^{-3}$ | $4.41 \cdot 10^{0}$ | $1.01 \cdot 10^{-1}$ | $2.16 \cdot 10^{-2}$ |
| NO-LIDK$^\diamond$ | $8.79 \cdot 10^{-2}$ | $8.48 \cdot 10^{-2}$ | $8.67 \cdot 10^{-2}$ | $8.82 \cdot 10^{-3}$ | $1.14 \cdot 10^{-2}$ | $9.97 \cdot 10^{-3}$ | $4.95 \cdot 10^{-3}$ | $4.99 \cdot 10^{0}$ | $9.92 \cdot 10^{-2}$ | $1.74 \cdot 10^{-2}$ |
| **LOGLO-FNO** | $\mathbf{6.02 \cdot 10^{-2}}$ | $\mathbf{5.68 \cdot 10^{-2}}$ | $\mathbf{6.15 \cdot 10^{-2}}$ | $\mathbf{5.76 \cdot 10^{-3}}$ | $\mathbf{6.12 \cdot 10^{-3}}$ | $\mathbf{6.25 \cdot 10^{-3}}$ | $\mathbf{3.55 \cdot 10^{-3}}$ | $\mathbf{3.95 \cdot 10^{0}}$ | $\mathbf{7.66 \cdot 10^{-2}}$ | $\mathbf{9.85 \cdot 10^{-3}}$ |
| REL. % DIFF | -28.4% | -27.74% | -28% | -13.77% | -31.53% | -32.37% | -26.12% | -11.43% | -28.92% | -35.93% |

We observe that LOGLO-FNO significantly and consistently outperforms Base FNO and the state-of-the-art NO-LIDK baselines on both 1-step and autoregressive rollout evaluations, up to the full trajectory length of 20 timesteps.

**Turbulent Radiative Layer 3D (TRL3D).** To complement the results in the main paper, we present the full set of evaluation metrics for the 1-step evaluation on the test set of Turbulent Radiative Layer 3D problem in Table 12.

Table 12: 1-step evaluation of LOGLO-FNO compared with state-of-the-art baselines on the test set of Turbulent Radiative Mixing Layer 3D dataset (Fielding et al., 2020; Ohana et al., 2024). REL. % DIFF indicates an improvement (-) or degradation (+) with respect to base FNO, which uses a spectral filter size of 12 and 48 hidden channels. LOGLO-FNO uses 12 and (16, 16, 17)$^\star$ modes in the global and local branches, respectively, whereas the width is set to 52. ($^\star$This is because the Turbulent Radiative Layer 3D dataset has 2× more sampling points on the z-axis relative to the x and y axes resolutions – (128 × 128 × 256).) Base FNO$^\ddagger$ means our implementation of FNO (see Figure 1).

| Model | RMSE (↓) | nRMSE | bRMSE | cRMSE | fRMSE(L) | fRMSE(M) | fRMSE(H) | MaxError (↓) |
|---|---|---|---|---|---|---|---|---|
| **1-step Evaluation** | | | | | | | | |
| LSM | $5.67 \cdot 10^{-1}$ | $8.39 \cdot 10^{-1}$ | $1.25 \cdot 10^{0}$ | $1.41 \cdot 10^{-1}$ | $2.4 \cdot 10^{-1}$ | $9.7 \cdot 10^{-2}$ | $3.43 \cdot 10^{-2}$ | $1.59 \cdot 10^{1}$ |
| NO-LIDK$^\diamond$ | $1.04 \cdot 10^{0}$ | $4.34 \cdot 10^{-1}$ | $4.59 \cdot 10^{0}$ | $2.17 \cdot 10^{-2}$ | $6.87 \cdot 10^{-2}$ | $1.22 \cdot 10^{-1}$ | $1.23 \cdot 10^{-1}$ | $1.85 \cdot 10^{1}$ |
| Base FNO$^\ddagger$ | $2.84 \cdot 10^{-1}$ | $3.06 \cdot 10^{-1}$ | $7.05 \cdot 10^{-1}$ | $2.04 \cdot 10^{-2}$ | $4.87 \cdot 10^{-2}$ | $4.58 \cdot 10^{-2}$ | $2.89 \cdot 10^{-2}$ | $1.12 \cdot 10^{1}$ |
| **LOGLO-FNO** | $\mathbf{2.45} \cdot 10^{-1}$ | $\mathbf{2.65} \cdot 10^{-1}$ | $\mathbf{6.33} \cdot 10^{-1}$ | $\mathbf{1.76} \cdot 10^{-2}$ | $\mathbf{4.45} \cdot 10^{-2}$ | $\mathbf{3.68} \cdot 10^{-2}$ | $\mathbf{2.48} \cdot 10^{-2}$ | $\mathbf{1.06} \cdot 10^{1}$ |
| REL. % DIFF | -13.55% | -13.27% | -10.2% | -13.8% | -8.61% | -19.57% | -14.01% | -5.17% |

### G.2.2 Autoregressive Training and Evaluation

In Table 13, we present the evaluation results with the full set of evaluation metrics for the fully autoregressive training on the Diffusion-Reaction 2D problem.

Table 13: Fully autoregressive evaluation of LOGLO-FNO compared with SOTA baselines on the test set of challenging 2D Diffusion-Reaction coupled problem from PDEBench (Takamoto et al., 2022). We also report the REL. % DIFF to indicate the error improvement (-) or degradation (+) w.r.t FNO.

| Model | RMSE (↓) | nRMSE | bRMSE | cRMSE | fRMSE(L) | fRMSE(M) | fRMSE(H) | MaxError (↓) | MELR (↓) | WLR (↓) |
|---|---|---|---|---|---|---|---|---|---|---|
| U-Net | $6.1 \cdot 10^{-2}$ | $8.4 \cdot 10^{-1}$ | $7.8 \cdot 10^{-2}$ | $3.9 \cdot 10^{-2}$ | $1.7 \cdot 10^{-2}$ | $8.2 \cdot 10^{-4}$ | $5.7 \cdot 10^{-2}$ | $1.9 \cdot 10^{-1}$ | ✗ | ✗ |
| Transolver | $1.74 \cdot 10^{-2}$ | $2.63 \cdot 10^{-1}$ | $3.05 \cdot 10^{-2}$ | $3.37 \cdot 10^{-3}$ | $2.99 \cdot 10^{-3}$ | $2.03 \cdot 10^{-3}$ | $5.51 \cdot 10^{-4}$ | $1.11 \cdot 10^{-1}$ | $7.19 \cdot 10^{-1}$ | $1.15 \cdot 10^{-1}$ |
| U-FNO | $1.4 \cdot 10^{-2}$ | $2.6 \cdot 10^{-1}$ | $2.0 \cdot 10^{-2}$ | $4.3 \cdot 10^{-3}$ | $3.4 \cdot 10^{-3}$ | $1.6 \cdot 10^{-3}$ | $2.6 \cdot 10^{-4}$ | $7.8 \cdot 10^{-2}$ | $4.5 \cdot 10^{-1}$ | $7.9 \cdot 10^{-2}$ |
| FNO | $5.2 \cdot 10^{-3}$ | $8.3 \cdot 10^{-2}$ | $1.5 \cdot 10^{-2}$ | $1.2 \cdot 10^{-3}$ | $6.2 \cdot 10^{-4}$ | $5.6 \cdot 10^{-4}$ | $2.4 \cdot 10^{-4}$ | $7.3 \cdot 10^{-2}$ | $2.96 \cdot 10^{-1}$ | $1.3 \cdot 10^{-2}$ |
| F-FNO | $4.3 \cdot 10^{-3}$ | $7.0 \cdot 10^{-2}$ | $7.9 \cdot 10^{-3}$ | $2.8 \cdot 10^{-3}$ | $9.6 \cdot 10^{-4}$ | $4.7 \cdot 10^{-4}$ | $\mathbf{1.3} \cdot 10^{-4}$ | $5.3 \cdot 10^{-2}$ | $2.0 \cdot 10^{-1}$ | $1.3 \cdot 10^{-2}$ |
| LSM | $2.81 \cdot 10^{-2}$ | $4.47 \cdot 10^{-1}$ | $3.45 \cdot 10^{-2}$ | $5.92 \cdot 10^{-3}$ | $7.17 \cdot 10^{-3}$ | $2.4 \cdot 10^{-3}$ | $3.67 \cdot 10^{-4}$ | $1.32 \cdot 10^{-1}$ | $3.43 \cdot 10^{-1}$ | $2.08 \cdot 10^{-1}$ |
| NO-LIDK (loc. int) | $\mathbf{3.6} \cdot 10^{-3}$ | $\mathbf{6.3} \cdot 10^{-2}$ | $1.0 \cdot 10^{-2}$ | $4.8 \cdot 10^{-4}$ | $4.0 \cdot 10^{-4}$ | $4.6 \cdot 10^{-4}$ | $1.5 \cdot 10^{-4}$ | $5.0 \cdot 10^{-2}$ | ✗ | ✗ |
| **LOGLO-FNO** | $3.89 \cdot 10^{-3}$ | $6.4 \cdot 10^{-2}$ | $\mathbf{5.2} \cdot 10^{-3}$ | $\mathbf{4.6} \cdot 10^{-4}$ | $\mathbf{2.8} \cdot 10^{-4}$ | $\mathbf{3.2} \cdot 10^{-4}$ | $1.9 \cdot 10^{-4}$ | $\mathbf{2.2} \cdot 10^{-2}$ | $\mathbf{1.6} \cdot 10^{-1}$ | $\mathbf{7.9} \cdot 10^{-3}$ |
| REL. % DIFF | -25.19 % | -22.75 % | -65.13 % | -61.67 % | -54.84 % | -42.86 % | -20.83 % | -69.97 % | -44.09 % | -36.98 % |

### G.3 Further Improvements with Attentional Feature Fusion, Multi-scale Channel Attention, and $\mathbb{SPHERE}$ Loss

**2D Kolmogorov Flow.** In this section, we explore ways to further improve the results by considering advanced feature fusion strategies. The global and local branches in LOGLO-FNO models input features on different scales, whereas the HFP module learns different latent features that are representative of high frequencies. Therefore, we propose to employ sophisticated feature fusion modules targeted towards a seamless aggregation of these multi-scale and diverse features, as opposed to a simple summation (Figure 1). Towards this end, we adopt the multi-scale channel attention and attentional feature fusion modules introduced in Dai et al. (2021) to fuse the features of the local and global branches and the HFP branch. The results of an experiment with this fusion in LOGLO-FNO, as well as penalizing the $\mathbb{SPHERE}$ loss on the Kolmogorov Flow 2D dataset, are presented in Table 14. We term this model as LOGLO-FNO+. Comparing the results in Table 1 and Table 14, we observe that these improvements lead to an additional 12% in 1-step nRMSE and (6%, 11%) for the mid-and high-frequency errors, respectively. The MELR and WLR errors are also down by 8% and 7%, respectively. In a similar manner, we see consistent improvements in 5-step nRMSE (8%), fRMSE-mid (2%), fRMSE-high (8%), MELR and WLR (3% each) over the LOGLO-FNO model not using feature fusion and the $\mathbb{SPHERE}$ loss.

Table 14: 1-step and 5-step AR evaluation of LOGLO-FNO+ employing multi-scale feature fusion and patch-based energy spectra loss ($\mathbb{SPHERE}$) compared with SOTA baselines on the test set of 2D Kolmogorov Flow (Li et al., 2022b). We also report the REL. % DIFF to indicate the improvement (-) or the degradation (+) with respect to FNO. The number of modes used in the global branch is 40, and the local branch uses a patch size of $16 \times 16$. The number of hidden channels has been set as 65. NO-LIDK[†] indicates the usage of both differential kernel and localized integral kernel layers in its architecture, NO-LIDK[*] denotes the model employing only the local integral kernel layer, and NO-LIDK[◇] stands for the use of only the differential layer as an additional parallel layer to the global spectral convolution layer of base FNO (see Table 4 in Liu-Schiaffini et al. (2024)). Transolver[⋆] indicates a longer training time of the model for 500 epochs due to convergence issues at shorter training epochs of 136. We boldface the **best result** and underline the second-best result.

| Eval Type | Model | RMSE (↓) | nRMSE | bRMSE | cRMSE | fRMSE(L) | fRMSE(M) | fRMSE(H) | MaxError (↓) | MELR (↓) | WLR (↓) |
|---|---|---|---|---|---|---|---|---|---|---|---|
| | | | | | **1-step Evaluation** | | | | | | |
| | U-Net | $7.17 \cdot 10^{-1}$ | $1.3 \cdot 10^{-1}$ | $1.47 \cdot 10^{0}$ | $1.74 \cdot 10^{-2}$ | $2.24 \cdot 10^{-2}$ | $3.57 \cdot 10^{-2}$ | $4.39 \cdot 10^{-2}$ | $2.01 \cdot 10^{1}$ | $1.64 \cdot 10^{-1}$ | $1.48 \cdot 10^{-2}$ |
| | Transolver[⋆] | $7.69 \cdot 10^{-1}$ | $1.39 \cdot 10^{-1}$ | $1.42 \cdot 10^{0}$ | $7.65 \cdot 10^{-3}$ | $2.74 \cdot 10^{-2}$ | $4.48 \cdot 10^{-2}$ | $4.65 \cdot 10^{-2}$ | $1.52 \cdot 10^{1}$ | $1.43 \cdot 10^{-1}$ | $1.83 \cdot 10^{-2}$ |
| | FNO | $8.08 \cdot 10^{-1}$ | $1.47 \cdot 10^{-1}$ | $7.94 \cdot 10^{-1}$ | $1.1 \cdot 10^{-2}$ | $1.36 \cdot 10^{-2}$ | $2.05 \cdot 10^{-2}$ | $4.7 \cdot 10^{-2}$ | $1.46 \cdot 10^{1}$ | $5.2 \cdot 10^{-1}$ | $2.83 \cdot 10^{-2}$ |
| | F-FNO | $7.53 \cdot 10^{-1}$ | $1.37 \cdot 10^{-1}$ | $7.41 \cdot 10^{-1}$ | $1.5 \cdot 10^{-2}$ | $1.49 \cdot 10^{-2}$ | $2.15 \cdot 10^{-2}$ | $4.36 \cdot 10^{-2}$ | $1.42 \cdot 10^{1}$ | $4.74 \cdot 10^{-1}$ | $2.28 \cdot 10^{-2}$ |
| 1-step | LSM | $7.49 \cdot 10^{-1}$ | $1.36 \cdot 10^{-1}$ | $1.47 \cdot 10^{0}$ | $1.36 \cdot 10^{-2}$ | $2.59 \cdot 10^{-2}$ | $4.42 \cdot 10^{-2}$ | $4.64 \cdot 10^{-2}$ | $2.05 \cdot 10^{1}$ | $1.43 \cdot 10^{-1}$ | $1.68 \cdot 10^{-2}$ |
| | U-FNO | $6.13 \cdot 10^{-1}$ | $1.12 \cdot 10^{-1}$ | $1.0 \cdot 10^{0}$ | $\mathbf{7.09 \cdot 10^{-3}}$ | $1.27 \cdot 10^{-2}$ | $2.22 \cdot 10^{-2}$ | $3.71 \cdot 10^{-2}$ | $1.64 \cdot 10^{1}$ | $1.38 \cdot 10^{-1}$ | $1.09 \cdot 10^{-2}$ |
| | NO-LIDK[*] | $7.25 \cdot 10^{-1}$ | $1.33 \cdot 10^{-1}$ | $1.12 \cdot 10^{0}$ | $9.05 \cdot 10^{-3}$ | $1.45 \cdot 10^{-2}$ | $2.69 \cdot 10^{-2}$ | $4.55 \cdot 10^{-2}$ | $1.65 \cdot 10^{1}$ | $1.85 \cdot 10^{-1}$ | $1.54 \cdot 10^{-2}$ |
| | NO-LIDK[◇] | $6.13 \cdot 10^{-1}$ | $1.11 \cdot 10^{-1}$ | $5.95 \cdot 10^{-1}$ | $1.46 \cdot 10^{-2}$ | $1.65 \cdot 10^{-2}$ | $2.29 \cdot 10^{-2}$ | $3.91 \cdot 10^{-2}$ | $1.5 \cdot 10^{1}$ | $9.57 \cdot 10^{-2}$ | $1.1 \cdot 10^{-2}$ |
| | NO-LIDK[†] | $5.86 \cdot 10^{-1}$ | $1.07 \cdot 10^{-1}$ | $5.64 \cdot 10^{-1}$ | $1.05 \cdot 10^{-2}$ | $1.44 \cdot 10^{-2}$ | $2.46 \cdot 10^{-2}$ | $3.82 \cdot 10^{-2}$ | $1.47 \cdot 10^{1}$ | $\mathbf{7.11 \cdot 10^{-2}}$ | $1.0 \cdot 10^{-2}$ |
| | LOGLO-FNO | $\underline{5.89 \cdot 10^{-1}}$ | $\underline{1.07 \cdot 10^{-1}}$ | $6.74 \cdot 10^{-1}$ | $\underline{7.23 \cdot 10^{-3}}$ | $\mathbf{1.21 \cdot 10^{-2}}$ | $\underline{1.81 \cdot 10^{-2}}$ | $\underline{3.54 \cdot 10^{-2}}$ | $\mathbf{1.33 \cdot 10^{1}}$ | $1.29 \cdot 10^{-1}$ | $1.06 \cdot 10^{-2}$ |
| | LOGLO-FNO+ | $\mathbf{4.88 \cdot 10^{-1}}$ | $\mathbf{8.87 \cdot 10^{-2}}$ | $\mathbf{5.57 \cdot 10^{-1}}$ | $8.78 \cdot 10^{-3}$ | $\underline{1.31 \cdot 10^{-2}}$ | $\mathbf{1.68 \cdot 10^{-2}}$ | $\mathbf{3.04 \cdot 10^{-2}}$ | $\underline{1.35 \cdot 10^{1}}$ | $\underline{7.56 \cdot 10^{-2}}$ | $\mathbf{8.76 \cdot 10^{-3}}$ |
| REL. % DIFF | w/o fusion | -27.1 % | -27.21 % | -15.12 % | -34.31 % | -11.22 % | -11.88 % | -24.64 % | -8.99 % | -75.12 % | -62.63 % |
| REL. % DIFF | w/ fusion | -39.6 % | -39.78 % | -29.86% | -20.28% | -3.87 % | -18.13 % | -35.39 % | -7.91 % | -83.54 % | -69.03% |
| | | | | | **5-step Autoregressive Evaluation** | | | | | | |
| | U-Net | $1.51 \cdot 10^{0}$ | $2.65 \cdot 10^{-1}$ | $2.31 \cdot 10^{0}$ | $5.39 \cdot 10^{-2}$ | $6.53 \cdot 10^{-2}$ | $1.04 \cdot 10^{-1}$ | $9.38 \cdot 10^{-2}$ | $1.81 \cdot 10^{1}$ | $2.13 \cdot 10^{-1}$ | $3.52 \cdot 10^{-2}$ |
| | Transolver[⋆] | $1.91 \cdot 10^{0}$ | $3.36 \cdot 10^{-1}$ | $2.59 \cdot 10^{0}$ | $\mathbf{6.12 \cdot 10^{-3}}$ | $7.44 \cdot 10^{-2}$ | $1.51 \cdot 10^{-1}$ | $1.18 \cdot 10^{-1}$ | $2.36 \cdot 10^{1}$ | $2.45 \cdot 10^{-1}$ | $4.92 \cdot 10^{-2}$ |
| | FNO | $1.33 \cdot 10^{0}$ | $2.35 \cdot 10^{-1}$ | $1.34 \cdot 10^{0}$ | $1.46 \cdot 10^{-2}$ | $3.37 \cdot 10^{-2}$ | $5.8 \cdot 10^{-2}$ | $8.37 \cdot 10^{-2}$ | $1.6 \cdot 10^{1}$ | $6.18 \cdot 10^{-1}$ | $4.93 \cdot 10^{-2}$ |
| | F-FNO | $1.29 \cdot 10^{0}$ | $2.28 \cdot 10^{-1}$ | $1.27 \cdot 10^{0}$ | $2.28 \cdot 10^{-2}$ | $3.60 \cdot 10^{-2}$ | $5.52 \cdot 10^{-2}$ | $8.12 \cdot 10^{-2}$ | $1.50 \cdot 10^{1}$ | $5.37 \cdot 10^{-1}$ | $3.99 \cdot 10^{-2}$ |
| | LSM | $1.81 \cdot 10^{0}$ | $3.18 \cdot 10^{-1}$ | $2.76 \cdot 10^{0}$ | $3.87 \cdot 10^{-2}$ | $7.6 \cdot 10^{-2}$ | $1.44 \cdot 10^{-1}$ | $1.12 \cdot 10^{-1}$ | $2.08 \cdot 10^{1}$ | $2.04 \cdot 10^{-1}$ | $4.39 \cdot 10^{-2}$ |
| 5-step AR | U-FNO | $1.15 \cdot 10^{0}$ | $2.03 \cdot 10^{-1}$ | $1.45 \cdot 10^{0}$ | $\underline{6.3 \cdot 10^{-3}}$ | $\mathbf{2.69 \cdot 10^{-2}}$ | $5.33 \cdot 10^{-2}$ | $7.35 \cdot 10^{-2}$ | $1.56 \cdot 10^{1}$ | $2.01 \cdot 10^{-1}$ | $2.51 \cdot 10^{-2}$ |
| | NO-LIDK[*] | $1.36 \cdot 10^{0}$ | $2.39 \cdot 10^{-1}$ | $1.8 \cdot 10^{0}$ | $1.1 \cdot 10^{-2}$ | $3.14 \cdot 10^{-2}$ | $6.86 \cdot 10^{-2}$ | $8.91 \cdot 10^{-2}$ | $1.59 \cdot 10^{1}$ | $2.23 \cdot 10^{-1}$ | $2.93 \cdot 10^{-2}$ |
| | NO-LIDK[◇] | $1.17 \cdot 10^{0}$ | $2.04 \cdot 10^{-1}$ | $1.12 \cdot 10^{0}$ | $1.92 \cdot 10^{-2}$ | $3.71 \cdot 10^{-2}$ | $5.96 \cdot 10^{-2}$ | $7.60 \cdot 10^{-2}$ | $1.61 \cdot 10^{1}$ | $1.42 \cdot 10^{-1}$ | $2.12 \cdot 10^{-2}$ |
| | NO-LIDK[†] | $1.17 \cdot 10^{0}$ | $2.05 \cdot 10^{-1}$ | $1.14 \cdot 10^{0}$ | $1.23 \cdot 10^{-2}$ | $3.31 \cdot 10^{-2}$ | $5.94 \cdot 10^{-2}$ | $7.74 \cdot 10^{-2}$ | $1.56 \cdot 10^{1}$ | $\mathbf{1.03 \cdot 10^{-1}}$ | $2.18 \cdot 10^{-2}$ |
| | LOGLO-FNO | $\underline{1.09 \cdot 10^{0}}$ | $\underline{1.92 \cdot 10^{-1}}$ | $\underline{1.12 \cdot 10^{0}}$ | $8.99 \cdot 10^{-3}$ | $\underline{2.8 \cdot 10^{-2}}$ | $\underline{4.55 \cdot 10^{-2}}$ | $\underline{6.93 \cdot 10^{-2}}$ | $\mathbf{1.26 \cdot 10^{1}}$ | $1.67 \cdot 10^{-1}$ | $\underline{2.07 \cdot 10^{-2}}$ |
| | LOGLO-FNO+ | $\mathbf{9.81 \cdot 10^{-1}}$ | $\mathbf{1.72 \cdot 10^{-1}}$ | $\mathbf{1.02 \cdot 10^{0}}$ | $1.05 \cdot 10^{-2}$ | $2.9 \cdot 10^{-2}$ | $\mathbf{4.43 \cdot 10^{-2}}$ | $\mathbf{6.27 \cdot 10^{-2}}$ | $\underline{1.31 \cdot 10^{1}}$ | $\underline{1.17 \cdot 10^{-1}}$ | $\mathbf{1.90 \cdot 10^{-2}}$ |
| REL. % DIFF | w/o fusion | -18.26 % | -18.39 % | -16.12 % | -38.21 % | -16.86 % | -21.62 % | -17.26 % | -21.31 % | -73.04 % | -58.02 % |
| REL. % DIFF | w/ fusion | -26.37 % | -26.63 % | -23.43 % | -27.99 % | -14.03 % | -23.58 % | -25.09 % | -15.62 % | -77.76 % | -61.43 % |

Note that the terminology *REL. % DIFF w/ fusion* in the Table 14 above indicates the use of $\mathbb{SPHERE}$ loss in addition to feature fusion. This detail has been omitted in the row for an uncluttered presentation.

### G.4 Significance of Adding More Modes in the Fourier Layers of Global Branch

#### G.4.1 Kolmogorov Flow 2D

In this study, we analyze the influence of adding more modes in the global branch while keeping the number of modes constant in the local branch, which is done by using a single local branch with a patch size of $16 \times 16$. This setting yields the local spectral convolution branch with 9 modes, including the Nyquist frequency for the y-axis of the 2D spatial data. Further, we also fix the embedding dimension to 48. Therefore, the number of modes in the global branch is the only varying factor. The experiments are conducted on the turbulent 2D Kolmogorov Flow dataset (Li et al., 2022b).

**1-step Evaluation.** Here, we evaluate the baseline FNO and LOGLO-FNO models on 1-step errors. The LOGLO-FNO models have been trained with a single local branch (patch size $16 \times 16$) in addition to the global branch. Figures 14, 3, 15, and 16 plot xRMSE, MaxError, and the energy spectra (MELR and WLR) metrics.

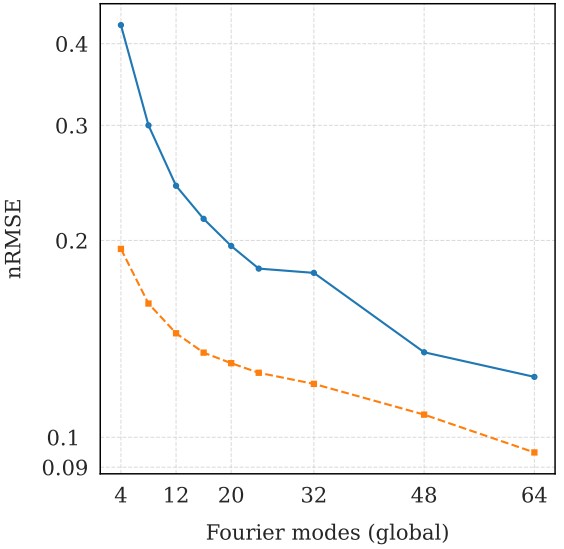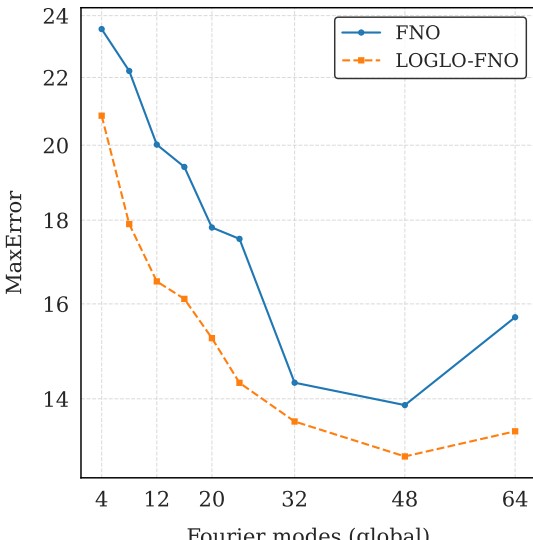

Figure 14: Comparison of FNO vs. LOGLO-FNO showing 1-step nRMSE and MaxError ($\downarrow$) on the test set of Kolmogorov Flow ($Re = 5k$) (Li et al., 2022b) for varying number of modes in global branch and full set of modes (i.e., (16, 9) for patch size $16 \times 16$) in local branch.

In Figure 14, we observe that as we increase the modes in the global branch, it generally leads to a reduction in nRMSE and MaxError for both models, except for a slight increase in MaxError when utilizing the full set of modes, potentially due to overfitting. Also, note that LOGLO-FNO outperforms base FNO significantly, particularly when the modes used in the global branch are fewer (i.e., sparse modes). We hypothesize that the local branch compensates in this case when there is a lack of expressivity. Even when using the full set of available modes, LOGLO-FNO yields improved error. This behavior shows that local convolutions always supplement global convolutions.

Figure 3 in the main paper visualizes the low, mid, and high-frequency band-classified errors for varying numbers of modes in the global branch. A similar trend as observed for nRMSE also holds here. Additionally, we observe that the error improvement for LOGLO-FNO over FNO for the high-frequency is more pronounced than for low and mid-frequencies. Note the logarithmic scale on the y-axis. Further, FNO tends to overfit when employing the full set of modes (fRMSE (Low) & fRMSE (Mid)), suggesting potential overfitting and corroborating the findings of Lanthaler et al. (2024). In contrast, LOGLO-FNO does not exhibit this behavior and achieves the lowest error when utilizing the full set of modes. Notably, LOGLO-FNO with

32 modes attains lower frequency errors than the base FNO with 48 modes, highlighting its more efficient representation of complex features while maintaining robustness against overfitting.

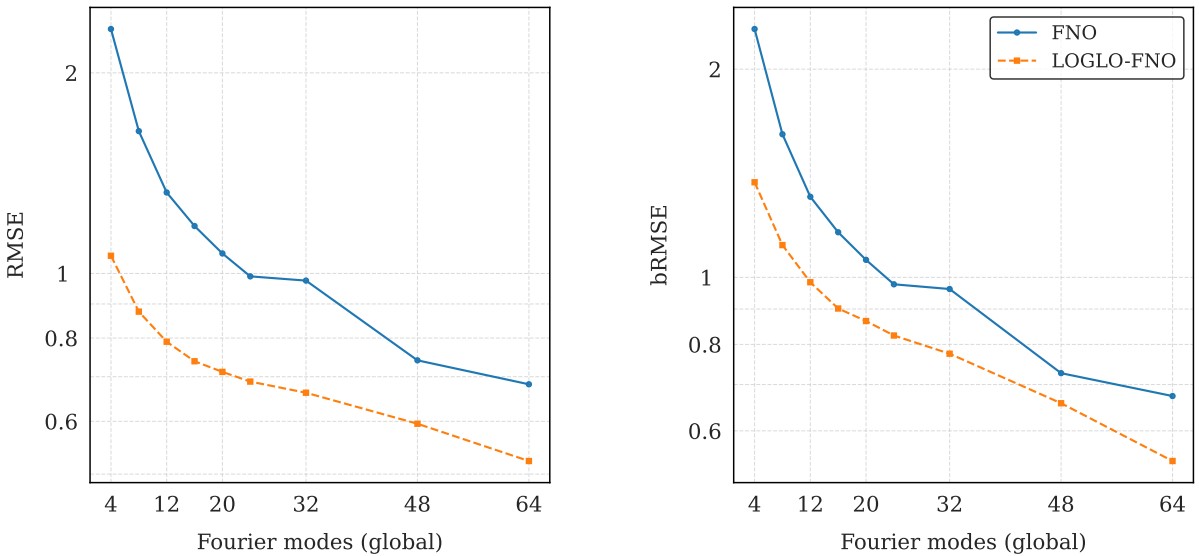

Figure 15: Comparison of FNO vs. LOGLO-FNO showing 1-step RMSE and bRMSE ($\downarrow$) on the test set of Kolmogorov Flow ($Re = 5k$) (Li et al., 2022b) for a varying number of modes in the global branch and full set of modes (i.e., (16, 9) for patch size $16 \times 16$) in the local branch.

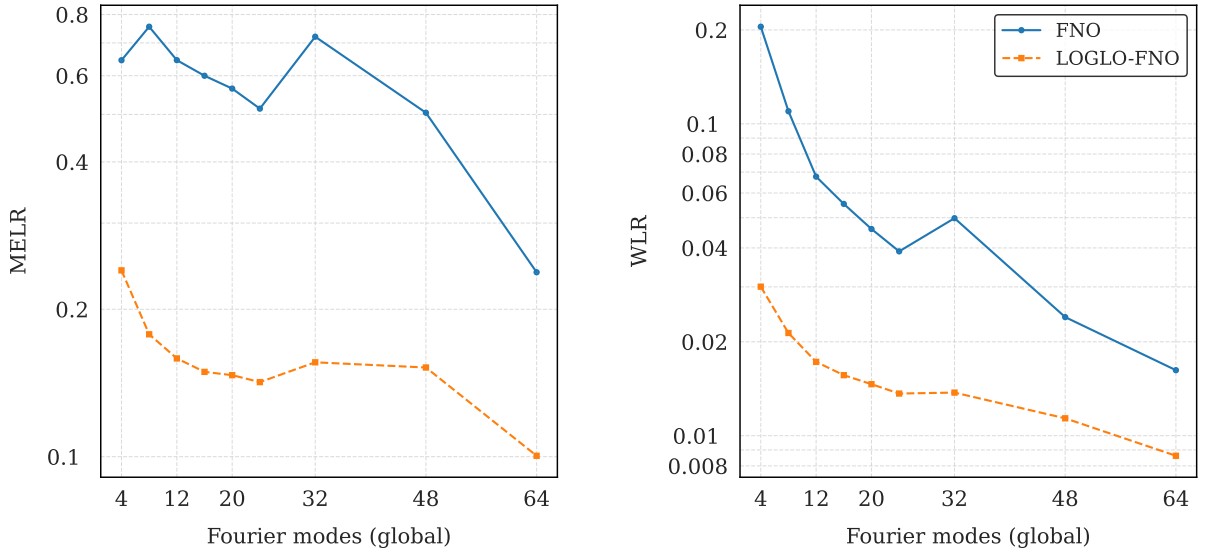

Figure 16: Comparison of FNO vs. LOGLO-FNO showing 1-step MELR and WLR ($\downarrow$) on the test set of Kolmogorov Flow ($Re = 5k$) (Li et al., 2022b) for a varying number of modes in the global branch and full set of modes (i.e., (16, 9) for patch size $16 \times 16$) in the local branch.

In Figure 16, we plot the energy spectra scalar measures, viz. MELR and WLR, which quantify the spectral discrepancy between the prediction and reference simulations. Here, we observe that LOGLO-FNO is significantly better than baseline FNO.

**Extended Autoregressive Evaluation.** In this inference setup, we evaluate the 1-step trained models on autoregressive rollouts for varying numbers of timesteps and evaluate the rollout errors. We roll out the trajectories until reaching 25 timesteps to understand the behavior of LOGLO-FNO compared to base FNO in terms of autoregressive error accumulation.

In Figures 17, 18, 5, and 19 we plot the autoregressive errors of metrics, such as nRMSE, MaxError, bRMSE, fRMSE, MELR, and WLR. We observe that, overall, LOGLO-FNO consistently yields lower errors compared to base FNO and maintains the same behavior throughout the rollout extent. Although the autoregressive error accumulation is not completely mitigated, we can conclude that LOGLO-FNO suffers less from this problem than the baseline FNO model across all evaluated metrics.

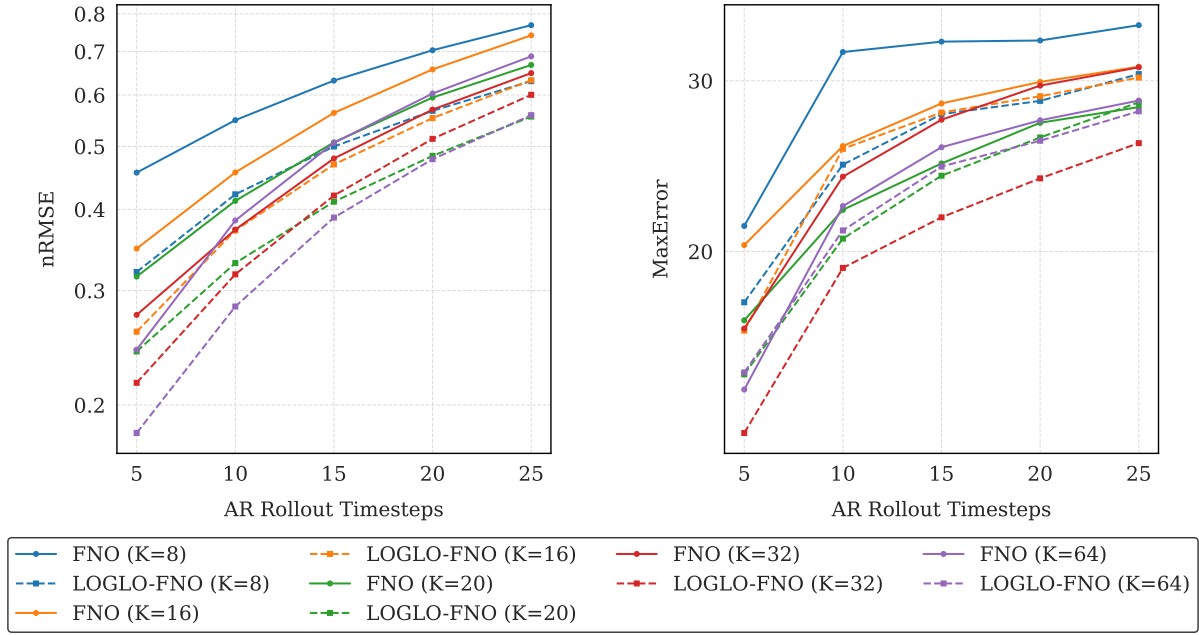

Figure 17: Comparison of FNO vs. LOGLO-FNO for autoregressive rollout showing nRMSE and MaxError (↓) growth over varying number of modes (K) in global branch and timesteps in the trajectories on the test set of Kolmogorov Flow ($Re = 5k$) (Li et al., 2022b). The local branch uses a patch size of $16 \times 16$.

In Figure 19, we visualize the energy spectra deviations through MELR and WLR metrics. We observe that LOGLO-FNO outperforms base FNO by a significant margin, indicating that the deviation or gap between the energy spectra of the ground truth and the predictions of LOGLO-FNO is narrower than that of the baseline FNO model.

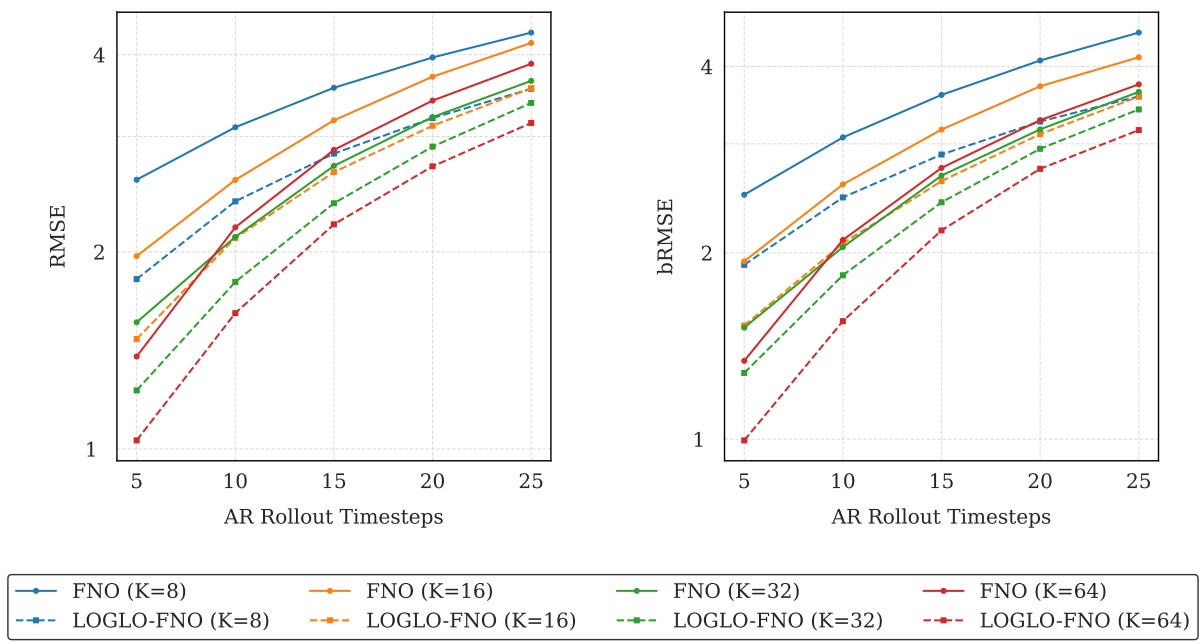

Figure 18: Comparison of FNO vs. LOGLO-FNO for autoregressive rollout showing RMSE and bRMSE ($\downarrow$) error growth over varying modes (K) in global branch and number of timesteps in the trajectories on the test set of Kolmogorov Flow ($Re = 5k$) (Li et al., 2022b). The local branch uses a patch size of $16 \times 16$.

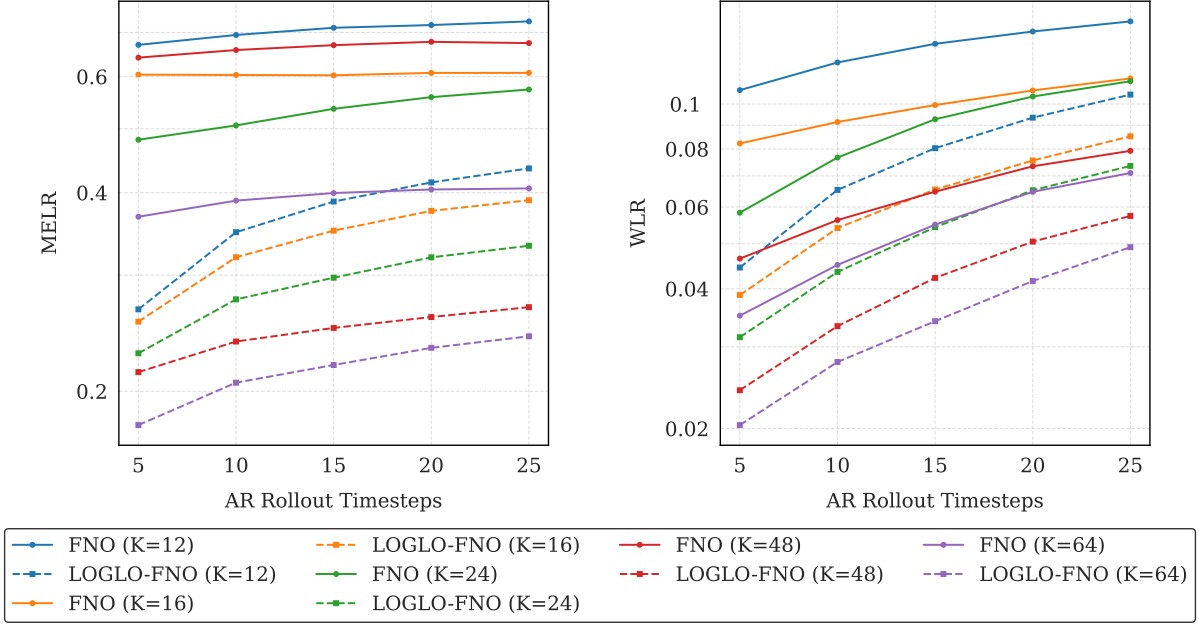

Figure 19: Comparison of FNO vs. LOGLO-FNO for autoregressive rollout showing MELR and WLR ($\downarrow$) error growth over varying modes (K) in global branch and number of timesteps in the trajectories on the test set of Kolmogorov Flow ($Re = 5k$) (Li et al., 2022b). The local branch uses a patch size of $16 \times 16$.

### G.4.2  Turbulent Radiative Layer 3D

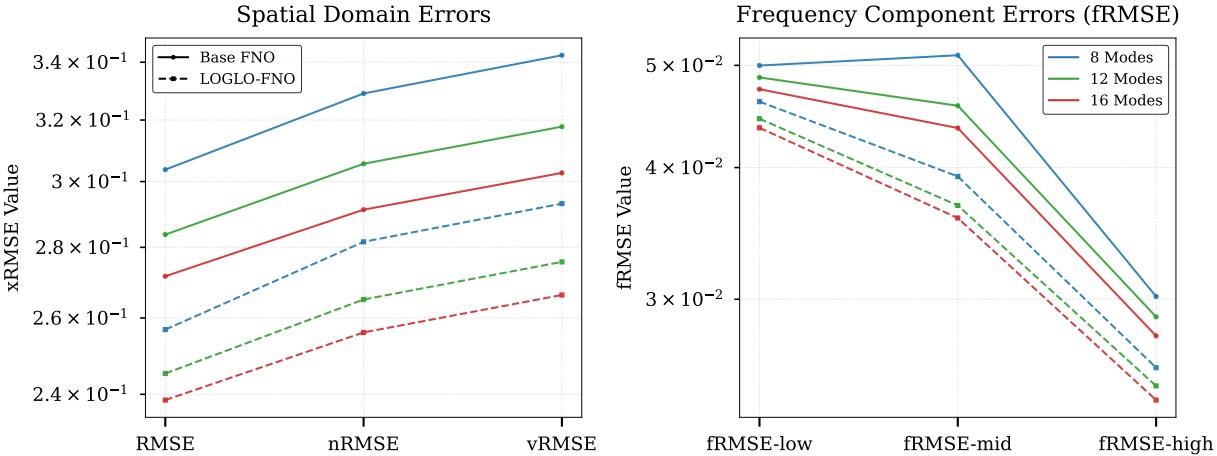

Figure 20: Comparison of base FNO vs. LOGLO-FNO showing 1-step spatial domain xRMSE (left) and frequency space fRMSE (right) over varying Fourier modes (K) in the global branch and the local branch utilizing a patch size of $16 \times 16 \times 32$ and the resulting full set of modes (16, 16, 17) on the test set of the Turbulent Radiative Layer 3D dataset (Ohana et al., 2024).

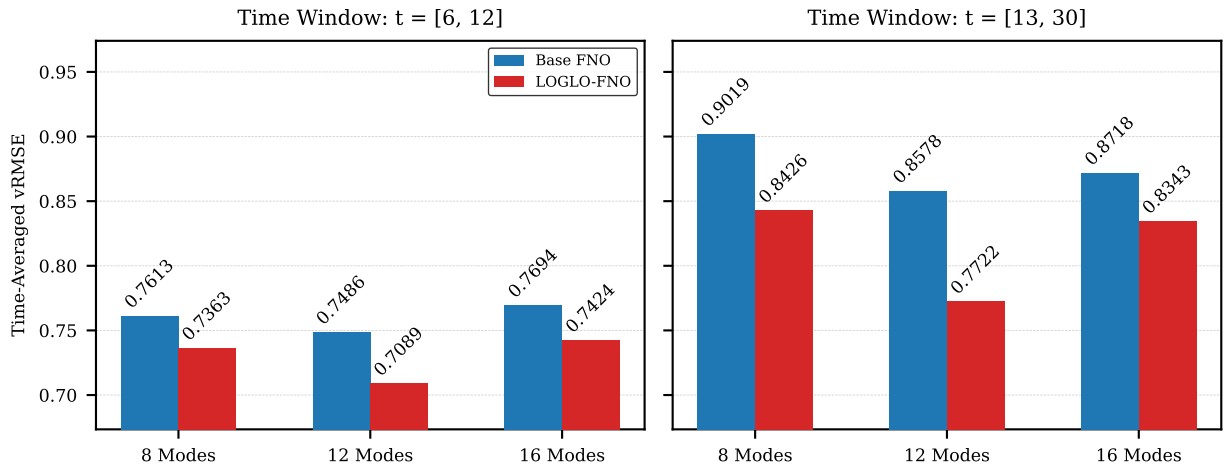

Figure 21: Comparison of base FNO vs. LOGLO-FNO showing vRMSE values for the autoregressive rollout over the time windows [6:12] (left) and [13:30] (right, both ends inclusive), for variable Fourier modes (K) in the global branch while the local branch employing a patch size of $16 \times 16 \times 32$ and the resulting full set of modes (16, 16, 17) on the test set of the Turbulent Radiative Layer 3D dataset (Ohana et al., 2024).

# H    Qualitative Results and Analysis

### H.0.1    Kolmogorov Flow 2D

In this section, we provide qualitative visualizations of the predictions of a few random trajectories from the test set of Kolmogorov Flow 2D dataset (Li et al., 2022b). The predictions are obtained from the models trained with 32 modes in the global branch and a patch size of 16 in the local branch for LOGLO-FNO. Both models use a constant width of 48.

Figure 22 provides a qualitative visualization comparing the ground truth, predictions, and the corresponding absolute errors of two consecutive timesteps of two random trajectories. We observe that LOGLO-FNO retains sharp details, resulting in a noticeable reduction of errors, whereas the base FNO yields smoothed-out predictions. Note that the absolute errors of both predictions are placed on the same scale for a fair comparison.

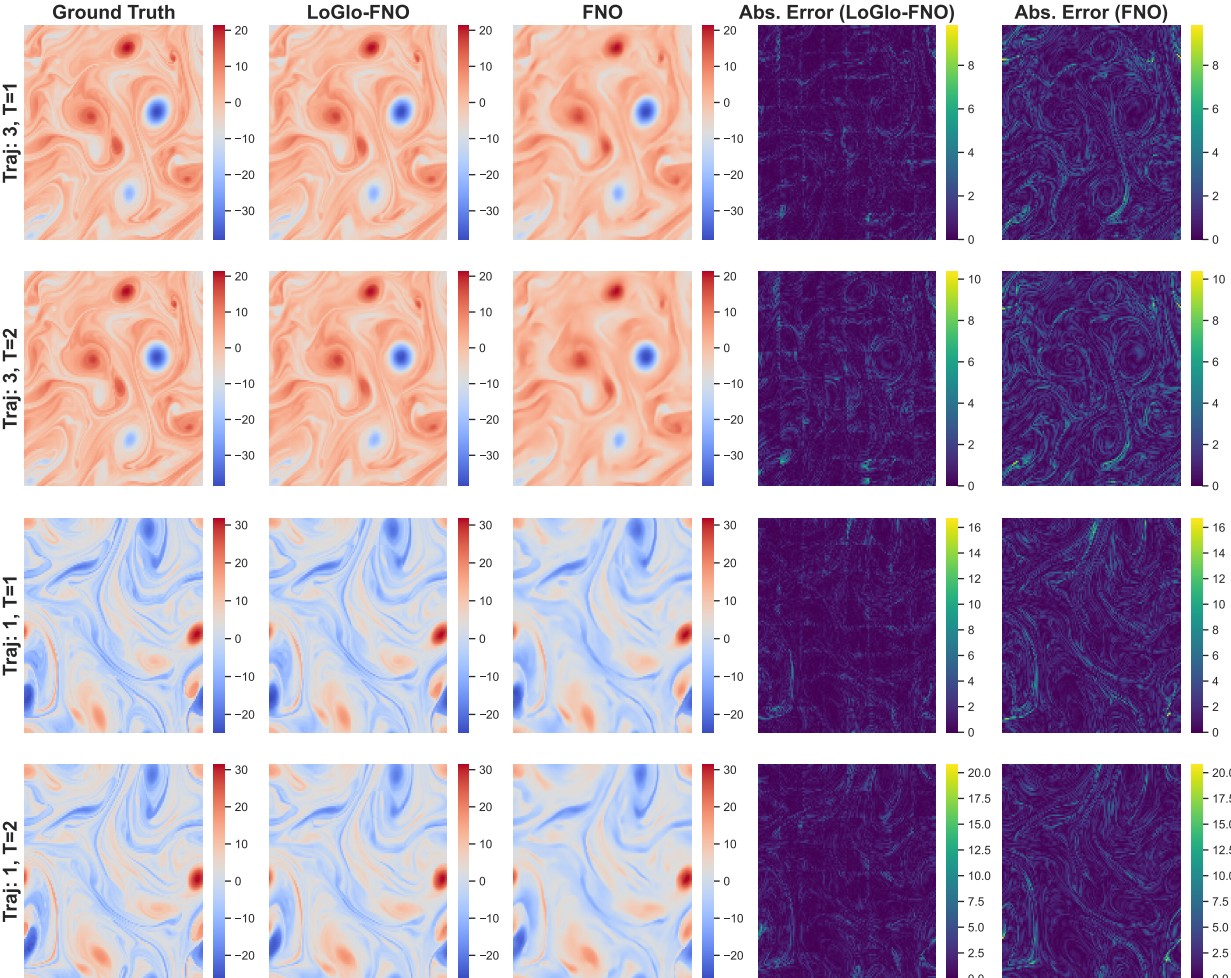

Figure 22: Qualitative comparison of predictions, ground truths, and absolute errors of FNO and LOGLO-FNO for random trajectories from the test set of Kolmogorov Flow ($Re = 5k$) (Li et al., 2022b).

Similar to Figure 22, we visualize the errors of two different trajectories from the test set in Figure 23. However, we additionally localize the regions in the predictions corresponding to regions in the absolute error that exceed 10% of the maximum error in the whole domain. This could result in multiple such regions. Therefore, we limit ourselves to a single contiguous region with the highest intensity in the interest

of providing an uncluttered visualization. The displayed regions inside the red bounding boxes are the ones with the highest error intensity. We observe that the regions overlap for both trajectories at timestep 1. This implies that both models struggle to get the details correct in this region. However, note that the bounding box regions are smaller for LOGLO-FNO compared to base FNO. Hence, we can deduce that LOGLO-FNO is effective in mitigating errors in regions where the base FNO struggles. In scenarios when the bounding boxes do not overlap, as is the case in both model predictions for timestep 2, it is generally desirable to have bounding boxes with smaller areas, as this indicates the errors are localized to a small region and not spread out over a large area.

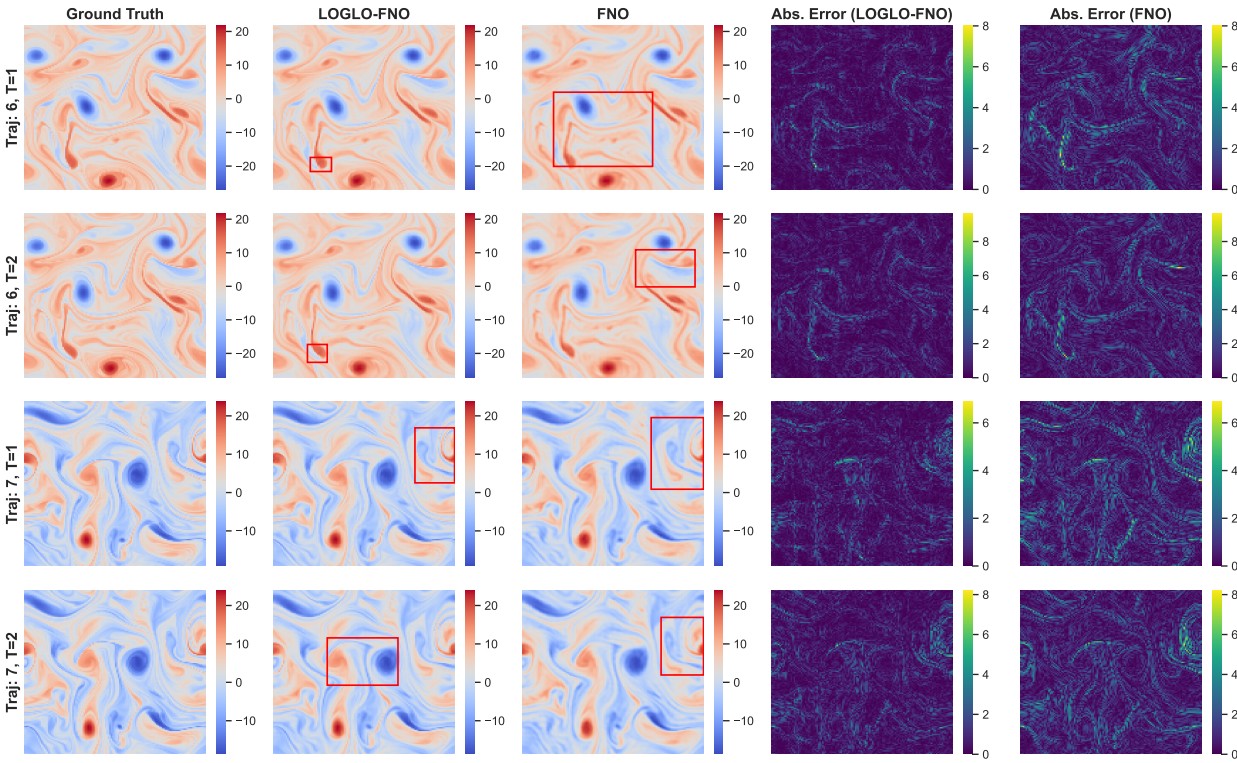

Figure 23: Qualitative comparison of FNO vs. LOGLO-FNO on random samples from the test set of Kolmogorov Flow ($Re = 5k$). The red bounding boxes highlight a single spatially contiguous region where the absolute error exceeds 10% of the maximum error in the domain.

### H.0.2  Turbulent Radiative Layer 3D

In this section, we provide qualitative visualizations of predictions of a few random trajectories from the test set of the Turbulent Radiative Layer 3D dataset (Ohana et al., 2024). The LOGLO-FNO and base FNO predictions are obtained using 1-step evaluation. In the following plots, we show the test set trajectory corresponding to the $t_{cool}$ parameter 0.03. The x, y, and z axes show the spatial extents ($L_x$, $L_y$, and $L_z$) of the simulation. The ground truth solution is repeated in the left column for ease of comparison and clarity in the representation. Note the deviation in the range of density and velocity values predicted by base FNO and LOGLO-FNO with respect to the ground truth. Further, we observe from Figure 24 that base FNO predicts unphysical values (i.e., negative density), whereas the density values predicted by LOGLO-FNO are non-negative.

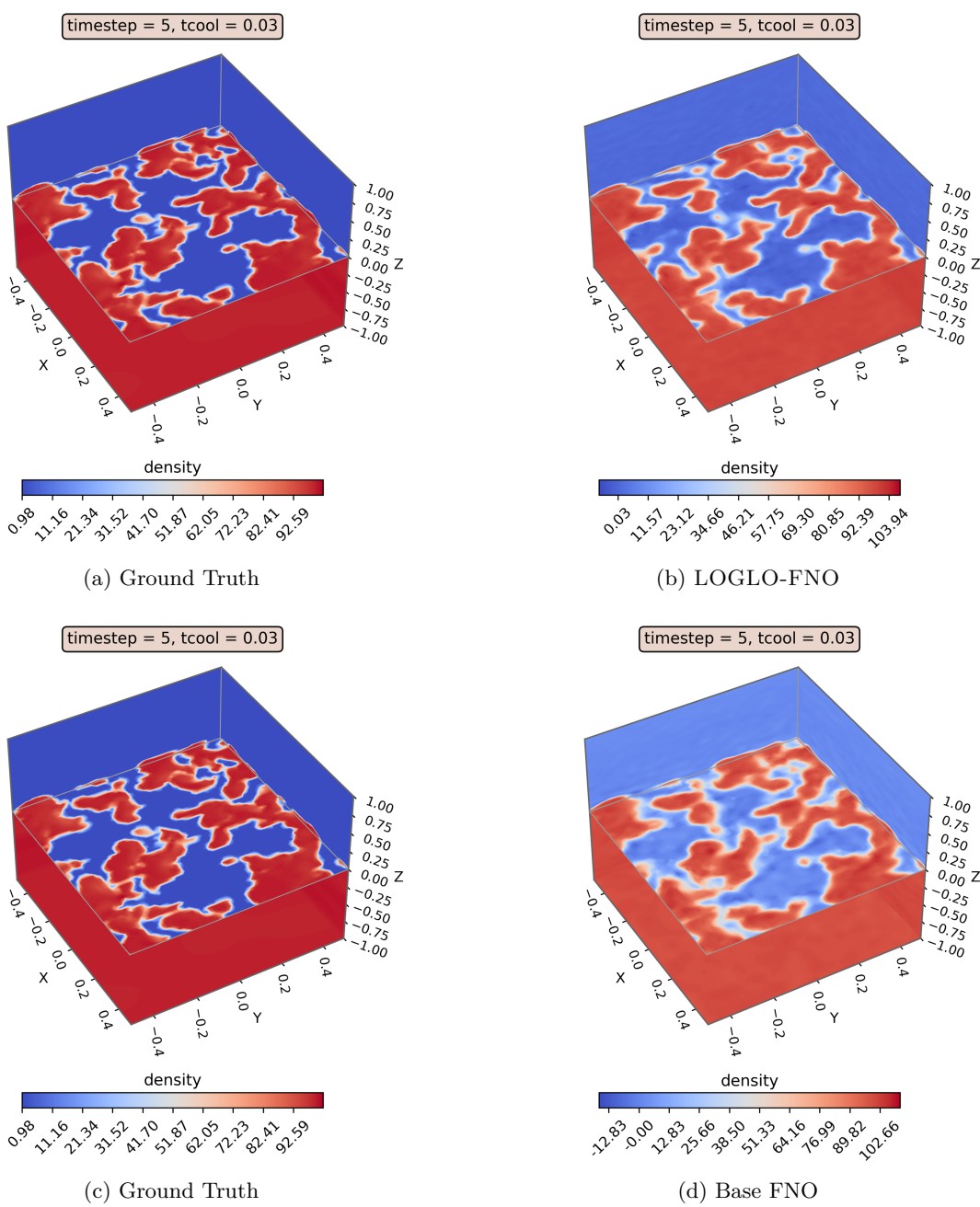

Figure 24: A cross-sectional (XY) slice at the middle of the Z-axis, embedded in the XY, YZ, and XZ planar view, showing the *density* channel from the test set trajectory (for $t_{cool} = 0.03$) at timestep 5 comparing the ground truth (left), LOGLO-FNO and base FNO predictions (right).

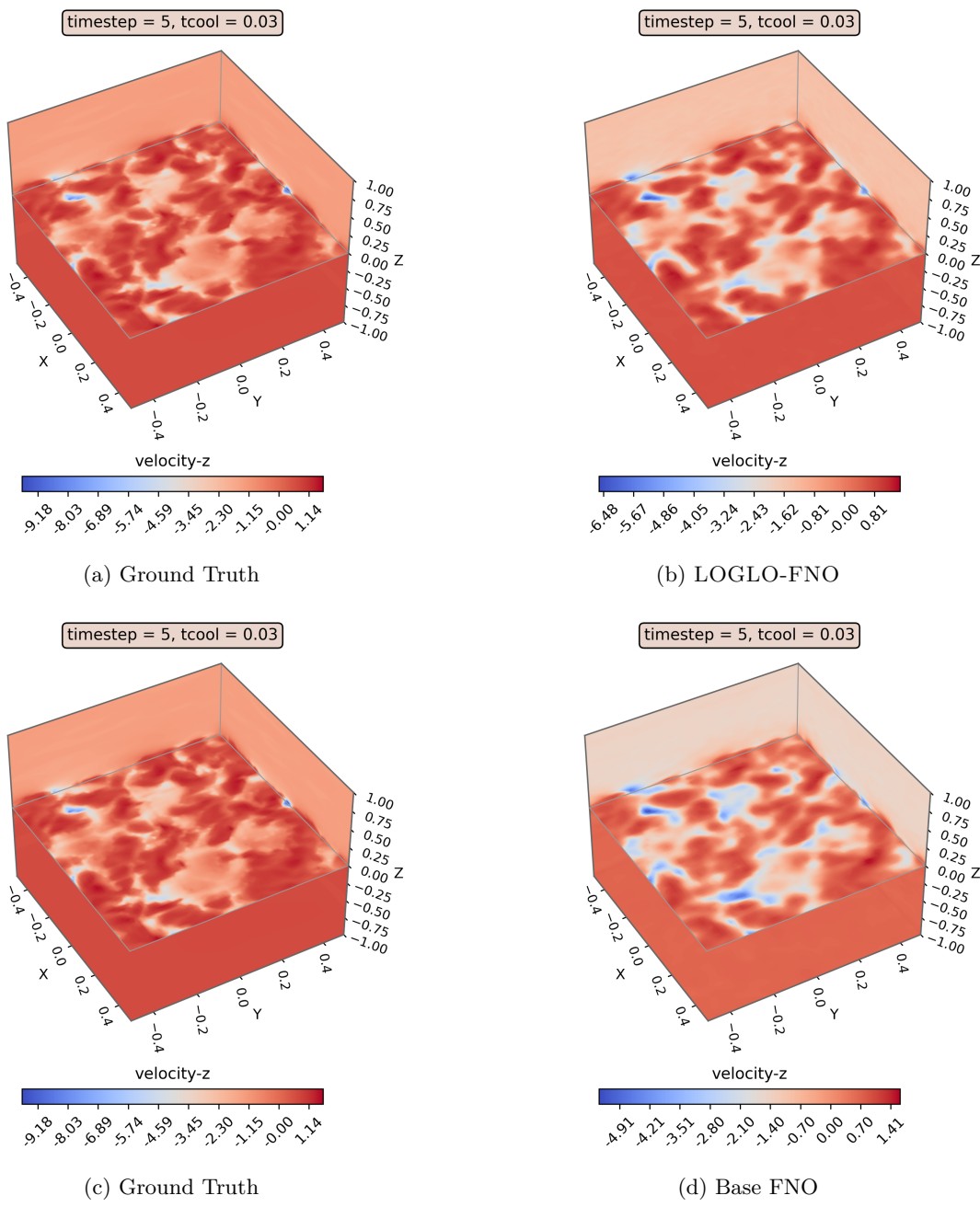

Figure 25: A cross-sectional (XY) slice at the middle of the Z-axis, embedded in the XY, YZ, and XZ planar view, showing the *velocity-z* channel from the test set trajectory (for $t_{cool} = 0.03$) at timestep 5 comparing the ground truth (left), LOGLO-FNO and base FNO predictions (right).

# I    Energy Spectra Visualization and Analysis

**Kolmogorov Flow.**   In this section, we provide a comparative analysis of the energy spectra of predictions of state-of-the-art neural operator baselines versus the reference solution $u$ on the Kolmogorov Flow 2D dataset. The predictions are obtained by autoregressive rollout of the 1-step trained models for the entire duration of the trajectory. In Figures 26, 27, 28, 29, 30, 31, 32, and 33, we plot the energy spectra for two consecutive timesteps at 50 timestep intervals.

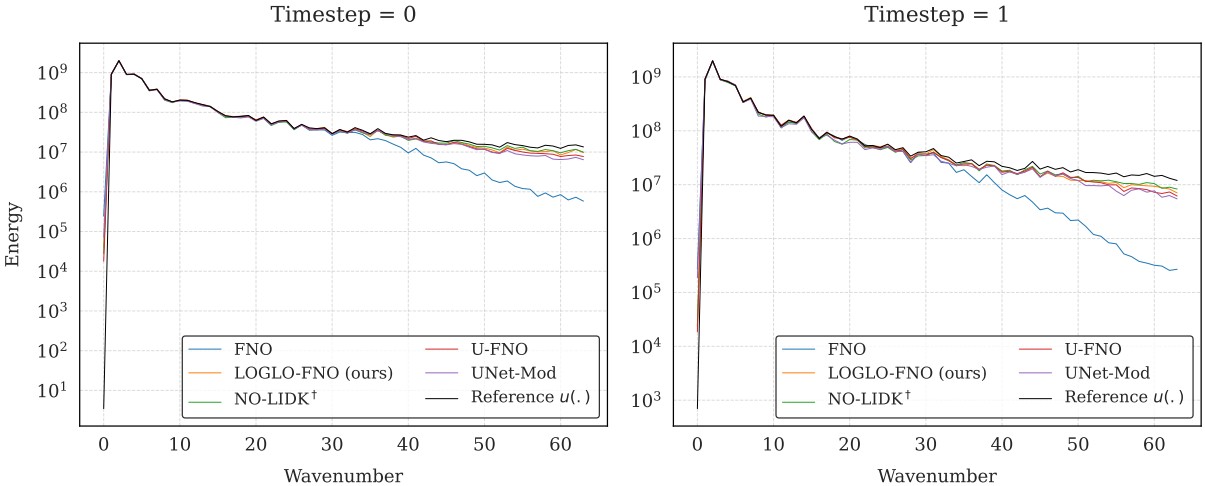

Figure 26: Energy spectra comparison of predictions of state-of-the-art neural operator baselines and our LOGLO-FNO vs. ground truth at two consecutive timesteps on a random sample from the test set of Kolmogorov Flow 2D ($Re = 5k$) (Li et al., 2022a). NO-LIDK$^{\dagger}$ denotes the use of both local integral and differential kernels for local convolutions.

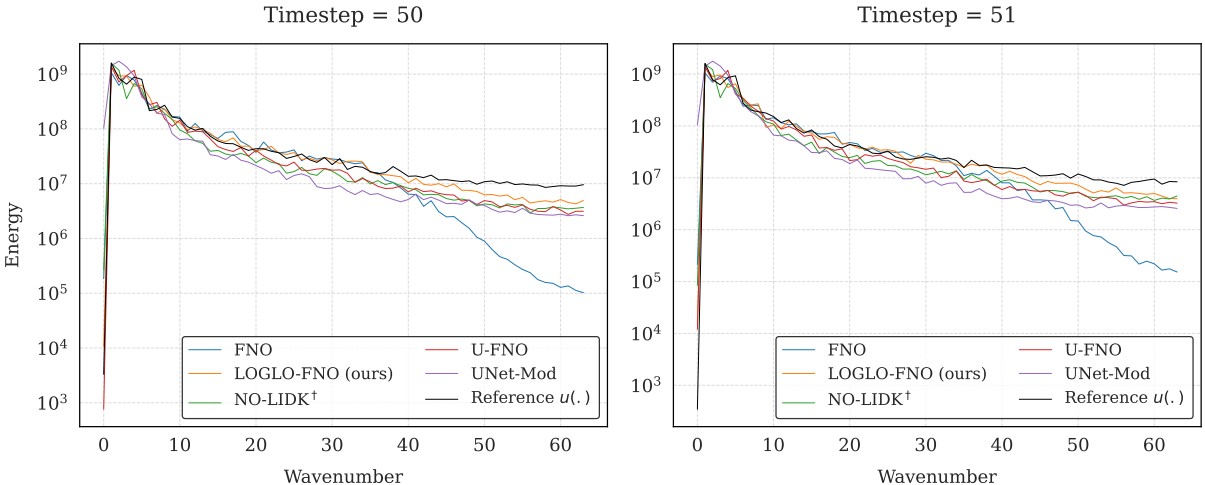

Figure 27: Energy spectra comparison of predictions of state-of-the-art neural operator baselines and our LOGLO-FNO vs. ground truth at two consecutive timesteps on a random sample from the test set of Kolmogorov Flow 2D ($Re = 5k$) (Li et al., 2022a). NO-LIDK$^{\dagger}$ indicates the use of both local integral and differential kernels for local convolutions.

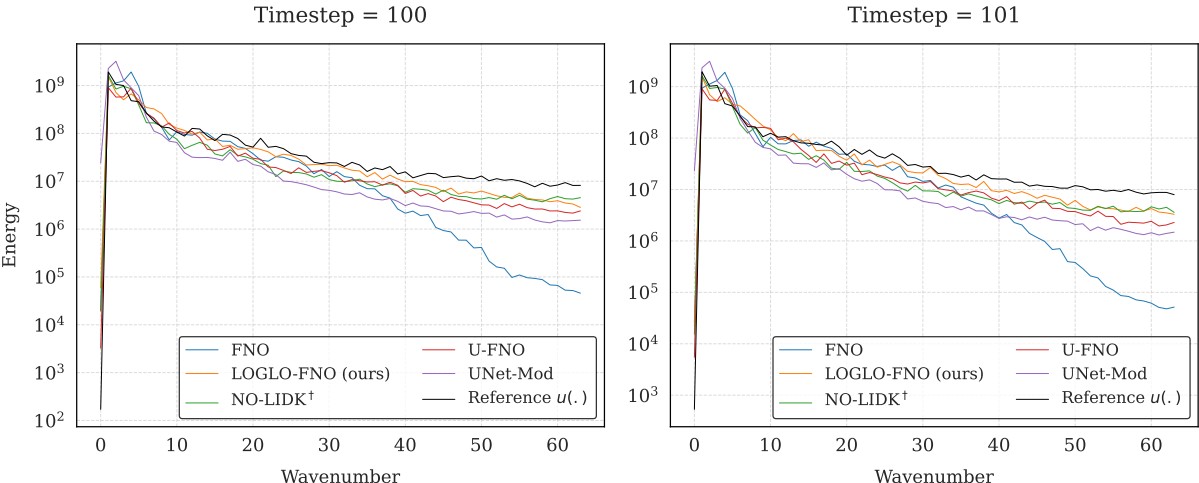

Figure 28: Energy spectra comparison of predictions of state-of-the-art neural operator baselines and our LOGLO-FNO vs. ground truth at two consecutive timesteps on a random sample from the test set of Kolmogorov Flow 2D ($Re = 5k$) (Li et al., 2022a). NO-LIDK[†] denotes the use of both local integral and differential kernels for local convolutions.

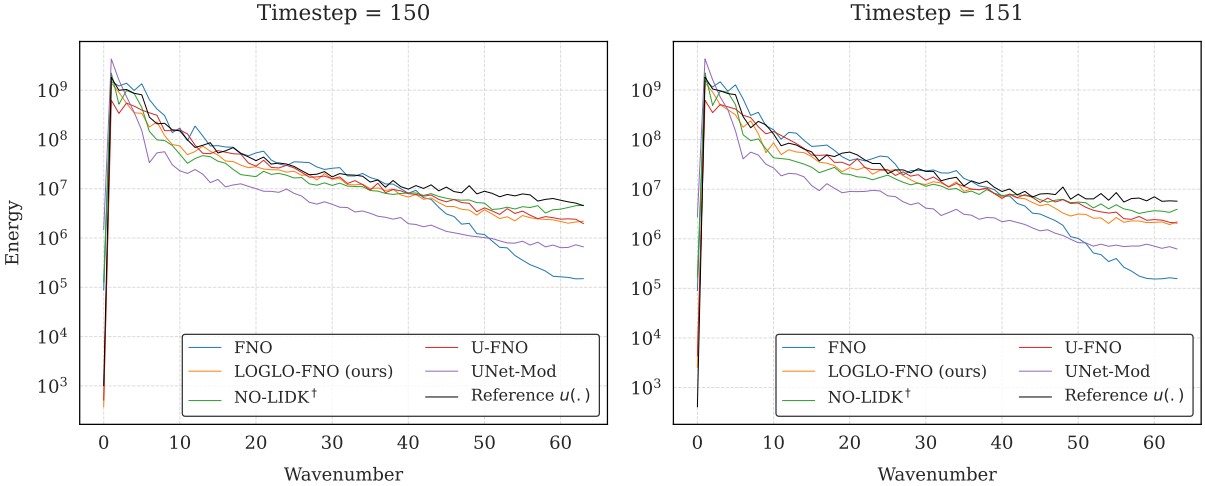

Figure 29: Energy spectra comparison of predictions of state-of-the-art neural operator baselines and our LOGLO-FNO vs. ground truth at two consecutive timesteps on a random sample from the test set of Kolmogorov Flow 2D ($Re = 5k$) (Li et al., 2022a). NO-LIDK[†] indicates the use of both local integral and differential kernels for local convolutions.

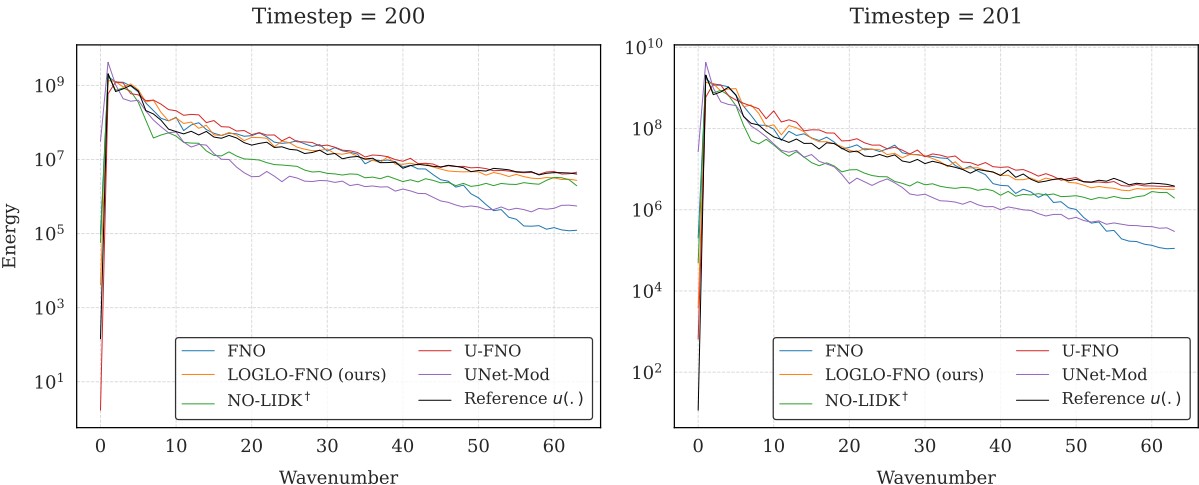

Figure 30: Energy spectra comparison of predictions of state-of-the-art neural operator baselines and our LOGLO-FNO vs. ground truth at two consecutive timesteps on a random sample from the test set of Kolmogorov Flow 2D ($Re = 5k$) (Li et al., 2022a). NO-LIDK$^\dagger$ represents the use of both local integral and differential kernels for local convolutions.

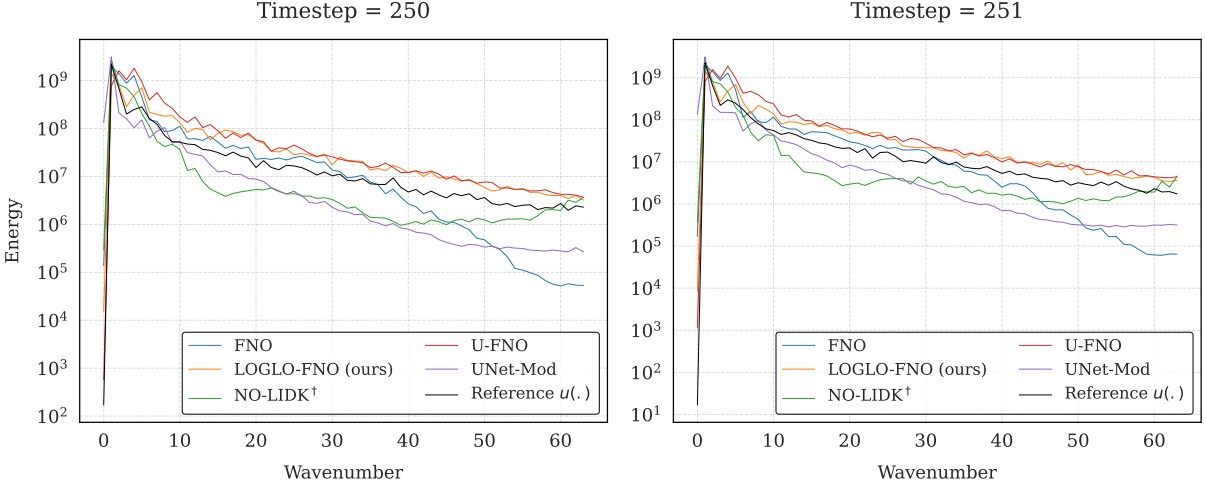

Figure 31: Energy spectra comparison of predictions of state-of-the-art neural operator baselines and our LOGLO-FNO vs. ground truth at two consecutive timesteps on a random sample from the test set of Kolmogorov Flow 2D ($Re = 5k$) (Li et al., 2022a). NO-LIDK$^\dagger$ indicates the use of both local integral and differential kernels for local convolutions.

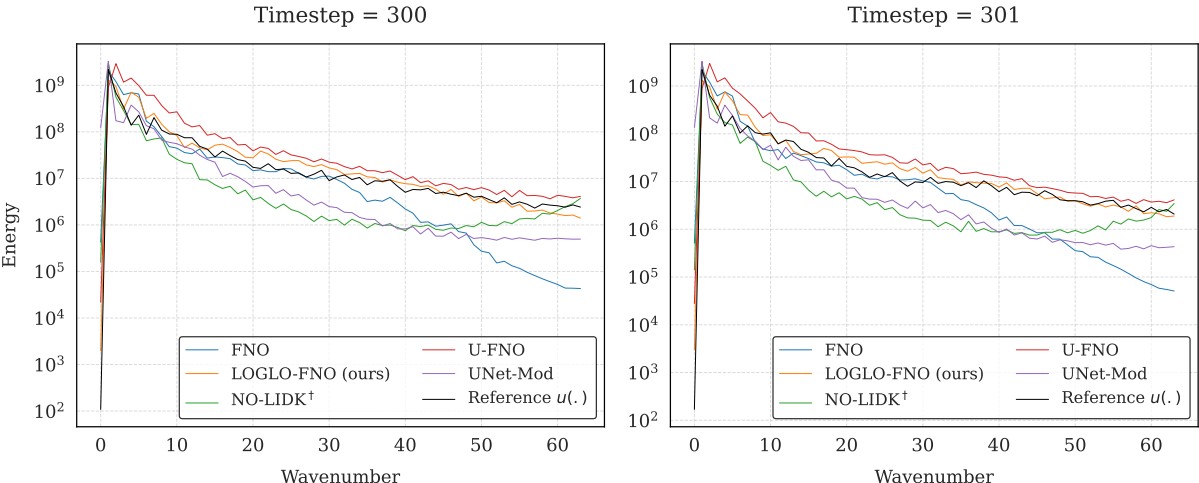

Figure 32: Energy spectra comparison of predictions of state-of-the-art neural operator baselines and our LOGLO-FNO vs. ground truth at two consecutive timesteps on a random sample from the test set of Kolmogorov Flow 2D ($Re = 5k$) (Li et al., 2022a). NO-LIDK$^{\dagger}$ denotes the use of both local integral and differential kernels for local convolutions.

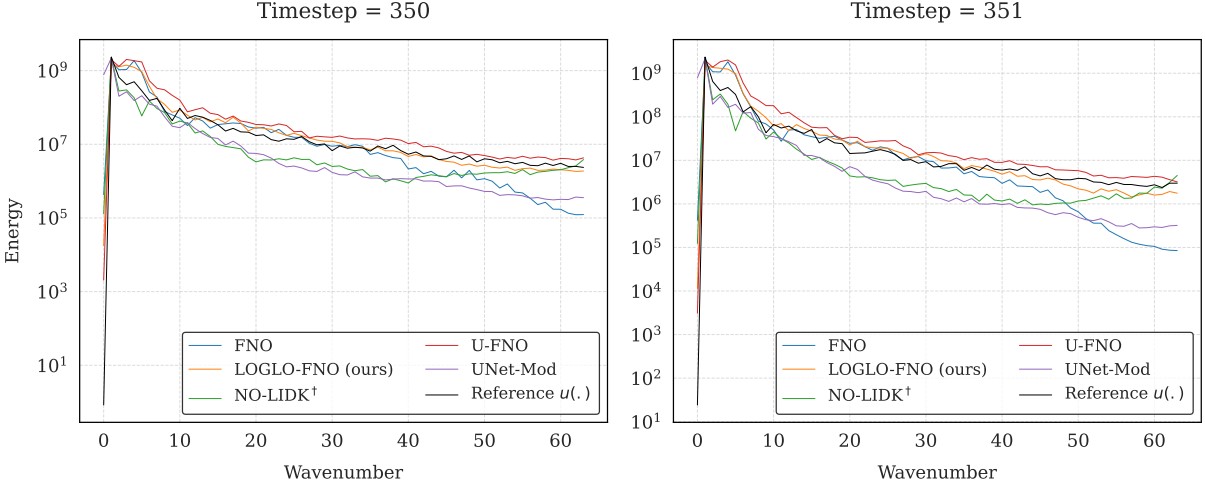

Figure 33: Energy spectra comparison of predictions of state-of-the-art neural operator baselines and our LOGLO-FNO vs. ground truth at two consecutive timesteps on a random sample from the test set of Kolmogorov Flow 2D ($Re = 5k$) (Li et al., 2022a). NO-LIDK$^{\dagger}$ indicates the use of both local integral and differential kernels for local convolutions.

# J  Analysis of Correlation with Ground Truth on AR Rollout

## J.1  Pearson's Correlation of Sample Trajectories with Ground Truth

In this section, we first describe the steps for Pearson's correlation coefficient computation and then provide visualizations of Pearson's correlation of the baselines and our proposed LOGLO-FNO model predictions with respect to the reference solution on randomly sampled trajectories from the test set of Kolmogorov Flow 2D ($Re = 5k$) (Li et al., 2022a).

Let $\mathbf{X} \in \mathbb{R}^{N_b \times N_t \times N_x \times N_y}$ and $\mathbf{Y} \in \mathbb{R}^{N_b \times N_t \times N_x \times N_y}$ represent the predictions from a model and the ground truth, respectively, where:

- $N_b$ is the batch size and $N_t$ is the sequence length or the number of time frames,

- $N_x$ and $N_y$ are the resolutions of spatial dimensions (e.g., width and height for 2D data),

We compute the *Pearson correlation coefficient* between $\mathbf{X}$ and $\mathbf{Y}$ in a vectorized manner as follows.

**Step 1: Reshape Tensors.**  The tensors $\mathbf{X}$ and $\mathbf{Y}$ are reshaped by flattening the spatial dimensions $N_x$ and $N_y$ into a single dimension $S = N_x \times N_y$.

$$\mathbf{X} = \mathbf{X}_{\text{reshape}} \in \mathbb{R}^{N_b \times N_t \times S}, \quad \mathbf{Y} = \mathbf{Y}_{\text{reshape}} \in \mathbb{R}^{N_b \times N_t \times S}.$$

**Step 2: Compute Mean.**  The mean values of $\mathbf{X}$ (predictions) and $\mathbf{Y}$ (ground truth) along the spatial dimension $S$ are computed as,

$$\boldsymbol{\mu}_{\mathbf{X}} = \frac{1}{S} \sum_{i=1}^{S} \mathbf{X}_{:,:,i}, \qquad \boldsymbol{\mu}_{\mathbf{Y}} = \frac{1}{S} \sum_{i=1}^{S} \mathbf{Y}_{:,:,i},$$

where:

- $\boldsymbol{\mu}_{\mathbf{X}} \in \mathbb{R}^{N_b \times N_t}$ and $\boldsymbol{\mu}_{\mathbf{Y}} \in \mathbb{R}^{N_b \times N_t}$ are the mean tensors for $\mathbf{X}$ and $\mathbf{Y}$, respectively,

- The colon notation $:,:,i$ indicates that the operation is performed over the spatial dimension $S$ for each sample in the batch $N_b$ and for each time step in $N_t$.

**Step 3: Compute Standard Deviation.**  The standard deviations of $\mathbf{X}$ (predictions) and $\mathbf{Y}$ (ground truth) along the spatial dimension $S$ are computed as,

$$\boldsymbol{\sigma}_{\mathbf{X}} = \sqrt{\frac{1}{S} \sum_{i=1}^{S} (\mathbf{X}_{:,:,i} - \boldsymbol{\mu}_{\mathbf{X}})^2}, \qquad \boldsymbol{\sigma}_{\mathbf{Y}} = \sqrt{\frac{1}{S} \sum_{i=1}^{S} (\mathbf{Y}_{:,:,i} - \boldsymbol{\mu}_{\mathbf{Y}})^2},$$

where $\boldsymbol{\sigma}_{\mathbf{X}} \in \mathbb{R}^{N_b \times N_t}$ and $\boldsymbol{\sigma}_{\mathbf{Y}} \in \mathbb{R}^{N_b \times N_t}$ are the standard deviations for $\mathbf{X}$ and $\mathbf{Y}$, respectively.

**Step 4: Compute Covariance.**  The covariance between $\mathbf{X}$ (predictions) and $\mathbf{Y}$ (ground truth) can then be computed as,

$$\text{cov}(\mathbf{X}, \mathbf{Y}) = \frac{1}{S} \sum_{i=1}^{S} (\mathbf{X}_{:,:,i} - \boldsymbol{\mu}_{\mathbf{X}}) \odot (\mathbf{Y}_{:,:,i} - \boldsymbol{\mu}_{\mathbf{Y}}),$$

where $\text{cov}(\mathbf{X}, \mathbf{Y}) \in \mathbb{R}^{N_b \times N_t}$ is the covariance tensor and $\odot$ denotes element-wise multiplication.

**Step 5: Compute Pearson's Correlation Coefficient.** The Pearson's correlation coefficient for each sample in the batch $N_b$ and time step $N_t$ is computed using,

$$\text{corr}(\mathbf{X}, \mathbf{Y}) = \frac{\text{cov}(\mathbf{X}, \mathbf{Y})}{\boldsymbol{\sigma}_{\mathbf{X}} \odot \boldsymbol{\sigma}_{\mathbf{Y}}}$$

To ensure numerical stability, the denominator is clamped to a small positive value $\epsilon$ (for instance, set $\epsilon =$ torch.finfo(torch.float32).tiny ):

$$\text{corr}(\mathbf{X}, \mathbf{Y}) = \frac{\text{cov}(\mathbf{X}, \mathbf{Y})}{\max(\boldsymbol{\sigma}_{\mathbf{X}} \odot \boldsymbol{\sigma}_{\mathbf{Y}}, \epsilon)}$$

where $corr(\mathbf{X}, \mathbf{Y}) \in \mathbb{R}^{N_b \times N_t}$ is the output tensor containing the Pearson's correlation coefficients for each sample in the batch and time step.

**Interpretation.** Pearson's correlation coefficient measures the linear relationship between the model predictions $\mathbf{X}$ and the ground truth solution $\mathbf{Y}$. Therefore, a value of

- 1 indicates a perfect positive linear relationship,

- $-1$ indicates a perfect negative linear relationship, and

- 0 indicates no linear relationship.

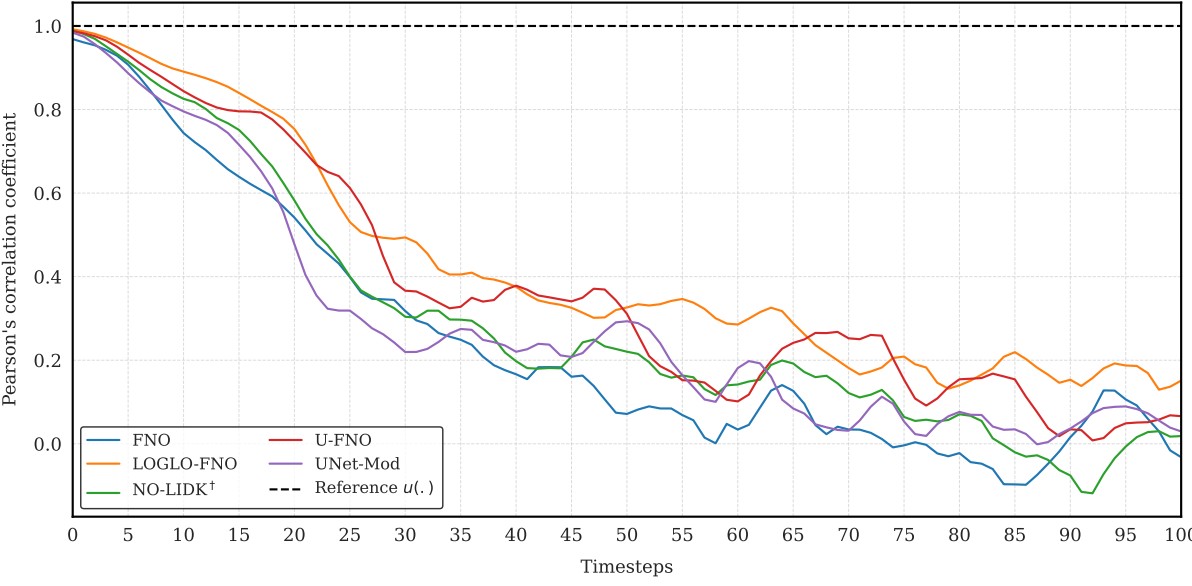

Figure 34: Comparison of Pearson's correlation of predictions of state-of-the-art neural operator baselines and our LOGLO-FNO vs. ground truth on a random sample from the test set of Kolmogorov Flow 2D ($Re = 5k$) (Li et al., 2022a). NO-LIDK$^{\dagger}$ indicates the use of both local integral and differential kernels for local convolutions.

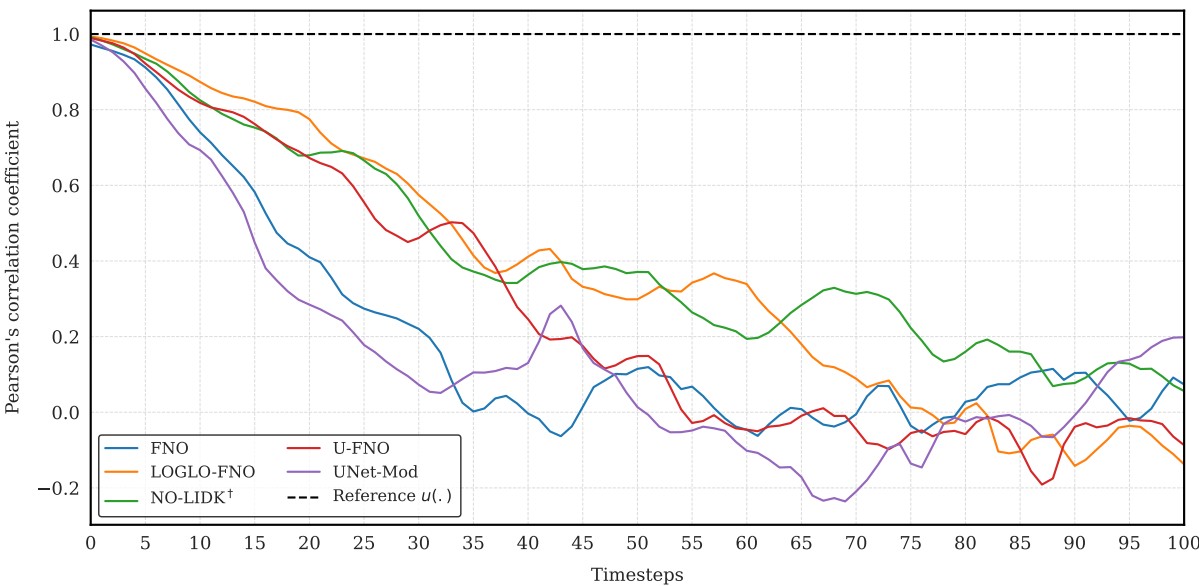

Figure 35: Comparison of Pearson's correlation of predictions of state-of-the-art neural operator baselines and our LOGLO-FNO vs. ground truth on a different random sample from the test set of Kolmogorov Flow 2D ($Re = 5k$) (Li et al., 2022a). NO-LIDK$^{\dagger}$ indicates the use of both local integral and differential kernels for local convolutions.

# K   Ablation Studies

Through the following ablation studies, we attempt to ascertain the impact and importance of each of the introduced components (i.e., local spectral convolution branch and high-frequency propagation module) as well as the patch size sensitivity, high-frequency adaptive Gaussian noise, and radially binned frequency-aware loss term.

## K.1   LOGLO-FNO Devoid of Radially Binned Frequency Loss ($\mathcal{C}_{freq}$)

We conduct an ablation study to assess the relative importance of the proposed radially binned spectral energy of errors as a frequency-aware loss term compared to the architectural modifications by the introduction of layers for local spectral convolution and high-frequency propagation. Therefore, we train LOGLO-FNO by retaining only the local branch and high-frequency propagation module in parallel to the global branch. By setting $\mathcal{C}_{freq} = 0$, the cost function $\mathcal{C}$ for model training then reduces to the plain MSE loss.

$$\theta^* = \arg\min_{\theta} \sum_{n=1}^{N} \sum_{t=1}^{T-1} \mathcal{C}_{MSE}(\mathcal{N}_\theta(u^t), u^{t+1})$$

The results of this study on the Turbulent Radiative Layer 3D dataset are provided in Table 7 on row 3.

## K.2   LOGLO-FNO Devoid of HFP Module and Radially Binned Frequency Loss (HFP, $\mathcal{C}_{freq}$)

In this ablation setup, we remove the high-frequency propagation module and keep only the local branch. The radially binned frequency loss term is also disabled by setting $\mathcal{C}_{freq} = 0$. Therefore, the LOGLO-FNO architecture comprises only the local and global branches in the Fourier layers, and is trained with plain MSE.

The results of this ablation choice on Turbulent Radiative Layer 3D are provided in Table 7 on row 4.

## K.3   LOGLO-FNO Devoid of High Frequency Propagation Module (HFP)

This ablation setup is similar to the previous study in §K.2, except that we now enable the radially binned frequency loss in the training objective to isolate the effect of removing the HFP module. The 1-step evaluation results for this model setup on Turbulent Radiative Layer 3D are placed in the last row of Table 7.

Table 15: 1-step evaluation of LOGLO-FNO on the test set of Turbulent Radiative Mixing Layer 3D dataset (Fielding et al., 2020; Ohana et al., 2024). Base FNO uses a spectral filter size of 18 and 48 hidden channels. LOGLO-FNO uses 16 and $(16, 16, 17)^*$ Fourier modes in the global and local branches, respectively, whereas the width is set to 52. ($^*$This is because the Turbulent Radiative Layer 3D dataset has $2\times$ more sampling points on the z-axis relative to the x- and y-axes resolutions – $(128 \times 128 \times 256)$.)

| Model | RMSE ($\downarrow$) | nRMSE | bRMSE | cRMSE | fRMSE(L) | fRMSE(M) | fRMSE(H) | vRMSE ($\downarrow$) | MaxError ($\downarrow$) |
|---|---|---|---|---|---|---|---|---|---|
| | | | | **1-step Evaluation** | | | | | |
| Base FNO | $2.76 \cdot 10^{-1}$ | $2.97 \cdot 10^{-1}$ | $6.83 \cdot 10^{-1}$ | $2.3 \cdot 10^{-2}$ | $5.17 \cdot 10^{-2}$ | $4.45 \cdot 10^{-2}$ | $2.76 \cdot 10^{-2}$ | $3.09 \cdot 10^{-1}$ | $1.11 \cdot 10^{1}$ |
| LOGLO-FNO (+freq loss, +HFP) | $\mathbf{2.42 \cdot 10^{-1}}$ | $\mathbf{2.58 \cdot 10^{-1}}$ | $\mathbf{6.19 \cdot 10^{-1}}$ | $\mathbf{1.75 \cdot 10^{-2}}$ | $\mathbf{4.1 \cdot 10^{-2}}$ | $\mathbf{3.62 \cdot 10^{-2}}$ | $\mathbf{2.45 \cdot 10^{-2}}$ | $\mathbf{2.68 \cdot 10^{-1}}$ | $\mathbf{1.06 \cdot 10^{1}}$ |
| LOGLO-FNO (-freq loss, +HFP) | $2.57 \cdot 10^{-1}$ | $2.75 \cdot 10^{-1}$ | $6.51 \cdot 10^{-1}$ | $1.85 \cdot 10^{-2}$ | $4.4 \cdot 10^{-2}$ | $4.09 \cdot 10^{-2}$ | $2.63 \cdot 10^{-2}$ | $2.86 \cdot 10^{-1}$ | $1.09 \cdot 10^{1}$ |
| LOGLO-FNO (-freq loss, -HFP) | $2.58 \cdot 10^{-1}$ | $2.76 \cdot 10^{-1}$ | $6.55 \cdot 10^{-1}$ | $1.83 \cdot 10^{-2}$ | $4.4 \cdot 10^{-2}$ | $4.13 \cdot 10^{-2}$ | $2.64 \cdot 10^{-2}$ | $2.87 \cdot 10^{-1}$ | $1.1 \cdot 10^{1}$ |
| LOGLO-FNO (+freq loss, -HFP) | $2.47 \cdot 10^{-1}$ | $2.66 \cdot 10^{-1}$ | $6.32 \cdot 10^{-1}$ | $2.03 \cdot 10^{-2}$ | $4.53 \cdot 10^{-2}$ | $3.76 \cdot 10^{-2}$ | $2.48 \cdot 10^{-2}$ | $2.76 \cdot 10^{-1}$ | $1.07 \cdot 10^{1}$ |

As we observe from Table 7, our model achieves the lowest errors across all metrics when the local branch is combined with both the radially binned frequency loss and the HFP module.

## K.4 Patch-Size Sensitivity of LOGLO-FNO

In this section, we first provide the results of an ablation study investigating the sensitivity of LOGLO-FNO on the patch sizes for the local branch through an experiment on the Turbulent Radiative Layer 3D PDE problem. Across the tested patch size variations ($p_{\text{size}} = \{4, 8, 16\}$), only the patch size values are altered while all other hyperparameters, including the learning rate and random seed, remain the same during training.

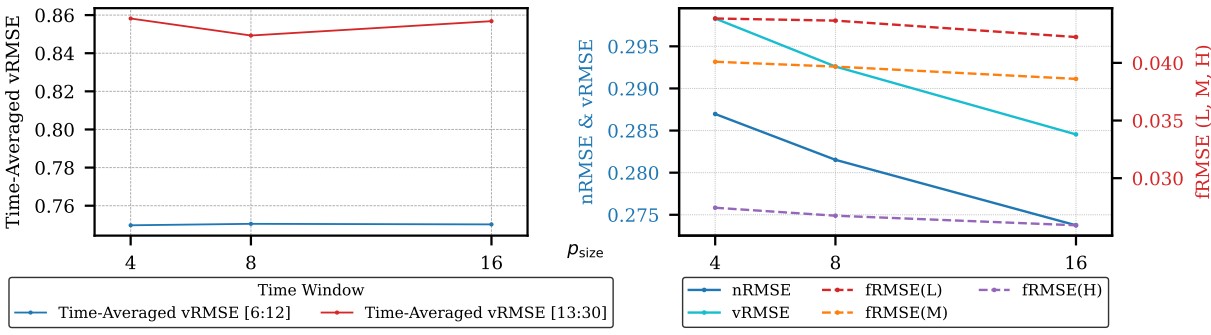

Figure 36: Effect of patch sizes on the Time-Averaged vRMSE and 1-step spatial and spectral evaluation metrics on the Turbulent Radiative Layer 3D dataset.

In Figure 36, we visualize the Time-Averaged vRMSE and 1-step xRMSE evaluation scores for patch sizes 4, 8, and 16. While we observe a trend of minor improvement in 1-step errors with the increase of patch sizes, the Time-Averaged autoregressive errors do not exhibit this clear trend. We note from Figure 36 that an increase in patch size directly results in an increased number of trainable parameters (and by virtue of it, increases model expressivity) since we use the full set of Fourier modes in the local branch. Therefore, as a trade-off between errors and training time or memory usage, we use a patch size of 16.

As an additional case study on patch size sensitivity analysis, we conduct experiments on the Compressible Euler four-quadrant Riemann problem 2D from Poseidon (Herde et al., 2024), and provide the 1-step and autoregressive error evolution of overall (nRMSE) and frequency-decomposed (fRMSE) evaluations over an increasing number of timesteps for four different patch size configurations of LOGLO-FNO in Figures 37 and 38, respectively.

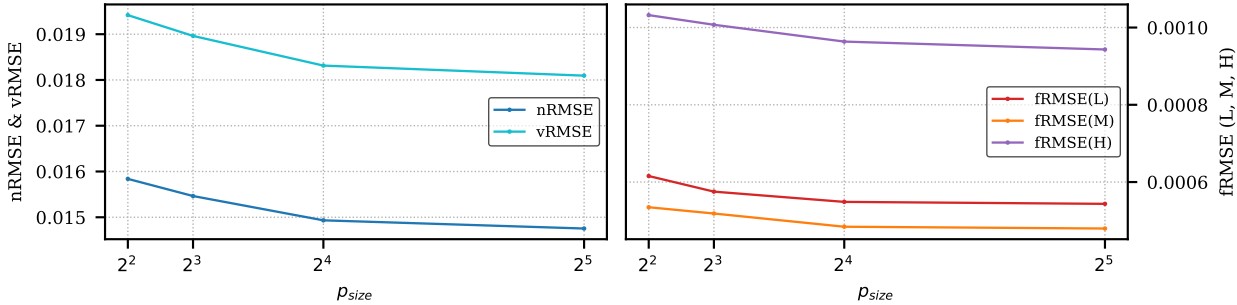

Figure 37: Effect of patch sizes on the 1-step spatial (left: nRMSE and vRMSE) and spectral (right: fRMSE) evaluation metrics on the Compressible Euler four-quadrant Riemann problem 2D (Herde et al., 2024).

Shifting our attention to the rollout evaluations, we note from Figure 38 a trend of error improvements as we increase the patch sizes from 4 to 16. However, a further increase in patch size to 32 does not yield an improvement, but rather detrimental, over long rollout scenarios. Therefore, as was previously found in the

case of the Turbulent Radiative Layer 3D problem, we come to the conclusion that a patch size of 16 is a sufficiently good choice.

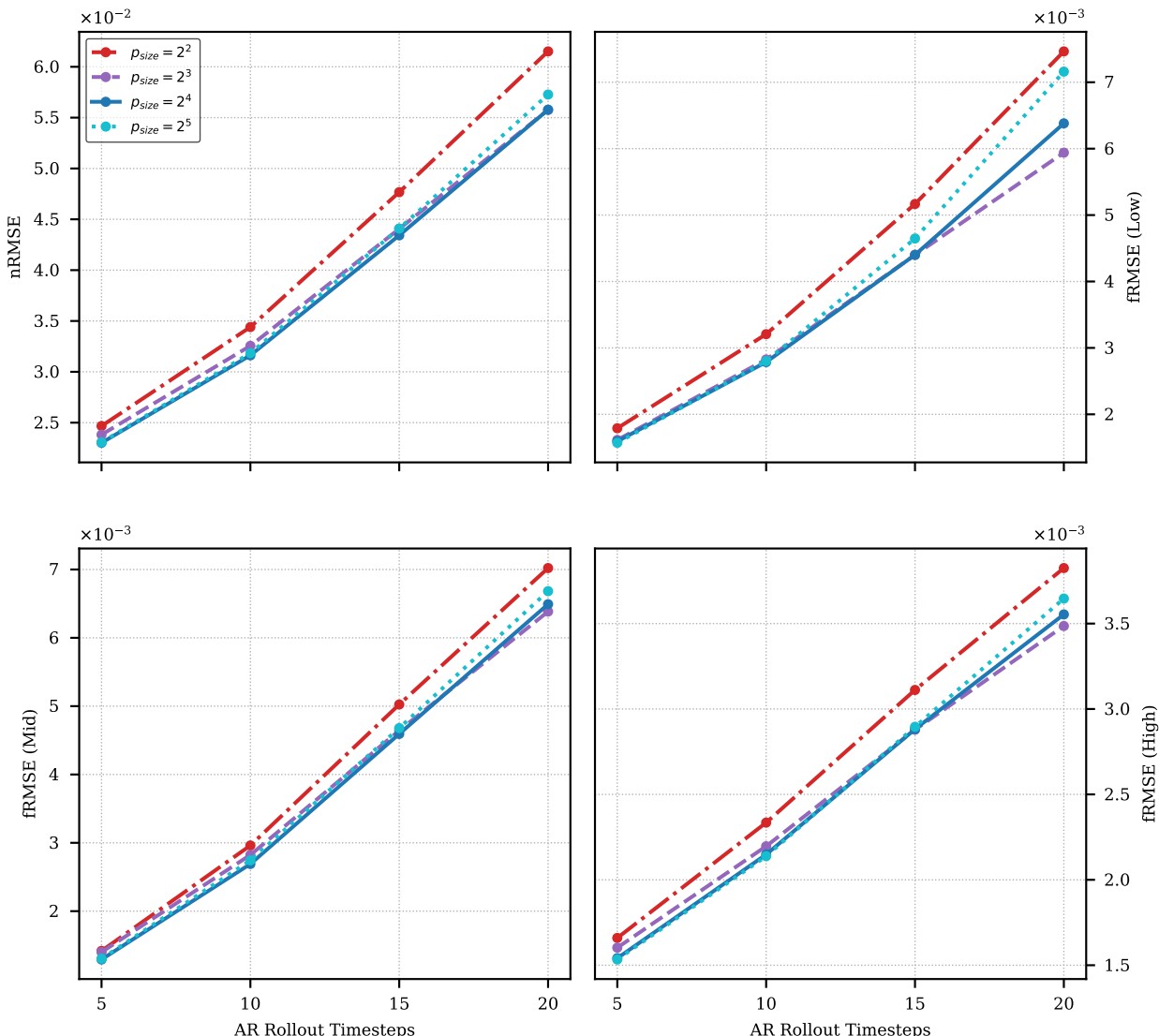

Figure 38: Error accumulation and patch size sensitivity during autoregressive rollouts on the Compressible Euler four-quadrant Riemann problem 2D (Herde et al., 2024).

### K.5 Influence of High-Frequency Adaptive Gaussian Noise on Rollout Stability in LOGLO-FNO

In this section, we conduct an ablation study to investigate the influence of high-frequency adaptive Gaussian noise (HF-AGN) in LOGLO-FNO on the stability of trajectory rollouts for the Kolmogorov Flow 2D (Re5000) dataset. We report the mean and standard deviation values of errors at timestep 400 over three‖ runs in Table 16. The results suggest that HF-AGN is helpful for the stability of longer rollout scenarios.

Table 16: Rollout errors of LOGLO-FNO, with and without adaptive Gaussian noise (HF-AGN) injected during 1-step training, at timestep 400 on the Kolmogorov Flow 2D (Re5000) dataset. The autoregressive evaluation is over three runs.

| Group | RMSE (↓) | nRMSE | bRMSE | cRMSE | fRMSE(L) | fRMSE(M) | fRMSE(H) | MaxError (↓) | MELR (↓) | WLR (↓) |
|---|---|---|---|---|---|---|---|---|---|---|
| LOGLO-FNO (-HF-AGN) | 7.8891 (±1.2168) | 1.4572 (±0.2319) | 7.5518 (±1.0478) | 0.0472 (±0.0614) | 1.4525 (±0.2229) | 0.8524 (±0.2458) | 0.1726 (±0.0227) | 66.2614 (±13.5936) | 0.4811 (±0.0662) | 0.2797 (±0.0699) |
| LOGLO-FNO (+HF-AGN) | **7.1982** (±**0.1837**) | **1.3227** (±**0.0337**) | **7.0794** (±**0.2293**) | **0.0113** (±**0.0050**) | **1.2965** (±**0.0512**) | **0.7362** (±**0.0151**) | **0.1667** (±**0.0012**) | **58.9953** (±**1.7618**) | **0.4061** (±**0.0795**) | **0.2228** (±**0.0070**) |

### K.6 Choice of Radial Bin Cutoff Values for the Radially Binned Frequency Loss in LOGLO-FNO

In order to assess the sensitivity of the cutoff boundaries when demarcating the low, mid, and high frequency regions for the radially binned spectral loss, we conduct an ablation study on the 2D Kolmogorov Flow dataset and provide the 5-step AR errors in Table 17.

We note that the choice iLow=2 and iHigh=10 achieves the lowest spatial (nRMSE) and low-frequency errors and ties with the iLow=4 and iHigh=12 setup on the high-frequency error. However, the latter might be an optimal choice since it exhibits the lowest variance on the spatial (nRMSE), mid-, and high-frequency errors and is more stable over long rollouts.

Table 17: Sensitivity analysis of radial bin cutoffs for the radially binned frequency loss applied for training LOGLO-FNO. The experiments are on the Kolmogorov Flow 2D (Re5000) dataset, and the evaluations are over five independent runs with unique random seeds.

| Cutoff Boundary | Unpenalized Region | Penalized Region | nRMSE (↓) | fRMSE(L) | fRMSE(M) | fRMSE(H) (↓) |
|---|---|---|---|---|---|---|
| iLow=2, iHigh=10 | $r < 2$ | $r \geq 2$ | **0.1917** (± **0.0039**) | **0.0262** (± **0.0008**) | 0.0466 (± 0.0025) | **0.0694** (± 0.0014) |
| iLow=4, iHigh=12 | $r < 4$ | $r \geq 4$ | 0.1931 (± **0.0021**) | 0.0285 (± 0.0025) | **0.0462** (± **0.0020**) | **0.0694** (± **0.0008**) |
| iLow=6, iHigh=15 | $r < 6$ | $r \geq 6$ | 0.1942 (± 0.0034) | 0.0281 (± 0.0010) | 0.0485 (± 0.0025) | 0.0702 (± 0.0014) |

---

‖Of the total four runs we conducted, we exclude a run from the mean computation due to a blow-up in errors for the model without HF-AGN.

# L  Zero-Shot Super-Resolution (ZSSR) Capability and Performance of LOGLO-FNO

In this section, we evaluate the capability and performance of LOGLO-FNO on spatial ZSSR tasks and compare it with the results of NO-LIDK (Liu-Schiaffini et al., 2024) and the baseline FNO model.

Table 18: ZSSR evaluation of LOGLO-FNO on the test set of Kolmogorov Flow 2D dataset (Li et al., 2022b). Base FNO and NO-LIDK$^*$ use a spectral filter size of 12 and 20 hidden channels. NO-LIDK$^*$ denotes using only localized integral kernel and NO-LIDK$^\dagger$ means employing both disco and diff layers. LOGLO-FNO uses 12 and (16, 9) Fourier modes in the global and local branches, respectively, whereas the width is set to 20. Training spatial resolution is $(64 \times 64)$ and denoted as $1\times$, whereas the ZSSR test resolution is $(128 \times 128)$ and denoted as $2\times$. We boldface the lowest errors and underline the value when the $2^{nd}$ best model is LOGLO-FNO.

| Model | Resolution | nRMSE ($\downarrow$) | fRMSE(L) | fRMSE(M) | fRMSE(H) | MELR ($\downarrow$) | WLR ($\downarrow$) |
|---|---|---|---|---|---|---|---|
| | Train: $1\times$ Test : $2\times$ | | | **1-step Evaluation** | | | |
| Base FNO | $1\times$ | $2.99 \cdot 10^{-1}$ | $4.71 \cdot 10^{-2}$ | $8.3 \cdot 10^{-2}$ | $1.59 \cdot 10^{-1}$ | $6.41 \cdot 10^{-1}$ | $1.2 \cdot 10^{-1}$ |
| | $2\times$ | $2.93 \cdot 10^{-1}$ | $\mathbf{3.14 \cdot 10^{-2}}$ | $7.38 \cdot 10^{-2}$ | $1.16 \cdot 10^{-1}$ | $9.61 \cdot 10^{-1}$ | $1.43 \cdot 10^{-1}$ |
| NO-LIDK$^*$ | $1\times$ | $2.79 \cdot 10^{-1}$ | $4.65 \cdot 10^{-2}$ | $8.88 \cdot 10^{-2}$ | $1.41 \cdot 10^{-1}$ | $5.89 \cdot 10^{-1}$ | $1.04 \cdot 10^{-1}$ |
| | $2\times$ | $3.71 \cdot 10^{-1}$ | $5.16 \cdot 10^{-2}$ | $7.87 \cdot 10^{-2}$ | $1.12 \cdot 10^{-1}$ | $9.21 \cdot 10^{-1}$ | $1.45 \cdot 10^{-1}$ |
| NO-LIDK$^\dagger$ | $1\times$ | $2.56 \cdot 10^{-1}$ | $4.61 \cdot 10^{-2}$ | $7.77 \cdot 10^{-2}$ | $1.28 \cdot 10^{-1}$ | $2.32 \cdot 10^{-1}$ | $5.89 \cdot 10^{-2}$ |
| | $2\times$ | $4.35 \cdot 10^{-1}$ | $1.11 \cdot 10^{-1}$ | $1.75 \cdot 10^{-1}$ | $1.39 \cdot 10^{-1}$ | $2.6 \cdot 10^{-1}$ | $6.6 \cdot 10^{-2}$ |
| LOGLO-FNO | $1\times$ | $\mathbf{2.32 \cdot 10^{-1}}$ | $\mathbf{4.57 \cdot 10^{-2}}$ | $\mathbf{7.35 \cdot 10^{-2}}$ | $\mathbf{1.22 \cdot 10^{-1}}$ | $\mathbf{2.12 \cdot 10^{-1}}$ | $\mathbf{4.68 \cdot 10^{-2}}$ |
| | $2\times$ | $\mathbf{2.9 \cdot 10^{-1}}$ | $\underline{3.19 \cdot 10^{-2}}$ | $\mathbf{6.76 \cdot 10^{-2}}$ | $\mathbf{1.05 \cdot 10^{-1}}$ | $\mathbf{2.32 \cdot 10^{-1}}$ | $\mathbf{6.34 \cdot 10^{-2}}$ |
| | | | | **5-step Autoregressive Evaluation** | | | |
| Base FNO | $1\times$ | $4.54 \cdot 10^{-1}$ | $1.05 \cdot 10^{-1}$ | $1.96 \cdot 10^{-1}$ | $2.44 \cdot 10^{-1}$ | $7.03 \cdot 10^{-1}$ | $1.68 \cdot 10^{-1}$ |
| | $2\times$ | $4.4 \cdot 10^{-1}$ | $9.17 \cdot 10^{-2}$ | $1.65 \cdot 10^{-1}$ | $1.55 \cdot 10^{-1}$ | $1.13 \cdot 10^{0}$ | $1.95 \cdot 10^{-1}$ |
| NO-LIDK$^*$ | $1\times$ | $4.32 \cdot 10^{-1}$ | $1.05 \cdot 10^{-1}$ | $2.03 \cdot 10^{-1}$ | $2.27 \cdot 10^{-1}$ | $8.58 \cdot 10^{-1}$ | $1.86 \cdot 10^{-1}$ |
| | $2\times$ | $4.65 \cdot 10^{-1}$ | $9.89 \cdot 10^{-2}$ | $1.97 \cdot 10^{-1}$ | $1.58 \cdot 10^{-1}$ | $1.28 \cdot 10^{0}$ | $2.59 \cdot 10^{-1}$ |
| NO-LIDK$^\dagger$ | $1\times$ | $3.85 \cdot 10^{-1}$ | $9.15 \cdot 10^{-2}$ | $1.69 \cdot 10^{-1}$ | $2.16 \cdot 10^{-1}$ | $3.75 \cdot 10^{-1}$ | $1.06 \cdot 10^{-1}$ |
| | $2\times$ | $7.58 \cdot 10^{-1}$ | $2.59 \cdot 10^{-1}$ | $5.06 \cdot 10^{-1}$ | $2.11 \cdot 10^{-1}$ | $\mathbf{4.38 \cdot 10^{-1}}$ | $1.78 \cdot 10^{-1}$ |
| LOGLO-FNO | $1\times$ | $\mathbf{3.78 \cdot 10^{-1}}$ | $\mathbf{8.23 \cdot 10^{-2}}$ | $\mathbf{1.55 \cdot 10^{-1}}$ | $\mathbf{2.02 \cdot 10^{-1}}$ | $\mathbf{3.56 \cdot 10^{-1}}$ | $\mathbf{9.46 \cdot 10^{-2}}$ |
| | $2\times$ | $\mathbf{4.29 \cdot 10^{-1}}$ | $\mathbf{7.95 \cdot 10^{-2}}$ | $\mathbf{1.47 \cdot 10^{-1}}$ | $\mathbf{1.51 \cdot 10^{-1}}$ | $\underline{4.39 \cdot 10^{-1}}$ | $\mathbf{1.4 \cdot 10^{-1}}$ |

We provide the spatial ZSSR results of baseline FNO, NO-LIDK variants, and our proposed LOGLO-FNO on the test set of the 2D Kolmogorov Flow dataset (Li et al., 2022a) in Table 18. The original spatial resolution of the dataset is $128 \times 128$. Therefore, we perform an even rate $2\times$ subsampling to obtain training data at a resolution of $64 \times 64$, denoted as $1\times$ in the table. Naturally, the ZSSR task is then to evaluate the capability of the models for the next higher resolution of $128 \times 128$, for which we have the ground truth. We observe that baseline FNO exhibits superior spatial ZSSR capabilities, except on the spectral metrics (MELR & WLR), on which it incurs degradation. LOGLO-FNO, on the other hand, incurs degradation in spatial and spectral errors. However, it exhibits superior improvements on the frequency domain errors, outperforming base FNO on all metrics, but fRMSE (Low). Most importantly, we find that LOGLO-FNO is better than the NO-LIDK variants on all metrics. Subsequently, we perform ZSSR evaluation on the 5-step autoregressive rollouts, and find a similar trend.

Table 19: Spatial ZSSR evaluation of LOGLO-FNO on the test set of CNS Turb 2D dataset (Takamoto et al., 2022). Base FNO uses a spectral filter size of 40 and 65 hidden channels. LOGLO-FNO uses 40 and (16, 9) Fourier modes in the global and local branches, respectively, whereas the width is set to 65. NO-LIDK* uses 40 modes and 65 hidden channels and a radius cutoff value of 0.00390625. NO-LIDK* denotes using only localized integral kernel and NO-LIDK$^\dagger$ means employing both disco and diff layers. Training spatial resolution is $(256 \times 256)$, being noted as $1\times$, whereas the ZSSR test resolution is $(512 \times 512)$ and denoted as $2\times$. We boldface the lowest errors and underline the value when the $2^{nd}$ best model is LOGLO-FNO.

| Model | Resolution | nRMSE (↓) | fRMSE(L) | fRMSE(M) | fRMSE(H) | MELR (↓) | WLR (↓) |
|---|---|---|---|---|---|---|---|
| | Train: $1\times$ Test : $2\times$ | | | **1-step Evaluation** | | | |
| Base FNO | $1\times$ | $4.82 \cdot 10^{-2}$ | $1.28 \cdot 10^{-3}$ | $1.78 \cdot 10^{-3}$ | $1.57 \cdot 10^{-3}$ | $2.78 \cdot 10^{-1}$ | $7.8 \cdot 10^{-3}$ |
| | $2\times$ | $4.81 \cdot 10^{-2}$ | $1.26 \cdot 10^{-3}$ | $1.77 \cdot 10^{-3}$ | $9.24 \cdot 10^{-4}$ | $5.5 \cdot 10^{-1}$ | $7.85 \cdot 10^{-3}$ |
| NO-LIDK* | $1\times$ | $6.4 \cdot 10^{-2}$ | $1.5 \cdot 10^{-3}$ | $2.14 \cdot 10^{-3}$ | $1.98 \cdot 10^{-3}$ | $2.98 \cdot 10^{-1}$ | $1.31 \cdot 10^{-2}$ |
| | $2\times$ | $7.61 \cdot 10^{-2}$ | $7.73 \cdot 10^{-3}$ | $2.92 \cdot 10^{-3}$ | $1.17 \cdot 10^{-3}$ | $5.29 \cdot 10^{-1}$ | $2.45 \cdot 10^{-2}$ |
| NO-LIDK$^\dagger$ | $1\times$ | $5.57 \cdot 10^{-2}$ | $1.37 \cdot 10^{-3}$ | $1.89 \cdot 10^{-3}$ | $1.77 \cdot 10^{-3}$ | $2.08 \cdot 10^{-1}$ | $9.54 \cdot 10^{-3}$ |
| | $2\times$ | $8.66 \cdot 10^{-2}$ | $4.8 \cdot 10^{-3}$ | $2.67 \cdot 10^{-3}$ | $1.93 \cdot 10^{-3}$ | $9.0 \cdot 10^{-1}$ | $1.21 \cdot 10^{-2}$ |
| LOGLO-FNO | $1\times$ | $\mathbf{4.34} \cdot 10^{-2}$ | $\mathbf{9.74} \cdot 10^{-4}$ | $\mathbf{1.17} \cdot 10^{-3}$ | $\mathbf{1.41} \cdot 10^{-3}$ | $\mathbf{1.83} \cdot 10^{-1}$ | $\mathbf{5.93} \cdot 10^{-3}$ |
| | $2\times$ | $\mathbf{4.54} \cdot 10^{-2}$ | $\mathbf{1.06} \cdot 10^{-3}$ | $\mathbf{1.27} \cdot 10^{-3}$ | $\mathbf{8.91} \cdot 10^{-4}$ | $3.56 \cdot 10^{-1}$ | $\mathbf{7.75} \cdot 10^{-3}$ |
| | | | | **5-step Autoregressive Evaluation** | | | |
| Base FNO | $1\times$ | $7.78 \cdot 10^{-2}$ | $3.76 \cdot 10^{-3}$ | $5.37 \cdot 10^{-3}$ | $2.4 \cdot 10^{-3}$ | $2.22 \cdot 10^{-1}$ | $9.36 \cdot 10^{-3}$ |
| | $2\times$ | $\mathbf{7.06} \cdot 10^{-2}$ | $3.31 \cdot 10^{-3}$ | $4.77 \cdot 10^{-3}$ | $\mathbf{1.37} \cdot 10^{-3}$ | $5.12 \cdot 10^{-1}$ | $\mathbf{8.1} \cdot 10^{-3}$ |
| NO-LIDK* | $1\times$ | $8.55 \cdot 10^{-2}$ | $4.14 \cdot 10^{-3}$ | $5.92 \cdot 10^{-3}$ | $2.62 \cdot 10^{-3}$ | $2.06 \cdot 10^{-1}$ | $1.15 \cdot 10^{-2}$ |
| | $2\times$ | $1.35 \cdot 10^{-1}$ | $1.5 \cdot 10^{-2}$ | $1.1 \cdot 10^{-2}$ | $2.11 \cdot 10^{-3}$ | $4.25 \cdot 10^{-1}$ | $3.48 \cdot 10^{-2}$ |
| NO-LIDK$^\dagger$ | $1\times$ | $7.91 \cdot 10^{-2}$ | $3.82 \cdot 10^{-3}$ | $5.38 \cdot 10^{-3}$ | $2.47 \cdot 10^{-3}$ | $1.45 \cdot 10^{-1}$ | $1.01 \cdot 10^{-2}$ |
| | $2\times$ | $3.9 \cdot 10^{0}$ | $4.17 \cdot 10^{-1}$ | $1.41 \cdot 10^{-1}$ | $7.21 \cdot 10^{-2}$ | $2.36 \cdot 10^{0}$ | $5.25 \cdot 10^{-1}$ |
| LOGLO-FNO | $1\times$ | $\mathbf{6.72} \cdot 10^{-2}$ | $\mathbf{2.8} \cdot 10^{-3}$ | $\mathbf{4.0} \cdot 10^{-3}$ | $\mathbf{2.14} \cdot 10^{-3}$ | $\mathbf{1.31} \cdot 10^{-1}$ | $\mathbf{6.78} \cdot 10^{-3}$ |
| | $2\times$ | $\underline{7.31} \cdot 10^{-2}$ | $\mathbf{3.2} \cdot 10^{-3}$ | $\mathbf{4.36} \cdot 10^{-3}$ | $\underline{1.43} \cdot 10^{-3}$ | $3.13 \cdot 10^{-1}$ | $\underline{8.36} \cdot 10^{-3}$ |
| | | | **20-step (entire trajectory) Autoregressive Evaluation** | | | | |
| Base FNO | $1\times$ | $1.81 \cdot 10^{-1}$ | $1.25 \cdot 10^{-2}$ | $1.44 \cdot 10^{-2}$ | $3.26 \cdot 10^{-3}$ | $2.8 \cdot 10^{-1}$ | $3.22 \cdot 10^{-2}$ |
| | $2\times$ | $1.82 \cdot 10^{-1}$ | $1.26 \cdot 10^{-2}$ | $1.45 \cdot 10^{-2}$ | $\mathbf{1.78} \cdot 10^{-3}$ | $5.52 \cdot 10^{-1}$ | $3.24 \cdot 10^{-2}$ |
| NO-LIDK* | $1\times$ | $1.97 \cdot 10^{-1}$ | $1.44 \cdot 10^{-2}$ | $1.6 \cdot 10^{-2}$ | $3.36 \cdot 10^{-3}$ | $2.95 \cdot 10^{-1}$ | $3.94 \cdot 10^{-2}$ |
| | $2\times$ | $3.72 \cdot 10^{-1}$ | $4.63 \cdot 10^{-2}$ | $3.16 \cdot 10^{-2}$ | $2.39 \cdot 10^{-3}$ | $3.66 \cdot 10^{-1}$ | $1.09 \cdot 10^{-1}$ |
| NO-LIDK$^\dagger$ | $1\times$ | $1.87 \cdot 10^{-1}$ | $1.35 \cdot 10^{-2}$ | $1.48 \cdot 10^{-2}$ | $3.37 \cdot 10^{-3}$ | $2.1 \cdot 10^{-1}$ | $3.7 \cdot 10^{-2}$ |
| | $2\times$ | $1.04 \cdot 10^{11}$ | $1.18 \cdot 10^{10}$ | $5.64 \cdot 10^{8}$ | $1.77 \cdot 10^{9}$ | $1.33 \cdot 10^{1}$ | $9.99 \cdot 10^{0}$ |
| LOGLO-FNO | $1\times$ | $\mathbf{1.62} \cdot 10^{-1}$ | $\mathbf{1.07} \cdot 10^{-2}$ | $\mathbf{1.26} \cdot 10^{-2}$ | $\mathbf{3.09} \cdot 10^{-3}$ | $\mathbf{1.87} \cdot 10^{-1}$ | $\mathbf{2.7} \cdot 10^{-2}$ |
| | $2\times$ | $\mathbf{1.72} \cdot 10^{-1}$ | $\mathbf{1.16} \cdot 10^{-2}$ | $\mathbf{1.34} \cdot 10^{-2}$ | $\underline{1.79} \cdot 10^{-3}$ | $3.49 \cdot 10^{-1}$ | $\mathbf{3.02} \cdot 10^{-2}$ |

# M   Training and Inference Time Comparison of LOGLO-FNO

## M.1   Comparison with Numerical Solver

The inference time of LOGLO-FNO to generate predictions on the Turbulent Radiative Layer 3D dataset is provided in Table 20 and compared against the CPU-only numerical solver Athena++ used to generate the original simulations of the TRL3D dataset. However, in order to establish a fair comparison with the numerical solver, the training time of the neural surrogate (LOGLO-FNO), which is about 12 hours for 50 epochs on TRL3D, should also be considered.

Table 20: Comparison of runtimes to generate all trajectories at the full resolution for the Turbulent Radiative Layer 3D dataset.

| Solver Type | Hardware | Runtime |
|---|---|---|
| Numerical Solver: Athena++ (Stone et al., 2020; Ohana et al., 2024) | CPU (128 cores) | 34,560 CPU-h |
| Neural Surrogate: LOGLO-FNO | one H100 GPU | 38.143767 mins |

Table 21: Scaling factor of runtimes to generate all trajectories at training and higher ($2\times$) spatial resolutions for the Turbulent Radiative Mixing Layer 3D dataset.

| Metric | Data Split | Spatial Resolution ($64\times64\times128$) | Spatial Resolution ($128\times128\times256$) | Scaling Factor |
|---|---|---|---|---|
| Total Grid Points | ✗ | 524,288 | 4,194,304 | 8x |
| Runtime (9 Trajectories) | Test | 31,430.94 $\pm$ 0.80 ms (~31.4 sec) | 228,862.60 $\pm$ 0.81 ms (~3.8 min) | ~7.28x |
| Runtime (90 Trajectories) | Full | 314,309.4 $\pm$ 8.0 ms (~5.2 min) | 2,288,626.0 $\pm$ 8.1 ms (~38.1 min) | ~7.28x |

We measure the runtimes by generating the trajectory predictions only for the *test set*, containing 9 trajectories, each of which has 101 timesteps (row 2 in Table 21). The runtime consumed by the numerical solver Athena++ is taken from Ohana et al. (2024)[¶]. The runtime for the full dataset containing 90 trajectories (row 3 in Table 21) is obtained by naive scaling (i.e., multiplying by 10) of the GPU runtime for the 9 trajectories. During inference, we use a batch size of 1 and report the mean runtime and standard deviation over 10 iterations for both spatial resolutions listed in Table 21.

## M.2   Training & Inference Time and GPU Memory Usage Comparison with Base FNO and NO-LIDK

In this part, we provide the training and inference times as well as the GPU memory usage of LOGLO-FNO and compare them with FNO and NO-LIDK variants of baseline models since these are the most related model architectures for LOGLO-FNO. We note that all experiments, configured through the Slurm workload manager, are conducted on a single H100 NVIDIA graphics card with 96 GB of memory. The benchmark problem we consider is the 2D Kolmogorov Flow (Li et al., 2022a) and 1-step training outlined in §4.0.1.

In order to establish a complete picture of the trade-off among these models in terms of memory usage and total epoch times, we list the *training times* and GPU memory usage for the forward and backward passes in Table 22. We observe that baseline FNO🏆 wins all other models, both in terms of GPU memory consumed and time needed for the completion of the full training cycle of 136 epochs. NO-LIDK◇🥈, employing a differential kernel as a parallel branch, occupies the second place. LOGLO-FNO🥉 comes third in training time, while it requires slightly more memory than NO-LIDK*, which uses a local integral kernel, takes fourth

---

[¶]https://polymathic-ai.org/the_well/datasets/turbulent_radiative_layer_3D

place when comparing training time. In last place is the NO-LIDK variant, which utilizes both kernels as parallel branches in addition to the global Fourier branch.

Table 22: Training times and GPU memory usage for the forward and backward passes during the 1-step training for a total of 136 epochs on the Kolmogorov Flow 2D training dataset. The reported values are the average over 4 training runs on the training set. NO-LIDK*, NO-LIDK$^\diamond$, and NO-LIDK$^\dagger$ represent the variants trained with the local integral (DISCO), differential kernel, and both layers as additional parallel branches, in that corresponding order.

| Model | Training Time | Slowdown wrt. FNO | GPU Mem. (GB) | Mem. Fact. vs. FNO | Batch Size | Spatial Res. | Fourier Modes | Hidden Channel | $P_{size}$ | Radius Cutoff |
|---|---|---|---|---|---|---|---|---|---|---|
| Base FNO🏆 | **1h 36m 47s** $\pm$ 8.35s | ref. | **33.90** $\pm$ 0.81 | ref. | 256 | 128x128 | 40 | 65 | ✗ | ✗ |
| LOGLO-FNO🏆 | 5h 19m 40s $\pm$ 16.28s | 3.30× | 76.02 $\pm$ 2.10 | 2.24× | 256 | 128x128 | 40 | 65 | 16 × 16 | ✗ |
| NO-LIDK* | 7h 23m 30s $\pm$ 16.59s | 4.58× | 69.33 $\pm$ 2.02 | 2.05× | 256 | 128x128 | 20 | 129 | ✗ | $0.05\pi$ |
| NO-LIDK$^\diamond$🏆 | 2h 52m 54s $\pm$ 7.55s | 1.79× | 35.63 $\pm$ 1.12 | 1.05× | 256 | 128x128 | 40 | 65 | ✗ | ✗ |
| NO-LIDK$^\dagger$ | 7h 31m 11s $\pm$ 13.54s | 4.66× | 78.75 $\pm$ 4.63 | 2.32× | 256 | 128x128 | 20 | 127 | ✗ | $0.05\pi$ |

Subsequently, we compute the *inference times* and GPU memory usage for the 1-step predictions and list them in Table 23, with the provided values averaged over 100 repetitions.

Table 23: Inference times and GPU memory usage for the forward pass during 1-step evaluation on the Kolmogorov Flow 2D test dataset. The reported values are the average over 100 iterations on the test set. NO-LIDK*, NO-LIDK$^\diamond$, and NO-LIDK$^\dagger$ represent the variants trained with the local integral (DISCO), differential kernel, and both layers as additional parallel branches, in that corresponding order.

| Model | Inference Time (ms) | Slowdown wrt. FNO | GPU Mem. (GB) | Batch Size | Spatial Res. | Fourier Modes | Hidden Channel | $P_{size}$ | Radius Cutoff |
|---|---|---|---|---|---|---|---|---|---|
| Base FNO🏆 | **1471.41** $\pm$ 0.06 | ref. | 10.74 $\pm$ 0.0 | 256 | 128x128 | 40 | 65 | ✗ | ✗ |
| LOGLO-FNO🏆 | 4899.53 $\pm$ 0.15 | 3.33× | 18.31 $\pm$ 0.0 | 256 | 128x128 | 40 | 65 | 16 × 16 | ✗ |
| NO-LIDK* | 7601.22 $\pm$ 7.17 | 5.17× | 19.63 $\pm$ 0.0 | 256 | 128x128 | 20 | 129 | ✗ | $0.05\pi$ |
| NO-LIDK$^\diamond$🏆 | 2477.68 $\pm$ 0.14 | 1.68× | **10.19** $\pm$ 0.0 | 256 | 128x128 | 40 | 65 | ✗ | ✗ |
| NO-LIDK$^\dagger$ | 7096.81 $\pm$ 4.34 | 4.82× | 21.43 $\pm$ 0.0 | 256 | 128x128 | 20 | 127 | ✗ | $0.05\pi$ |

We observe a similar trend with baseline FNO being the fastest, however, consuming slightly higher GPU memory than NO-LIDK$^\diamond$, which is about 1.7× slower. In the third place is our LOGLO-FNO model, incurring a 3.3× slowdown factor. Pertinent to our model is NO-LIDK*, and it is about 1.6× slower while also incurring a slightly higher memory consumption than LOGLO-FNO under the considered hyperparameter configurations, following Liu-Schiaffini et al. (2024). Finally, the NO-LIDK$^\dagger$ variant, consisting of both branches, incurs the highest GPU memory, although it is slightly faster than the NO-LIDK* variant with only the local integral kernel branch as an additional pathway.

# N   Confidence Analysis of LOGLO-FNO and Baselines

In this section, we report the mean and standard deviation values for the base FNO and NO-LIDK variants, comparing them with our proposed LOGLO-FNO model. We repeat the 1-step training setup discussed in §4.0.1 on the 2D Kolmogorov Flow dataset four times with different random seeds, and evaluate the 1-step as well as the 5-step autoregressive rollouts. The plot in Figure 39 shows the mean and 95% confidence interval values of fRMSE and WLR errors, with the hatched and solid bars visualizing the 1-step and 5-step autoregressive rollout errors in that order.

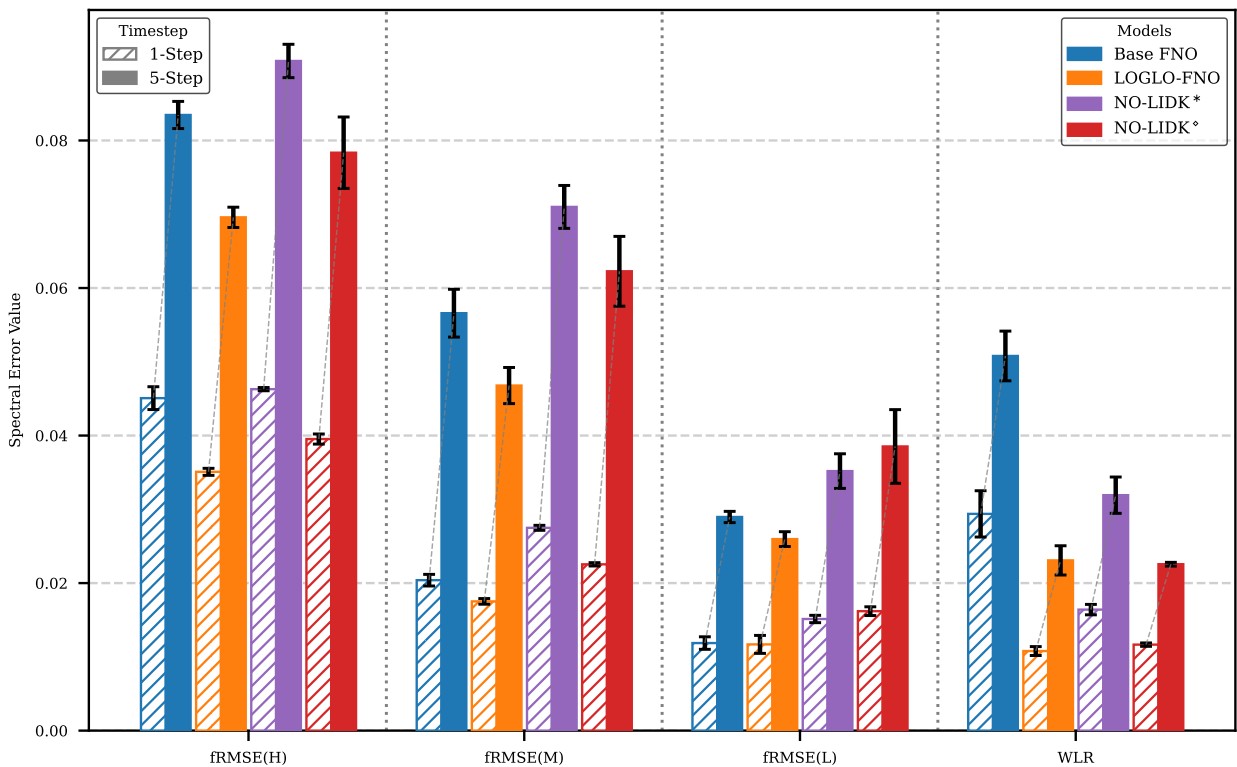

Figure 39: Confidence analysis of baseline FNO, NO-LIDK (Liu-Schiaffini et al., 2024), and proposed LOGLO-FNO for the 1-step and 5-step autoregressive evaluation of fRMSE and WLR (Wan et al., 2023) metrics on the test set of 2D Kolmogorov Flow (Li et al., 2022b). NO-LIDK* and NO-LIDK$^\diamond$ represent the variants trained with the local integral (DISCO) and differential kernel layers, respectively.

We observe that LOGLO-FNO consistently outperforms baseline FNO and is significantly better than NO-LIDK*, utilizing the local integral kernel, across all metrics. Particularly noteworthy are the improvements we observe in the rollout scenarios.

In a similar vein, in Figure 40, we visualize the mean and 95% confidence interval values of nRMSE and MELR errors, whereas Figure 41 establishes a comparison of the plain and boundary RMSE values.

We find that while LOGLO-FNO is the best model on the nRMSE metric, NO-LIDK$^\diamond$ employing a differential kernel layer outperforms all models, reinforcing the results found in Table 8. However, we note that the results on NO-LIDK$^\diamond$ are provided only for the sake of completeness, and LOGLO-FNO should be compared with NO-LIDK* when assessing its effectiveness.

Liu-Schiaffini et al. (2024) find the differential kernel layer to be effective in reducing the boundary error for non-periodic boundary conditions. Additionally, in our study on Kolmogorov Flow 2D, it has been found to be effective for periodic boundary conditions as well, enforced through padding. However, this advantage is not maintained in rollouts (see Figure 41).

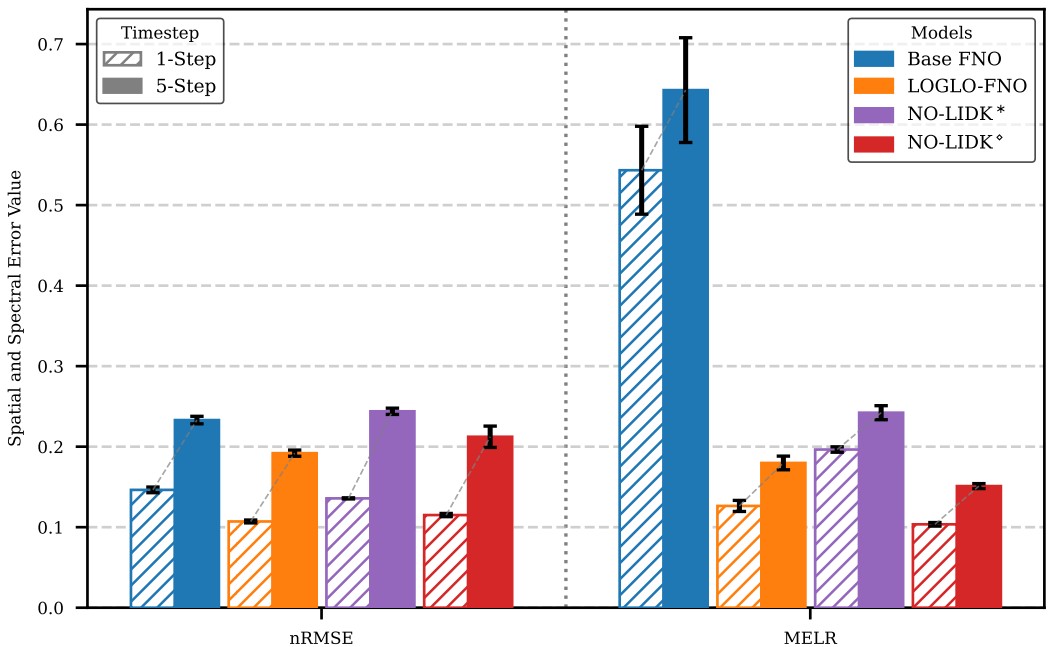

Figure 40: Confidence analysis of baseline FNO, NO-LIDK (Liu-Schiaffini et al., 2024), and proposed LOGLO-FNO for the 1-step and 5-step autoregressive evaluation of nRMSE and MELR (Wan et al., 2023) metrics on the test set of 2D Kolmogorov Flow (Li et al., 2022b). NO-LIDK* and NO-LIDK⋄ represent the variants trained with the local integral (DISCO) and differential kernel layers, respectively.

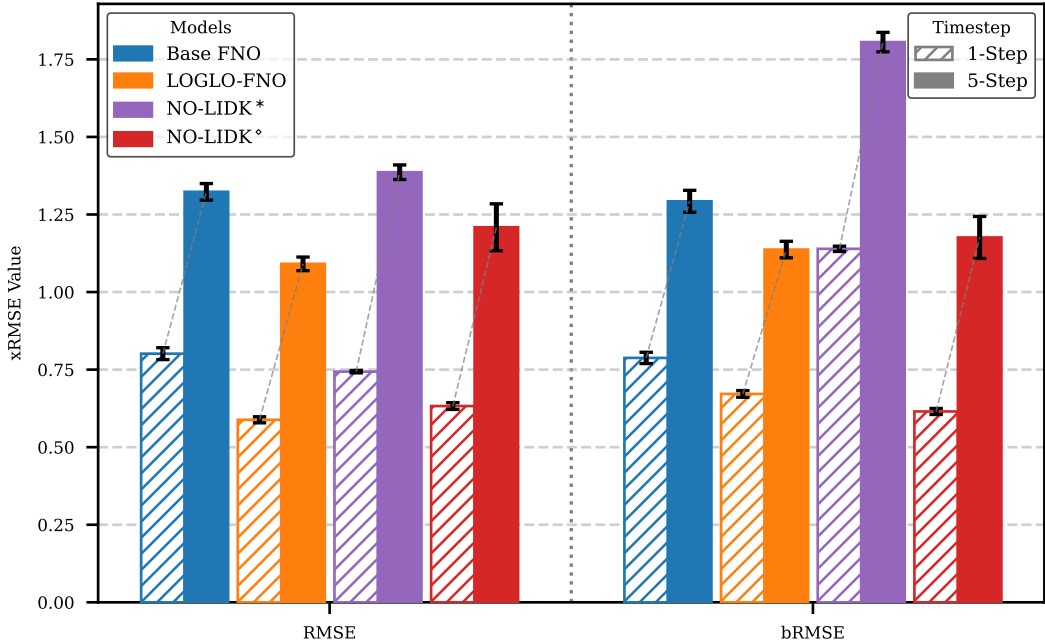

Figure 41: Confidence analysis of baseline FNO, NO-LIDK (Liu-Schiaffini et al., 2024), and proposed LOGLO-FNO for the 1-step and 5-step autoregressive evaluation of xRMSE (plain and boundary) metrics on the test set of 2D Kolmogorov Flow (Li et al., 2022b). NO-LIDK* and NO-LIDK⋄ represent the variants trained with the local integral (DISCO) and differential kernel layers, respectively.

## O  Model Hyperparameters

### O.1  Modern U-Net Hyperparameters

The Table 24 below presents the hyperparameters for the modern version of U-Net baseline we use from the PDEArena (Gupta & Brandstetter, 2023) benchmark codebase.

| PDE Problem | Channel Multiplier | Learning Rate (LR) | LR Sched. | Sched. Steps | Sched. Gamma | Train Type | Train Loss | Total Epochs |
|---|---|---|---|---|---|---|---|---|
| 2D Kolmogorov Flow | (1, 2, 2, 3, 4) | $1.0 \cdot e^{-3}$ | StepLR | 33 | 0.5 | 1-step | MSE | 136 |
| 2D CNS Turb | (1, 2, 2, 3, 4) | $1.0 \cdot e^{-3}$ | StepLR | 33 | 0.5 | 1-step | MSE | 136 |

Table 24: Model hyperparameters for the modern U-Net (Gupta & Brandstetter, 2023) baseline.

### O.2  Base FNO Hyperparameters

The Table 25 below lists the hyperparameters for training the FNO baseline model on the two-dimensional and three-dimensional PDE problems considered in our experiments. Note that we borrow the hyperparameters from Liu-Schiaffini et al. (2024) for the Kolmogorov Flow case and Takamoto et al. (2022) for the Diffusion-Reaction 2D PDE.

| PDE Problem | Fourier Modes | Channel Dimens. | Learning Rate (LR) | LR Scheduler | Sched. Steps | Sched. Gamma | Train Type | Train Loss | Total Epochs |
|---|---|---|---|---|---|---|---|---|---|
| 2D Diffusion-Reaction | 12 | 20 | $1.0 \cdot e^{-3}$ | StepLR | 100 | 0.5 | AR | MSE | 500 |
| 2D Kolmogorov Flow | 40 | 65 | $1.0 \cdot e^{-3}$ | StepLR | 33 | 0.5 | 1-step | MSE | 136 |
| 2D Wave-Gauss | 20 | 32 | $1.0 \cdot e^{-3}$ | StepLR | 33 | 0.5 | 1-step | MSE | 136 |
| 2D CNS Turb | 40 | 65 | $1.0 \cdot e^{-3}$ | StepLR | 33 | 0.5 | 1-step | MSE | 136 |
| 2D Comp. Euler RP | 40 | 65 | $1.0 \cdot e^{-3}$ | StepLR | 33 | 0.5 | 1-step | MSE | 136 |
| Turbulent Radiative Layer 3D | 12 | 48 | $1.0 \cdot e^{-3}$ | StepLR | 10 | 0.5 | 1-step | MSE | 50 |

Table 25: Model hyperparameters for FNO baseline.

### O.3  F-FNO Hyperparameters

Table 26 provides the hyperparameters for the F-FNO baseline model from Tran et al. (2023). As with the baseline FNO model, we use four Fourier layers.

| PDE Problem | Fourier Modes | Channel Dimens. | Learning Rate (LR) | LR Scheduler | Warmup Steps | Train Type | Train Loss | Total Epochs |
|---|---|---|---|---|---|---|---|---|
| 2D Diffusion-Reaction | 16 | 32 | $6.0 \cdot e^{-4}$ | Cosine Warmup | 200 | AR | MSE | 500 |
| 2D Kolmogorov Flow | 40 | 65 | $1.0 \cdot e^{-3}$ | Cosine Warmup | 60 | 1-step | MSE | 136 |

Table 26: Model hyperparameters for F-FNO (Tran et al., 2023) baseline.

### O.4  LSM Hyperparameters

Tables 27 provides the hyperparameters used for training the Latent Spectral Model (Wu et al., 2023) baseline on the Kolmogorov Flow 2D, CNS Turb 2D, Wave-Gauss 2D, and Diffusion-Reaction 2D datasets, whereas Table 28 lists the model hyperparameters used for the LSM architecture configuration.

We found the model to be very sensitive to the learning rate in the full autoregressive training setup and, hence, we use a low learning rate of $5.0 \cdot e^{-5}$ in addition to employing gradient clipping with a max norm of 5.0.

| PDE Problem | Learning Rate (LR) | LR Scheduler | Sched. Steps | Sched. Gamma | Train Type | Train Loss | Total Epochs |
|---|---|---|---|---|---|---|---|
| 2D Diffusion-Reaction | $5.0 \cdot e^{-5}$ | StepLR | 100 | 0.5 | AR | MSE | 500 |
| 2D Kolmogorov Flow | $1.0 \cdot e^{-3}$ | StepLR | 33 | 0.5 | 1-step | MSE | 136 |
| 2D CNS Turb | $1.0 \cdot e^{-3}$ | StepLR | 33 | 0.5 | 1-step | MSE | 136 |
| 2D Wave-Gauss | $1.0 \cdot e^{-3}$ | StepLR | 33 | 0.5 | 1-step | MSE | 136 |
| 3D Turbulent Radiative Layer | $1.0 \cdot e^{-3}$ | StepLR | 10 | 0.5 | 1-step | MSE | 50 |

Table 27: Training settings for the LSM (Wu et al., 2023) baseline model.

| PDE Problem | Num. Tokens | Num. Basis | Channel Dimens. (init.) | Downsample Ratio | Num. Scales | Patch Size | Interpol. Type |
|---|---|---|---|---|---|---|---|
| 2D Diffusion-Reaction | 4 | 8 | 32 | $^5/_{10}$ | 5 | (4,4) | bilinear |
| 2D Kolmogorov Flow | 4 | 16 | 64 | $^5/_{10}$ | 5 | (4,4) | bilinear |
| 2D CNS Turb | 4 | 16 | 64 | $^5/_{10}$ | 5 | (4,4) | bilinear |
| 2D Wave-Gauss | 4 | 8 | 32 | $^5/_{10}$ | 5 | (4,4) | bilinear |
| 3D Turbulent Radiative Layer | 4 | 16 | 64 | $^5/_{10}$ | 5 | (4,4,8) | trilinear |

Table 28: Further Model hyperparameters for LSM (Wu et al., 2023) baseline.

Channel dimensions (init.) represent the number of channels used for the incoming full spatial resolution, including which the remaining four spatial scales and their respective channels follow.

## O.5  Transolver Hyperparameters

Tables 29 provides the hyperparameters used for training the Transolver Model (Wu et al., 2024) baseline on the Kolmogorov Flow 2D, CNS Turb 2D, Wave-Gauss 2D, and Diffusion-Reaction 2D datasets, whereas Table 30 lists the model hyperparameters used for the Transolver model architecture configuration.

| PDE Problem | Optimizer | Learning Rate (LR) | LR Scheduler | Weight Decay | Train Type | Train Loss | Total Epochs |
|---|---|---|---|---|---|---|---|
| 2D Diffusion-Reaction | AdamW | $4.0 \cdot e^{-4}$ | OneCycleLR | $1.0 \cdot e^{-4}$ | AR | MSE | 500 |
| 2D Kolmogorov Flow | AdamW | $6.0 \cdot e^{-3}$ | OneCycleLR | $1.0 \cdot e^{-4}$ | 1-step | MSE | 500 |
| 2D CNS Turb | AdamW | $6.0 \cdot e^{-3}$ | OneCycleLR | $1.0 \cdot e^{-4}$ | 1-step | MSE | 500 |
| 2D Wave-Gauss | AdamW | $6.0 \cdot e^{-3}$ | OneCycleLR | $1.0 \cdot e^{-4}$ | 1-step | MSE | 500 |

Table 29: Training settings for the Transolver (Wu et al., 2024) baseline model.

| PDE Problem | Num. Layers | Num. Heads | Num. Slices | Hidden Channels | Activation Function | MLP Ratio | Apply Dropout |
|---|---|---|---|---|---|---|---|
| 2D Diffusion-Reaction | 4 | 8 | 32 | 64 | GeLU | 1 | False |
| 2D Kolmogorov Flow | 9 | 8 | 64 | 128 | GeLU | 1 | False |
| 2D CNS Turb | 9 | 8 | 64 | 128 | GeLU | 1 | False |
| 2D Wave-Gauss | 9 | 8 | 64 | 32 | GeLU | 1 | False |

Table 30: Model architecture hyperparameters for Transolver (Wu et al., 2024) baseline.

### O.6 U-FNO Hyperparameters

Table 31 below lists the hyperparameters for the U-FNO baseline model from Wen et al. (2022).

| PDE Problem | Fourier Modes | Channel Dimens. | Learning Rate (LR) | LR Scheduler | Sched. Steps | Sched. Gamma | Train Type | Train Loss | Total Epochs |
|---|---|---|---|---|---|---|---|---|---|
| 2D Diffusion-Reaction | 10 | 19 | $5.0 \cdot e^{-4}$ | StepLR | 10 | 0.9 | AR | MSE | 500 |
| 2D Kolmogorov Flow | 36 | 48 | $1.0 \cdot e^{-3}$ | StepLR | 2 | 0.9 | 1-step | MSE | 136 |
| 2D CNS Turb | 36 | 48 | $5.0 \cdot e^{-4}$ | StepLR | 10 | 0.9 | 1-step | MSE | 136 |
| 2D Wave-Gauss | 16 | 32 | $5.0 \cdot e^{-4}$ | StepLR | 10 | 0.9 | 1-step | MSE | 136 |

Table 31: Model hyperparameters for U-FNO (Wen et al., 2022) baseline.

### O.7 NO-LIDK Hyperparameters

Table 32 lists the hyperparameters for different configurations of the Neural Operators with Localized Integral and Differential Kernels (NO-LIDK) model from Liu-Schiaffini et al. (2024). For the Kolmogorov Flow 2D data, following the authors' training setup (see Table 4 in Liu-Schiaffini et al. (2024)), we use 20 modes and a hidden channel dimension of 127 for the model employing both local integral and differential kernel layers (NO-LIDK$^\dagger$), 20 modes and a hidden channel dimension of 129 for the model using only the local integral kernel layer (NO-LIDK$^*$), and 40 modes and a hidden channel dimension of 65 for the model with just the differential kernel layer (NO-LIDK$^\diamond$) as parallel layers in addition to the global spectral convolution layer of base FNO. All these variants use 4 Fourier layers in total.

| PDE Problem | Integral Kernel | Differential Kernel | Learning Rate (LR) | LR Sched. | Sched. Steps | Sched. Gamma | Train Type | Train Loss | Total Epochs |
|---|---|---|---|---|---|---|---|---|---|
| 2D Kolmogorov Flow | ✓ | ✓ | $1.0 \cdot e^{-3}$ | StepLR | 33 | 0.5 | 1-step | MSE | 136 |
| 2D Wave-Gauss | ✓ | ✓ | $1.0 \cdot e^{-3}$ | StepLR | 33 | 0.5 | 1-step | MSE | 136 |
| 2D Comp. Euler RP | ✓ | ✓ | $1.0 \cdot e^{-3}$ | StepLR | 33 | 0.5 | 1-step | MSE | 136 |
| 2D CNS Turb | ✓ | ✓ | $1.0 \cdot e^{-3}$ | StepLR | 33 | 0.5 | 1-step | MSE | 136 |
| Turbulent Radiative Layer 3D | ✗ | ✓ | $6.0 \cdot e^{-3}$ | OneCycleLR | 10 | ✗ | 1-step | MSE | 50 |

Table 32: Model hyperparameters for NO-LIDK (Liu-Schiaffini et al., 2024) baseline.

The local integral kernel for the disco convolution employs a radius cutoff value of 0.0078125, 0.0078125, and 0.00390625 for the Wave-Gauss 2D, Compressible Euler four-quadrant Riemann problem 2D, and CNS Turb 2D problems, respectively, following the heuristic suggested by Liu-Schiaffini et al. (2024). As in the case of the baseline FNO model, the number of spectral filters and hidden channels is set as 40 and 65, resp., for all three configurations of the trained models (i.e., disco layers – NO-LIDK$^*$, differential layers – NO-LIDK$^\diamond$, and disco+differential layers – NO-LIDK$^\dagger$) for the CNS Turb 2D dataset, whereas we use 20 modes and 32 hidden channels for the Wave-Gauss 2D problem. We use the hyperparameter settings applied for 2D Kolmogorov Flow for the Compressible Euler four-quadrant Riemann problem 2D PDE.

For the Turbulent Radiative Layer 3D data, we use 16, 16, and 20 modes along the x, y, and z axes. The number of hidden channels is set as 96, and the model consists of 4 layers, as in the case of all 2D experiments.

### O.8 LOGLO-FNO Implementation and Hyperparameters

**Implementation.** Our implementation uses PyTorch v2 (Ansel et al., 2024), NumPy (Harris et al., 2020), and FNO base implementation adapted from the neural operator library (Kossaifi et al., 2024a).

**Hyperparameters.** The Kolmogorov Flow problem uses a batch size of 256 and a patch size of $16 \times 16$ (in the local branch) for the 1-step training strategy, whereas it is set as $8 \times 8$ for Diffusion-Reaction 2D due to memory limitations with fully autoregressive training on long trajectories of 91 timesteps. An experiment

employing data-parallel training using a patch size of 16 has not resulted in improved performance. Therefore, we set the patch size to 8 when extracting non-overlapping patches on the domain for the Diffusion-Reaction problem and report the results for this setting.

| PDE Problem | Fourier Modes (Global) | Channel Dimens. | Learning Rate (LR) | LR Scheduler | Sched. Steps | Sched. Gamma | Train Type | Train Loss | Total Epochs |
|---|---|---|---|---|---|---|---|---|---|
| 2D Diffusion-Reaction | 10 | 24 | $8.0 \cdot e^{-4}$ | StepLR | 100 | 0.5 | AR | MSE + (mid&high) Freq. | 500 |
| 2D Kolmogorov Flow | 40 | 65 | $4.4 \cdot e^{-3}$ | StepLR | 33 | 0.5 | 1-step | MSE + (mid&high) Freq. | 136 |
| 2D Wave-Gauss | 20 | 32 | $4.4 \cdot e^{-3}$ | StepLR | 33 | 0.5 | 1-step | MSE + (mid&high) Freq. | 136 |
| 2D Comp. Euler RP | 40 | 65 | $4.4 \cdot e^{-3}$ | StepLR | 33 | 0.5 | 1-step | MSE + (mid&high) Freq. | 136 |
| 2D CNS Turb | 40 | 65 | $4.4 \cdot e^{-3}$ | StepLR | 33 | 0.5 | 1-step | MSE + (mid&high) Freq. | 136 |
| Turbulent Radiative Layer 3D | 12 | 52 | $4.0 \cdot e^{-3}$ | StepLR | 10 | 0.5 | 1-step | MSE + (mid&high) Freq. | 50 |

Table 33: Model hyperparameters for our proposed LOGLO-FNO in the main paper with simple summation (+) as the fusion method (Fig. 1) for the features from the HFP, global, and local branches.

All models use a batch size of 64, 256, and 256 for the CNS Turb 2D, Compressible Euler four-quadrant Riemann problem 2D, and Wave-Gauss 2D problems, respectively. LOGLO-FNO uses a patch size of $16 \times 16$ for the Compressible Euler four-quadrant Riemann problem 2D and Wave-Gauss 2D from Poseidon (Herde et al., 2024) and CNS Turb 2D dataset, which is provided by PDEBench (Takamoto et al., 2022).

The kernel size and stride have been set to 4 in the high-frequency extraction and propagation modules for all PDE problems. The local branch (e.g., with a patch size of $8 \times 8$ or $16 \times 16$) uses all available frequency modes. This would yield 16 and 9 Fourier modes for the spatial dimensions x and y, respectively, when a patch size of $16 \times 16$ is considered. Analogously, assuming an equal number of sampling points along all three spatial axes of a 3D spatial data, the local branch configured with a patch size of $16 \times 16 \times 16$ would use all the 16, 16, and 9 Fourier modes for the spatial dimensions x, y, and z, respectively. Since Turbulent Radiative Layer 3D (Ohana et al., 2024) has 2x more observation points on the z-axis (i.e., $128 \times 128 \times 256$), we use a proportionate patch size of $16 \times 16 \times 32$, resulting in 16, 16, and 17 Fourier modes along the x, y, and z axes, respectively.

## P    Table of Notations and Mathematical Symbols

To serve as a handy guide, the following tabulation describes the symbols used within this manuscript.

| | |
|---|---|
| $\mathbb{R}_+$ | Set of positive real numbers. Specifically, $t \in (0, \mathrm{T}]$ |
| $\nabla$ | Gradient or vector derivative operator ($\frac{\partial}{\partial x}$, $\frac{\partial}{\partial y}$, ...) |
| $\Delta$ | Laplacian ($\nabla^2$) |
| $\mathcal{A}, \mathcal{U}$ | Banach spaces of functions |
| $\partial_t$ | Partial derivative with respect to $t$ |
| $\partial_x$ | Partial derivative with respect to $x$ |
| $\partial_y$ | Partial derivative with respect to $y$ |
| $u(x, t)$ | Solution of a PDE at time $t$ for a given spatial coordinate $x$ |
| $\mathbf{v}$ | Velocity vector field ($\vec{\mathbf{x}}$, $\vec{\mathbf{y}}$, ...) |
| $\mathbf{p}$ | PDE parameter values, either a scalar or vector |
| $\mathcal{N}_\theta$ | Neural network with learnable parameters $\theta$ |
| $\mathcal{P}, \mathcal{Q}$ | Lifting and projection layers, respectively |
| $\mathcal{K}_l, \mathcal{K}_g$ | Local and global kernel integral operator, respectively |
| $\mathcal{L}$ | Number of layers in the network |
| $\mathcal{C}$ | Cost function |
| $u^0(x)$ | Initial data of the PDE at time $t = 0$ |
| $\Omega$ | Domain of the PDE under consideration |
| $\partial\Omega$ | Boundary of the domain $\Omega$ under consideration |
| $N_t$ | Total number of timesteps in the simulation |
| $N_x$ | Total number of spatial points along the x-axis of the simulation |
| $N_y$ | Total number of spatial points along the y-axis of the simulation |
| $N_z$ | Total number of spatial points along the z-axis of the simulation |
| $N_c$ | Total number of channels or fields in the simulation |
| $d_c$ | channel dimension a.k.a. width of the network |
| $p_{size}$ | Patch size or sub-domain size |
| $\nu$ | Viscosity coefficient of Navier-Stokes equations |
| $Re$ | Reynolds Number |
| $\rho$ | Mass density in Navier-Stokes equations |
| $\eta$ | Shear viscosity in Compressible Navier-Stokes equations |
| $\zeta$ | Bulk viscosity in Compressible Navier-Stokes equations |
| $\epsilon$ | Internal energy in Compressible Navier-Stokes equations |
| $D_u, D_v$ | Diffusion coefficients for the activator and inhibitor in the Diffusion-Reaction equation, respectively |
| $\mathcal{F}$ | Fourier transform of a signal |

