# OpenReview forum: "LOGLO-FNO: Efficient Learning of Local and Global Features in Fourier Neural Operators"
_TMLR — Accepted by TMLR_

### Review · Reviewer_CTmE · 2025-08-04

**Summary Of Contributions:**

This paper introduces a variant of the Fourier Neural Operator (FNO) designed to improve its ability to capture high-frequency and localized features. The proposed modifications include:
- A patch-wise FNO module running in parallel with the standard FNO.
- An additional input head restricted to high-frequency components.
- A frequency-aware loss leveraging radial binning.

The method is conceptually sound and shows consistent empirical gains over strong baselines. However, the added complexity raises questions about generalization and computational cost.

**Additional Comments:**

Based on the implementation of the FNO block [1], I'm not sure that equation (1) is correct. Using notation from equation (1), the implementation seems to be:

$Z' = W_f*Z + b_f$

$Y = \sigma\big(K(Z') + Z'\big)$

Could the authors double-check this equation and ensure that it corresponds to the one implemented?

[1] https://github.com/neuraloperator/neuraloperator/blob/main/neuralop/layers/fno_block.py

**Audience:**

Yes

**Audience Explanation:**

The paper does not constitute a major conceptual breakthrough, but the approach is technically sound and empirically convincing. Its ease of use and good performance may make it appealing in applied settings.

**Broader Impact Concerns:**

- **Training performance**: The authors discussed extensively the impact of their method on the number of parameters, but this should be completed by a training and inference time comparison, as I would expect that the additional terms and the frequency loss impacts greatly the efficiency. This is important to assess the balance between the added computational cost and the performance gain.  The impact on GPU memory usage could also be very relevant.
- **Resolution invariance**: The authors propose to slice the input domain into patches to handle local features, akin to ViTs.  Yet, this might impact the generalizability of the method to higher or lower resolutions, as we now ViTs does not necessarily generalize to resolution outside their training set. Moreover, zero-shot multi-resolution is the core motivation behind FNO, hence assessing that LOGLO-FNO maintains this feature is detrimental. Such evaluation is missing from the paper.
- **Missing ablations**: Several architectural components are introduced but not validated. Missing ablations include:
	- Patch-size sensitivity, especially given spectral resolution trade-offs (decreasing patch-size might improve the model on local features, but also decreases the number of modes per patches)
	- Adaptive Gaussian noise
	- High-frequency component in the model. It is unclear for me how this affect the model performances, since the additional filtered input remains high frequency, and thus difficult to learn with a neural network even if the low frequency component has been removed.
- **Performances**: It is unclear why the author also supervise mid-range frequencies while the paper focuses on high-frequencies. In that case, why not supervising all frequency ranges ? Moreover, according to Table (1), the gain in high frequency RMSE does not transfer well when increasing the prediction horizon.

To summarize, the paper shares an interesting solution, yet not groundbreaking. The method is sound, addresses a known limitation of FNOs, and seems to perform reasonably well. My main concerns are on the writing and organisation of the paper, which deserves a refactoring effort. As it stands, it does not align with the standard of rigor for a TMLR paper. Yet this can be quickly addressed during revision.

I also have several scientific concerns, especially about the trade-off between performances and impact on training time and memory usage. Few ablations are also missing to confirm that LOGLO-FNO maintains the important properties of FNO (resolution invariance), and assessing all introduced contributions properly.

**Claims And Evidence:**

Yes

**Claims Explanation:**

The method is compared on standard benchmarks with competitive baselines, but some contributions are not properly assessed.

**Requested Changes:**

## Mathematical clarity
While the paper is technically solid, the presentation suffers from unnecessary mathematical overhead that makes reading uneasy. I strongly recommend improving the notation and offloading details to the appendix. Below are some recommendations that would (in my opinion), improve the paper

1) Avoid using uncommon symbols, unless all other characters have already been used. For instance, $\Upsilon$, $\mathbf{W}$square and $\varkappa$ are not standard and could be readily replaced with $Y, W_i$ and $p_\text{size}$, making the equation easier to read.

2) Many symbols appear only once and add little to no value. I suggest removing them from the main paper to improve clarity and delaying all mathematical details in the appendix. Few examples:

	- Bounded domains: $\Omega_{d_a}$ and $\Omega_{d_u}$,
	- All matrix parameters in Equations (1) and (2). Equation (1) could be simplified as:

$Y = \sigma\big(K(Z) + \text{Conv1D}^0(Z))$

$\mathcal{L}(Z) = \text{ChannelMLP}^0(Y) + \text{SoftGating}^0(Z)$

which aligns with typical FNO implementation, is easy to read, and does not introduce unnecessary variables. The same remark applies to Equation (2).

3) There are several ambiguous or inconsistent definitions:
	- **Equation (2)**: I suppose $K_g$ and $\Upsilon_{\text{global}}$ are aliases for $\Upsilon$ and $\mathcal{K}$ from equation (1) ? If so, why changing notations? $\hat Z$ and $Z'$ are also not clearly introduced.
	- **Equation (3)** What does $a$ refer to ? It is used as global input to the network  $u = \mathcal{G}(a)$ and in Equation (3). Why not using $\hat Z$ ? What are $z_t^{(m)}$, $t$ and $m$ ? If they represent time and patch indices, state so explicitly.
	- **Next unlabeled equation**: Please add a label. Add a label. Symbols like $\mathcal{H}$, $X_S$, and $X_U$ are inconsistent with previous notation and absent from Equation (2) and Figure 1. Why changing the notation again?
	- **Algorithm 1**: The introduction of new symbols $\hat P$ and $\hat T$ ? and conflicting naming ($X$ and $Y$ to denote the spatial axis and $x$ and $y$ for the frequency grid) introduce confusion.

In particular, I strongly encourage the author to improve the readability and clarity of Equation (2), as it is one of the main contributions of the paper.

## Reorganize the paper

The paper, especially with its appendix, is overly long and would benefit greatly from a reorganization, since many important results are in the appendix. Here are my main suggestions:

**Move important content to the main paper:**

- **Figures 4–6** are critical to understanding the HF loss and should be integrated into the main text, ideally with a consolidated version for readability.
- **Baseline discussion** should be summarized early in the main paper, along with key distinctions from LOGLO-FNO.
- **Table 7**, showing performance breakdowns, is important enough to warrant inclusion in the main body.

Algorithm (1) and Equation (4) are difficult to parse and can be relegated to the appendix with a clearer summary or diagram in the main text. However, Figure 4-6 are critical for understanding the loss. A figure inspired from these plots should be integrated into the main paper.

Table 1 can be condensed to highlight the most informative metrics (e.g., RMSE, and fRMSE for L, M, H). The full table can stay in the appendix. Figure 4, 5 and 6 (right) could also be merged and displayed in the main paper for rapid evaluation of the difference between LOGLO-FNO and the baselines.  While extensively detailed in the appendix, the baselines and metrics should be quickly presented in the main paper, especially by stating the difference with LOGLO-FNO.

The related work section is complete to the best of my knowledge, however I would recommend improving the 3rd paragraph to not only list competitors but also motivate how LOGLO-FNO differs from existing methods.

Finally, the appendix can be cleaned and shorten by avoiding unnecessary white spaces, repeating figures from the main paper, and repeating table 5, 6, and 7 which have redundant lines. I also find Table 7 quite important, and would recommend moving it to the main paper. Section G.2 feels disconnected.

---

> ### Author Response · Authors · 2025-10-14
> **Rebuttal on improving mathematical clarity**
>
> We thank the reviewer **CTmE** for the thorough review, insightful questions, and constructive comments. We
> address each of the concerns, questions, and comments separately.
>
> **Mathematical clarity**
>
> > 1. Avoid using uncommon symbols, unless all other characters have already been used. For instance, $\Upsilon$, **W**square, and $\varkappa$ are not standard and could be readily replaced with Y, $W_i$, and $P_{size}$, making the equation easier to read.
>
> We thank the reviewer for pointing out the need to improve notations for easier readability. Following your suggestions, these symbols have been replaced with your proposed ones.
>
> > 2. Many symbols appear only once and add little to no value. I suggest removing them from the main paper to improve clarity and delaying all mathematical details in the appendix. Few examples:
> > - Bounded domains: $\Omega_{d_{a}}$ and $\Omega_{d_{u}}$
> > - All matrix parameters in Equations (1) and (2). Equation (1) could be simplified as:
> > Y = $\sigma(K(Z)$ + Conv1D$^0$(Z))
> > $\mathcal{L}(Z)$ = ChannelMLP$^0$(Y) + SoftGating$^0$(Z)
> > which aligns with typical FNO implementation, is easy to read, and does not introduce unnecessary variables. The same remark applies to Equation (2).
>
> Thank you for pointing out these changes in favor of simplification. In accordance with your suggestions, we made the modifications; however, we use superfix$^1$  in Equation 1 to be consistent and in congruence with the notations introduced above.
>
>
> > 3. There are several ambiguous or inconsistent definitions:
> > - **Equation (2)**: I suppose $K_g$ and $\Upsilon_{global}$ are aliases for $K$ and $\Upsilon$ from equation (1)? If so, why changing notations? $\hat{Z}$ and $Z'$ are also not clearly introduced.
> > - **Equation (3)**: What does $a$ refer to ? It is used as global input to the network $u = \mathcal{G}(a)$ and in Equation (3). Why not using $\hat{Z}$? What are $z_{t}^{(m)}$, $t$, and $m$? If they represent time and patch indices, state so explicitly.
> > - **Next unlabeled equation**: Please add a label. Add a label. Symbols like $\mathcal{H}$, $X_{S}$, and $X_{U}$ are inconsistent with previous notation and absent from Equation (2) and Figure 1. Why changing the notation again?
> > - **Algorithm 1**: The introduction of new symbols $\hat{P}$ and $\hat{T}$? and conflicting naming ($X$ and $Y$ to denote the spatial axis and $x$ and $y$ for the frequency grid) introduce confusion.
> > In particular, I strongly encourage the author to improve the readability and clarity of Equation (2), as it is one of the main contributions of the paper.
>
> We are thankful for your careful reading and pointing out these inconsistencies. We address these individually.
>
> - Equations (1) and (2) now use the same notation for the global branch, making it consistent. We have also defined $\hat{Z}$ as well as $Z'$, which depends on $\mathbf{X}_{H}$.
> - We have simplified the definition of the local kernel integral operator.
> - We have added the label for the unlabeled equation and removed the inconsistent notations. Following your thoughtful comment, we have also added the notations to the architecture diagram (see Figure 1).
> - The conflicting notations have been fixed in Algorithm 1 and moved to the Appendix following your suggestion.
> - Equation (2) is now improved with your suggestion.

---

> ### Author Response · Authors · 2025-10-14
> **Rebuttal on reorganizing the paper**
>
> > The paper, especially with its appendix, is overly long and would benefit greatly from a reorganization, since many important results are in the appendix. Here are my main suggestions:
> > **Move important content to the main paper**:
> > - **Figures 4-6** are critical to understanding the HF loss and should be integrated into the main text, ideally with a consolidated version for readability.
> > - **Baseline discussion** should be summarized early in the main paper, along with key distinctions from LOGLO-FNO.
> > - **Table 7**, showing performance breakdowns, is important enough to warrant inclusion in the main body.
>
> Thank you for your thoughtful comment. We re-organize the Figures, Sections, and Tables following your suggestions, with new contents highlighted in dark red.
>
> > Algorithm (1) and Equation (4) are difficult to parse and can be relegated to the appendix with a clearer summary or diagram in the main text. However, Figure 4-6 are critical for understanding the loss. A figure inspired from these plots should be integrated into the main paper.
>
> We thank the reviewer for this crucial suggestion. In response, we move the algorithm to the appendix and include a representative figure in the main text to convey the core idea of radial binning-based frequency loss. However, we decide to keep the equation in the main paper.
>
> > Table 1 can be condensed to highlight the most informative metrics (e.g., RMSE, and fRMSE for L, M, H). The full table can stay in the appendix. Figure 4, 5 and 6 (right) could also be merged and displayed in the main paper for rapid evaluation of the difference between LOGLO-FNO and the baselines. While extensively detailed in the appendix, the baselines and metrics should be quickly presented in the main paper, especially by stating the difference with LOGLO-FNO.
>
> Thank you for bringing out this point. We intended to give a full picture by assessing a variety of metrics, and, hence, included the entire suite of metrics. However, we agree that common metrics such as nRMSE and fRMSE(L, M, H) would be enough to gain a sufficient understanding of the effectiveness of the proposed method. Therefore, we formatted the tables accordingly in the main paper while placing the full table in the Appendix, as per your suggestion.
>
> > The related work section is complete to the best of my knowledge, however I would recommend improving the 3rd paragraph to not only list competitors but also motivate how LOGLO-FNO differs from existing methods.
>
> Thank you for pointing out this important detail. Following your suggestion, we now include this missing detail.
>
> > Finally, the appendix can be cleaned and shorten by avoiding unnecessary white spaces, repeating figures from the main paper, and repeating table 5, 6, and 7 which have redundant lines. I also find Table 7 quite important, and would recommend moving it to the main paper. Section G.2 feels disconnected.
>
> Thank you for bringing the formatting issues in the appendix to our attention, and the importance of Table 7. We follow your suggestion and reorganize the text. However, we would like to point out that Section G.2 is not disconnected since it has been referenced in the main text in Section 3. But we are open to removing this part if the reviewer strongly advises us to do so.

---

> ### Author Response · Authors · 2025-10-14
> **Rebuttal on Broader Impact Concerns**
>
> > **Training performance**: The authors discussed extensively the impact of their method on the number of parameters, but this should be completed by a training and inference time comparison, as I would expect that the additional terms and the frequency loss impacts greatly the efficiency. This is important to assess the balance between the added computational cost and the performance gain. The impact on GPU memory usage could also be very relevant.
>
> We agree with the reviewer's intuition and expectation that the additional branches incur additional memory overhead and longer training times. Due to time and compute constraints, we are still preparing these statistics and will provide them as soon as possible by updating the paper.
>
> > **Resolution invariance**: The authors propose to slice the input domain into patches to handle local features, akin to ViTs. Yet, this might impact the generalizability of the method to higher or lower resolutions, as we now ViTs does not necessarily generalize to resolution outside their training set. Moreover, zero-shot multi-resolution is the core motivation behind FNO, hence assessing that LOGLO-FNO maintains this feature is detrimental. Such evaluation is missing from the paper.
>
> Thank you for raising this important point. We agree with the reviewer that ViT is a neural network and not a neural operator. Given that ZSSR is one of the defining criteria of a neural operator and a core aspect of FNO, we conduct experiments on two datasets (*2D Kolmogorov Flow* and *2D CNS Turb*) to test this aspect of LOGLO-FNO and provide the results in the section **"Zero-Shot Super-Resolution (ZSSR) Capability and Performance of LOGLO-FNO"** in the Appendix. We observe that our model incurs some discretization errors, specifically on the spatial and energy spectra metrics. However, it is still superior to NO-LIDK variants, which also suffer from error propagation to higher spatial resolutions. Please note that the authors of NO-LIDK have excluded ZSSR experiments on the Kolmogorov Flow 2D dataset (see 4th paragraph of section C.6. in the NO-LIDK paper).
>
> In general, in our experience and in the literature, we find that ZSSR capabilities of neural operators have their limits (see Appendix "C.6. Zero-shot super-resolution results" in the NO-LIDK paper).
>
> > **Missing ablations**: Several architectural components are introduced but not validated. Missing ablations include:
> > - Patch-size sensitivity, especially given spectral resolution trade-offs (decreasing patch-size might improve the model on local features, but also decreases the number of modes per patches)
>
> Thank you for pointing this out. We have added the ablation study on "Patch-Size Sensitivity of LOGLO-FNO" and updated the paper with these results in the Appendix section K.4.
>
> Since we use parameters for all available Fourier modes in a patch, larger patch sizes can still handle the highest frequencies. Therefore, we find that larger patch sizes are usually slightly better, and choosing the patch size is a trade-off between accuracy and time+memory efficiency, as well as dependent on the nature and scale of interactions happening in the underlying fields of the PDE simulation problem under consideration.
>
> > - Adaptive Gaussian noise
>
> Thank you for your careful spotting. We have conducted an ablation study and updated the results in Appendix K.5 "**Influence of High-Frequency Adaptive Gaussian Noise on Rollout Stability in LOGLO-FNO**"
>
>
> > - High-frequency component in the model. It is unclear for me how this affect the model performances, since the additional filtered input remains high frequency, and thus difficult to learn with a neural network even if the low frequency component has been removed.
>
> While it is indeed difficult to say why exactly the high-frequency component helps, here is a hypothesis: The filtered high-frequencies input is passed through a channel MLP, i.e., pixels are processed independently, and therefore it remains high-frequency. The MLP could learn to remove certain parts of the high-frequency components while preserving others. Thanks to the high-frequency filter, this leaves the low-frequency components untouched, which are already well-processed by the regular FNO branch.
>
> > **Performances**: It is unclear why the author also supervise mid-range frequencies while the paper focuses on high-frequencies. In that case, why not supervising all frequency ranges ?
>
> We do supervise all frequency ranges: We use the frequency loss together with MSE. By Parseval's theorem, MSE weighs all frequencies equally in the loss, and the frequency loss then adds extra weight on the middle and high frequency bands. Empirically, we found this to be beneficial, even if the downstream metric is just RMSE. Of course, while the architectural modifications focus on improving high frequencies, the weakness of FNO, we want the resulting model to be good on all frequencies. We will add a sentence to the paper to clarify this.

---

> ### Author Response · Authors · 2025-10-14
> **Rebuttal on Broader Impact Concerns (contd.) and Additional Comments**
>
> > Moreover, according to Table (1), the gain in high frequency RMSE does not transfer well when increasing the prediction horizon.
>
> Table (1) presents results on the highly challenging Kolmogorov Flow (Re=5000), which is a fully turbulent flow presenting significant challenges for emulators. We agree that there is an uptick in high-frequency error, mainly due to the effect of autoregressive accumulation, which is mitigated but not entirely removed. However, it is still superior to other models compared.
>
> On the other hand, the improvements we observe for the 1-step results on Wave Equation 2D (see Table 8 in Appendix) are maintained over autoregressive rollouts, suggesting that this behaviour might be problem-specific.
>
> > **Additional Comments**:
> >     Based on the implementation of the FNO block [1], I'm not sure that equation (1) is correct. Using notation from equation (1), the implementation seems to be:
> > $Z' = W_f * Z + b_f $
> > $Y = \sigma(K(Z') + Z')$
> > Could the authors double-check this equation and ensure that it corresponds to the one implemented?
> > [1] https://github.com/neuraloperator/neuraloperator/blob/main/neuralop/layers/fno_block.py
>
> Thank you for your insightful comment. Our implementation is based on the codebase of the neural operator library, which incorporates the recent improvements made in enhancing FNO. And we use the implementation by enabling the channel MLP, which corresponds to Equation 1, and is depicted accordingly in Figure 1.

---

> ### Comment · Reviewer_CTmE · 2025-10-15
> **Reply to rebuttal**
>
> I would like to thank the authors for their work on the revised version of the manuscript. I have carefully read the updated paper, the rebuttal, and the other reviewers’ comments.
>
> In my opinion, the paper has clearly improved in both clarity and readability. I appreciate the careful rewriting of the methodology section, which is now much easier to follow. The additional ablation studies and experiments are also valuable contributions:
> - I understand that evaluating the impact of LOGLO-FNO contributions on training time may be time-consuming, but I look forward to seeing these results. Although LOGLO-FNO will likely introduce computational overhead during training, a quantitative assessment of this overhead would provide useful insights, and would help to estimate the trade-off between the additional complexity and the metrics improvement.
> - The ZSSR experiment is particularly interesting. While a decrease in performance with increasing resolution was expected, the metrics remain within the same order of magnitude. More importantly, the relative drop in performance is comparable to that observed for the baseline FNO, which supports the authors’ claims of robustness.
> - However, I am less convinced by the patch size sweep study. It is somewhat surprising that the patch size appears to have little impact on performance metrics. This raises questions regarding the precise role of this parameter in the proposed model.
>
>
> In conclusion, I consider that the authors have satisfactorily addressed most of my previous concerns and responded constructively to the requests for clarification and improvement. I will not engage further in the discussion but will carefully read the other reviewers’ replies. The paper is now easy to follow and presents a simple but effective idea, which is thoroughly tested through an extensive set of experiments. Overall, the paper is scientifically sound, and I am inclined to recommend acceptance. However, the contribution remains somewhat limited in terms of novelty: penalizing high-frequency terms in the loss is not a fundamentally new concept, and while the additional architectural branches compared to FNO improve performance, they do so at the cost of longer training time and greater model complexity.

---

> > ### Author Response · Authors · 2025-10-21
> > **Rebuttal with missing results on runtime and memory usage benchmarking**
> >
> > We sincerely thank the reviewer **CTmE** for promptly reviewing our revision and providing feedback, and a positive assessment of our work.
> >
> > > I understand that evaluating the impact of LOGLO-FNO contributions on training time may be time-consuming, but I look forward to seeing these results. Although LOGLO-FNO will likely introduce computational overhead during training, a quantitative assessment of this overhead would provide useful insights, and would help to estimate the trade-off between the additional complexity and the metrics improvement.
> >
> > Thank you for your patience. We have now added these results to the section **"M.2 Training & Inference Time and GPU Memory Usage Comparison with Base FNO and NO-LIDK"** in the Appendix.
> >
> > As we observe from the benchmarking results, we can conclude that LOGLO-FNO results in increased training time and memory usage. So does the NO-LIDK$^{\ast}$ (with local integral kernel) model, which is slower than LOGLO-FNO. In our experience, we find the baseline FNO as the most lightweight NO model and a hard-to-beat benchmark.
> >
> > However, we would like to draw attention to the fact that our model allows the flexibility to lower the number of modes on the global branch, which would then reduce the parameter counts, thereby accelerating the training time and reducing memory usage. In addition, a multi-resolution training strategy could be employed to accelerate training -- the global branch could be trained at a coarser resolution (i.e., forward FFT; although the inverse FFT still has to resolve to the fine-scale resolution for feature fusion with the local branch output) for high-level features, while the local branch can process full resolution to model fine-scale and intricate structures. Other viable options for reduced GPU memory usage include simulating larger batch sizes (e.g., 256) using gradient accumulation strategies. Moreover, efforts could also be directed towards custom GPU kernel implementations for architecture-aware optimizations, such as in [1]. All these aspects could be taken up in future work.
> >
> >
> >
> > > However, I am less convinced by the patch size sweep study. It is somewhat surprising that the patch size appears to have little impact on performance metrics. This raises questions regarding the precise role of this parameter in the proposed model.
> >
> > We have conducted a new study for the patch sensitivity analysis on the 2D CE-RP and provided the results in **"K.4 Patch-Size Sensitivity of LOGLO-FNO"**. We observe that the increase in patch size beyond a certain point is rather detrimental for extended rollout scenarios.
> >
> >
> > > In conclusion, I consider that the authors have satisfactorily addressed most of my previous concerns and responded constructively to the requests for clarification and improvement. I will not engage further in the discussion but will carefully read the other reviewers’ replies. The paper is now easy to follow and presents a simple but effective idea, which is thoroughly tested through an extensive set of experiments. Overall, the paper is scientifically sound, and I am inclined to recommend acceptance.
> >
> > We again thank the reviewer for the time and effort spent reviewing and providing constructive feedback, which has been helpful for improving our submission.
> >
> >
> > --------------------
> >
> > [1] TurboFNO: High-Performance Fourier Neural Operator with Fused FFT-GEMM-iFFT on GPU, https://arxiv.org/abs/2504.11681

---

### Review · Reviewer_UKSR · 2025-09-06

**Summary Of Contributions:**

The authors introduce LOGLO-FNO, an enhanced Fourier Neural Operator designed to overcome the common neural network issue of "spectral bias," where models struggle to capture high-frequency details crucial for complex physical simulations like turbulence. The authors' solution integrates three key architectural enhancements: a parallel branch for **local spectral convolutions** to learn fine-grained features from data patches, a **high-frequency propagation module** to ensure critical details aren't lost, and a novel **frequency-aware loss function** that specifically penalizes errors in non-dominant frequencies during training. However, these enhancements result in a model that is not only more accurate and stable over long-term predictions but is also significantly more parameter-efficient, achieving the accuracy of baseline models with up to 50% fewer parameters. The primary limitation noted by the authors is the intentional exclusion of advanced parameter-efficiency techniques for the sake of a fair comparison, suggesting a clear direction for future improvements. Overall its cool work but I have some concerns which I will talk about it more in the following sections.

**Additional Comments:**

N/A

**Audience:**

Yes

**Audience Explanation:**

I definitely think TMLR's audience would find this paper quite interesting and valuable. The neural operator community has been growing rapidly, especially with applications in scientific computing where traditional numerical methods are becoming computationally prohibitive for high-resolution simulations. The spectral bias problem that the authors address is a well-known issue that many researchers working with FNOs have encountered - anyone who has tried to model turbulent flows or high-frequency phenomena with neural operators has probably struggled with this exact limitation.

**Claims And Evidence:**

Yes

**Claims Explanation:**

Personally for me the authors have done a solid job supporting their claims with convincing evidence. The core assertion that FNOs struggle with high-frequency features due to spectral bias is well-established, and their proposed LOGLO-FNO architecture provides a reasonable solution. What I find particularly compelling is how they've structured their evaluation - they don't just show overall improvements, but specifically target frequency-domain metrics that directly relate to their main claims. The 11-25% improvements in high-frequency error reconstruction across three different PDE problems (Kolmogorov flow, diffusion-reaction, and turbulent radiative mixing) demonstrate consistency that goes beyond cherry-picking results. I appreciate that they've included comprehensive baselines including recent state-of-the-art methods like NO-LIDK, and the parameter efficiency gains (achieving better accuracy with 50% fewer parameters) make the contribution practically valuable. The experimental design shows good scientific rigor with proper ablation studies, energy spectra visualizations that provide intuitive understanding of the improvements, and thorough analysis of autoregressive stability. The mathematical framework appears sound, and I think the frequency-aware loss function is a clever addition that directly addresses the spectral bias problem. Thanks for the detailed ablations and studies in the Appendix!

However, there are some notable limitations that weaken the overall contribution. The evaluation scope is quite narrow - only three PDE problems, all from similar domains (fluid dynamics and related physics). While the authors claim generalizability, testing on a broader range of PDEs, especially those from different scientific domains like electromagnetics, quantum mechanics, or materials science, would strengthen their claims significantly. The choice of patch sizes and architectural decisions also seems somewhat ad-hoc - for instance, using 8×8 patches for diffusion-reaction but 16×16 for Kolmogorov flow due to "memory limitations" suggests the method may not scale as elegantly as claimed. Additionally, the comparison with some baselines isn't entirely fair since they disable tensor factorization techniques in FNO "for fair comparison with NO-LIDK," which artificially handicaps what could be stronger baselines.

The theoretical analysis could also be deeper - while they provide parameter complexity analysis, there's limited theoretical insight into why their local-global decomposition should work better than alternatives, or under what conditions we might expect it to fail. The frequency loss design, while intuitive, lacks principled justification for the specific binning strategy and weighting choices. Some experimental choices also raise questions - the different training epochs across datasets (50 for TRL3D vs 136 for Kolmogorov) and the model-specific hyperparameter tuning suggest the method may require careful dataset-specific optimization rather than being a robust drop-in replacement for standard FNOs.

**Requested Changes:**

Again some changes and thoughts that I had:

• Expand experimental evaluation beyond 3 PDEs - The current evaluation is too narrow with only Kolmogorov flow, diffusion-reaction, and turbulent radiative mixing. Include at least 2-3 additional PDE types from different domains (e.g., electromagnetics, wave equations, or heat conduction) to demonstrate true generalizability of the approach.

• Address inconsistent experimental settings - Different training epochs across datasets (50 for TRL3D, 136 for Kolmogorov, 500 for diffusion-reaction) and varying patch sizes (8×8 vs 16×16) due to "memory limitations" suggest the method may not be as robust as claimed. Provide systematic study of these choices or standardize the experimental protocol.

Some thoughts on what would strengthen this work:

• Improve frequency band justification - The radial binning strategy (low=[1,4], mid=[5,12], high=[13,M]) in Algorithm 1 appears arbitrary without theoretical or empirical justification for these specific cutoffs.

• Add statistical significance analysis - Results show consistent improvements but lack confidence intervals or significance tests to demonstrate improvements are statistically meaningful beyond empirical observation. This is fine if you cannot do in the limited time, but good for the full paper: )!

• Add failure case discussion - Paper only shows successful cases. Including examples where method doesn't improve over baselines and analyzing why would provide balanced perspective. Like i am curious where your method did indeed fail in your opinion as well as how well does your method scale?

Overall good job!

---

> ### Author Response · Authors · 2025-10-14
> **Rebuttal on Requested Changes**
>
> We thank the reviewer **UKSR** for the positive review, thoughtful suggestions for improvement, and for noticing the importance of our work.
>
> **Requested Changes:**
> > Again some changes and thoughts that I had:
> > Expand experimental evaluation beyond 3 PDEs - The current evaluation is too narrow with only Kolmogorov flow, diffusion-reaction, and turbulent radiative mixing. Include at least 2-3 additional PDE types from different domains (e.g., electromagnetics, wave equations, or heat conduction) to demonstrate true generalizability of the approach.
>
> Thank you for the suggestion to expand the scope of PDE problems. In response, we conduct experiments on the
> - 2D wave equation (Wave-Gauss) from Poseidon (Herde et al., 2024),
> - 2D Compressible Navier-Stokes from PDEBench, and
> - 2D Compressible Euler four-quadrant Riemann problem (Poseidon), [results with important baselines]
>
> and provide results in the revision. Please see the abridged results in the main paper and the full results in Appendix **G.2.1**.
>
> > - The choice of patch sizes and architectural decisions also seems somewhat ad-hoc - for instance, using 8×8 patches for diffusion-reaction but 16×16 for Kolmogorov flow due to "memory limitations" suggests the method may not scale as elegantly as claimed.
> > - Address inconsistent experimental settings - Different training epochs across datasets (50 for TRL3D, 136 for Kolmogorov, 500 for diffusion-reaction) and varying patch sizes (8×8 vs 16×16) due to "memory limitations" suggest the method may not be as robust as claimed. Provide systematic study of these choices or standardize the experimental protocol.
>
> Thank you for raising this point. The choice of the number of epochs for the Diffusion-Reaction and Kolmogorov Flow 2D was directly borrowed from NO-LIDK (please see **C.2** and **C.3** in [1]), and hence is in line with their setup. In the case of TRL3D, as we mentioned in the main paper, we experienced overfitting issues due to the limited amount of data from the Well dataset by Ohana et al. (2024). Moreover, and most importantly, the number of training epochs for a given PDE remains the *same across all baseline models* and ours, which we argue is a fair comparison.
>
> As for the patch sizes, we have now conducted an ablation study and updated results to "Patch-Size Sensitivity of LOGLO-FNO" in Appendix K.4. Further, we would like to add that autoregressive training on long trajectories (e.g., 90 timesteps) for dense prediction tasks on reasonably high spatial resolutions (e.g., 128x128) multi-channel data is, in general, slow and memory-intensive since the whole batch of trajectories has to be kept in GPU memory for backprop through time.
>
> > The frequency loss design, while intuitive, lacks principled justification for the specific binning strategy and weighting choices.
> > Improve frequency band justification - The radial binning strategy (low=[1,4], mid=[5,12], high=[13,M]) in Algorithm 1 appears arbitrary without theoretical or empirical justification for these specific cutoffs.
>
> The reviewer correctly notes that the specific cutoffs for our frequency bands require justification. While the numerical values are a heuristic, the underlying partitioning strategy is not arbitrary; it is a principled approach designed to mirror the distinct physical regimes of the turbulent energy cascade stemming from the fundamental physics of many fluid systems, particularly those exhibiting turbulence.  When considering Kolmogorov turbulence, the Fourier regime can be split into injection (low), cascade/inertia (mid), and dissipation (high) regimes, and the band classification implicitly considers this.
>
> We agree that the manuscript would benefit from a more thorough justification of this choice. We're running an ablation study and will provide the results as early as possible, as well as update the paper accordingly.
>
> > Add statistical significance analysis - Results show consistent improvements but lack confidence intervals or significance tests to demonstrate improvements are statistically meaningful beyond empirical observation. This is fine if you cannot do in the limited time, but good for the full paper: )!
>
> Thank you for your comment on this aspect. Due to the limited computational resources and budget, as well as in accordance with prior work in the literature (e.g., NO-LIDK), we are unable to perform sweeps over multiple random seeds for all baseline models and problems. Therefore, we could provide the standard deviations over three runs for base FNO, NO-LIDK, and LOGLO-FNO on the Kolmogorov Flow 2D problem.
>
> As a general comment, we'd like to add that we observed the loss evolution of LOGLO-FNO to be very well-behaved for the provided training settings and hyperparameters, as in the case of Base FNO and NO-LIDK, and, therefore, we do not expect a considerable variance.
>
> ---------
>
> [1] Neural Operators with Localized Integral and Differential Kernels, https://arxiv.org/abs/2402.16845

---

> > ### Author Response · Authors · 2025-10-21
> > **Rebuttal on Statistical Significance Analysis and Frequency Band Classification Justification**
> >
> > We are thankful to the reviewer **UKSR** for patiently awaiting the missing results.
> >
> >
> > > Add statistical significance analysis - Results show consistent improvements but lack confidence intervals or significance tests to demonstrate improvements are statistically meaningful beyond empirical observation. This is fine if you cannot do in the limited time, but good for the full paper: )!
> >
> > We have now added results for the confidence analysis on the Kolmogorov Flow 2D PDE in the Appendix section **"N. Confidence Analysis of LOGLO-FNO and Baselines"**, with the new contents highlighted in dark red as in the previous revision.
> >
> > > Improve frequency band justification - The radial binning strategy (low=[1,4], mid=[5,12], high=[13,M]) in Algorithm 1 appears arbitrary without theoretical or empirical justification for these specific cutoffs.
> >
> > Considering your thoughtful suggestion, we have conducted an ablation study, varying the cutoff boundaries in both directions. The results of this experiment can be found in the Appendix section **"K.6 Choice of Radial Bin Cutoff Values for the Radially Binned Frequency Loss in LOGLO-FNO"**, which is also referenced in the main paper (please see §3.2).
> >
> > > Expand experimental evaluation beyond 3 PDEs - The current evaluation is too narrow with only Kolmogorov flow, diffusion-reaction, and turbulent radiative mixing. Include at least 2-3 additional PDE types from different domains (e.g., electromagnetics, wave equations, or heat conduction) to demonstrate true generalizability of the approach.
> >
> > To complete the evaluation on the Compressible Euler Riemann Problem 2D, we have now also added the AR rollout results, which can be found in Appendix **G.2.1**. On counting this, we are able to demonstrate the effectiveness and generalization ability of our method on 6 PDE problems in total.
> >
> > We thank you again for your efforts, careful reading, and thoughtful suggestions, which have all helped to improve our submission.

---

> > > ### Comment · Reviewer_UKSR · 2025-10-21
> > > **Final Response**
> > >
> > > I thank the authors for a detailed response and for answering all my questions! The paper is technically sound, clearly improved by the rebuttal, and of high interest to the Neural Operator community. With the edits above, I’m happy to recommend accept. Great work!

---

> ### Author Response · Authors · 2025-10-14
> **Rebuttal on Requested Changes (contd.) and other comments.**
>
> > Add failure case discussion - Paper only shows successful cases. Including examples where method doesn't improve over baselines and analyzing why would provide balanced perspective. Like i am curious where your method did indeed fail in your opinion as well as how well does your method scale?
>
> Thank you for raising this point. The LOGLO-FNO method is *primarily* targeted towards PDE problems exhibiting turbulence, high-frequency structures, and rich local and multi-scale textures emanating from shock waves. Accordingly, we restricted ourselves to the demonstrated PDEs, including the newly added ones. Hence, choosing an appropriate problem that satisfies the aforementioned criteria is crucial to demonstrating the effectiveness of LOGLO-FNO over base FNO. Therefore, our model would likely not outperform the baseline FNO in the absence of high-frequency features.
>
> If you think this should be discussed in the paper, we can add it to the conclusion.
>
> As for scaling, as we have demonstrated in our experiments in Section G.4. "Significance of Adding More Modes in the Fourier Layers of Global Branch", our method scales well and yields consistent improvement as more frequency modes are added to the global branch (which directly results in more trainable parameters and increased model expressivity).
>
> > Additionally, the comparison with some baselines isn't entirely fair since they disable tensor factorization techniques in FNO "for fair comparison with NO-LIDK," which artificially handicaps what could be stronger baselines.
>
> Thank you for raising this point. In terms of the experimental setup, we aimed to be as close as possible to NO-LIDK since it also addresses the same problem of modeling local features in FNO, and is the closest related work to ours. The choice of disabling the factorization of spectral convolutions was made after clarifying with the authors of NO-LIDK.
>
> > The theoretical analysis could also be deeper - while they provide parameter complexity analysis, there's limited theoretical insight into why their local-global decomposition should work better than alternatives, or under what conditions we might expect it to fail.
>
> Thank you for bringing this up. We agree that a theoretical analysis would provide valuable insights and establish a guarantee on the expected performance gains of LOGLO-FNO. However, we would like to clarify that this aspect is beyond the scope of our current work and could be taken up in a future research effort.

---

### Review · Reviewer_143c · 2025-09-20

**Summary Of Contributions:**

This paper attempts to preserve high frequency content of the data and the solution in a Fourier Neural Operator. The authors introduce (i) a local convolution operation on smaller subdomains which helps capturing the local features (small scale or high frequency information), and (ii) a high-frequency propagator, i.e., a blurring and reconstruction process and then calculating the error introduced in that operation. In addition, the loss function is enhanced to capture the errors introduced by the truncation step of FNOs. Finally, the authors test their method on 2D Kolmogorov flow, 3D turbulent radiative mixing layer and a 2D diffusion-reaction problem. The methods produces a marginally lower error on these problems with respect to the ground truth.

**Audience:**

Yes

**Audience Explanation:**

The paper deals with retaining the high frequency contents of the data and the solution in a neural operator. The authors apply a local convolution operator and an artificial high-frequency error propagator to improve the high frequency errors in the final prediction. This is of great importance in the scientific machine learning research.

**Claims And Evidence:**

Yes

**Claims Explanation:**

* The paper is well-written, and is a good attempt at solving a genuine issue with FNOs
* The authors use two simple ideas from the literature and apply them to improve FNOs
* The ideas are discussed in detail
* Numerical results are presented supporting the claims.
* The paper is tightly focused toward a limited set of equations, but does a good job delivering on that.

**Requested Changes:**

Weaknesses:

* The applications are limited to a chosen few examples. However, I do not necessarily think this is a weakness.
* A pictorial representation of the high-frequence propagator would be helpful.
* Many of the metrics are non-standard. The authors have provided reference for all the metrics, and also listed them in Appendix G. However, not all metrics are listed. A complete list of formulas would be appreciated.
* The improvements in errors (as per Table 1, 2 and 3) are only marginal.
* How do the results compare with traditional numerical results? Is it possible to illustrate that with a standard problem?

Questions:

* I imagine that the local convolutions introduce a "zooming in" effect. Different convolution applied to subdomains rather than the full domain, helps in capturing the local features. Do the authors agree with this interpretation?
* Did the authors consider or try problems with shock features?
* I commend the authors on this method. Still, what is special in this method that warrants its application, especially considering that the gains are only marginal?

---

> ### Author Response · Authors · 2025-10-14
> **Rebuttal on Requested Changes**
>
> We thank the reviewer **143c** for the positive review, insightful questions, and for noticing the importance of our work on improving the modeling capabilities of FNO. We address them below.
>
> > **weaknesses:**
> > - The applications are limited to a chosen few examples. However, I do not necessarily think this is a weakness.
>
> Thank you for raising an apropos point and bringing this to our notice. We have added additional PDE problems on the 2D Wave Equation, 2D Compressible Navier-Stokes, and preliminary results on the 2D Compressible Euler four-quadrant Riemann problem, and updated the paper with the results highlighted in dark red.
>
> > A pictorial representation of the high-frequency propagator would be helpful.
>
> In the revised paper, we have explicitly marked the extracted high-frequency features **X**$_H$ in the architecture diagram (please see Figure 1). If you think a slightly bigger figure is better, we can embed this in Section 3.1, where the HFP branch is introduced.
>
> > Many of the metrics are non-standard. The authors have provided reference for all the metrics, and also listed them in Appendix G. However, not all metrics are listed. A complete list of formulas would be appreciated.
>
> We thank the reviewer for raising this point. As per your suggestion, we have updated Appendix G.1 with the formulas.
>
> > The improvements in errors (as per Table 1, 2 and 3) are only marginal.
>
> We would like to point out that our method yields consistent gains across all metrics on a variety of problems. As indicated in the main paper, we have improved the reported results of Ohana et al. (2024), of the baseline FNO on the TRL3D dataset by more than 40\%. Therefore, the baselines are very strong. In addition, tremendous efforts have been made to ensure the baselines are strong by tuning all the baseline models, such as retraining the models using AdamW optimizers and tuning learning rates.
>
> > How do the results compare with traditional numerical results? Is it possible to illustrate that with a standard problem?
>
> Since we train neural operators, the label itself comes from the numerical solver. Therefore, it might not be easy to measure the error of a traditional numerical solver fairly.
>
> However, we can measure the inference time and compare it with the time needed to generate solutions by the numerical solver. We have provided this for the Turbulent Radiative Mixing Layer 3D dataset (see Appendix **M.1 Comparison with Numerical Solver**) and compared the time reported by the Well dataset. We observe that the neural surrogate is 54,363 times faster than the Athena++ numerical solver. When also considering the one-time training time of 12 hours, the neural surrogate is still 2,735 times faster than the numerical solver.
>
> --------------------
>
> **Questions**:
>
> > I imagine that the local convolutions introduce a "zooming in" effect. Different convolution applied to subdomains rather than the full domain, helps in capturing the local features. Do the authors agree with this interpretation?
>
> Yes, the "zooming in" might be an apt metaphor for the effect of our method.
>
> > Did the authors consider or try problems with shock features?
>
> Thank you for your insightful and pertinent question. We have now added the 2D Compressible Navier-Stokes from PDEBench, specifically the sonic version (Mach=1.0), where the initial conditions comprise turbulent velocity. The CNS equations give rise to the formation and propagation of shock waves. Furthermore, we added preliminary results on the Compressible Euler four-quadrant Riemann problem 2D from the Poseidon project (https://camlab-ethz.github.io/poseidon/). This is a classic problem generalizing the Sod shock tube to 2D space. Please see the results in Table 9 (highlighted in dark red) in the updated paper.

---

> ### Author Response · Authors · 2025-10-14
> **Rebuttal on Requested Changes (contd.)**
>
> > I commend the authors on this method. Still, what is special in this method that warrants its application, especially considering that the gains are only marginal?
>
> LOGLO-FNO achieves stable AR rollouts, is parameter efficient, mitigates spectral bias to a considerable extent, and aids in better modeling of high frequencies that are prevalent in turbulence phenomena and shock waves.
>
> Moreover, LOGLO-FNO naturally extends to 3D as we demonstrated in TRL-3D, making it the only model that enhances FNO with local convolutions for 3D spatial data. The neural operator does **not** yet support DISCO convolutions (i.e., local integral kernel layers) for 3D (cf. [1]). Furthermore, we find in our experiments that the radius cutoff hyperparameter of DISCO convolutions in NO-LIDK greatly affects the training time and accuracy of the model and needs to be tuned for each PDE problem (see sections C.1, C.2, and C.5 in the Appendix of NO-LIDK paper [3]). The default value derived from the suggested heuristic of *spatial\_domain\_length / \#output\_points* (cf. [2]) may not always be the optimal choice. On the contrary, LOGLO-FNO has only the patch size as the hyperparameter for the local branch, for which we find 16x16x16 to be a good enough value.
>
> Furthermore, the ZSSR capabilities of NO-LIDK remain limited as noted by the authors in Appendix "C.6. ZSSR results" (cf. paragraphs 3 and 4). Stating these reasons, the authors exclude the ZSSR tests on the challenging Kolmogorov Flow (Re5000) PDE, which we also use in our experiments. Please see our ZSSR results in Appendix L "Zero-Shot Super-Resolution (ZSSR) Capability and Performance of LOGLO-FNO", which demonstrates the superior results over NO-LIDK on two PDE problems.
>
> Lastly, the proposed frequency losses are architecture agnostic and can be applied to any neural operator training, as long as the data is on a uniform discretization grid, which is mandatory since it makes use of FFTs.
>
>
> -------------------------
>
> \[1\]: https://github.com/neuraloperator/neuraloperator/blob/main/neuralop/layers/local_no_block.py#L186-L187
>
> \[2\]: https://github.com/neuraloperator/neuraloperator/blob/main/neuralop/layers/discrete_continuous_convolution.py#L711
>
> [3]: Neural Operators with Localized Integral and Differential Kernels, https://arxiv.org/abs/2402.16845

---

> ### Author Response · Authors · 2025-10-21
> **Rebuttal on Problems with Shock Features**
>
> We thank the reviewer **143c** for patiently waiting for the missing results.
>
> > The applications are limited to a chosen few examples. However, I do not necessarily think this is a weakness.
> We have now expanded the problems to a total of 6. We hope this addresses this point.
>
> > Did the authors consider or try problems with shock features?
>
> To complement the preliminary 1-step results on the Compressible Euler four-quadrant Riemann Problem, we have now completed the rollout evaluations and updated the results in the revised version in the Appendix section **G.2.1** (page 44). If you'd like to see, we are also very much open to including qualitative examples.
>
> Thank you for your efforts in reviewing our work and insightful questions. These have been helpful for improving our paper.

---

### Author Response · Authors · 2025-12-16
**Camera ready revision**

We sincerely thank the reviewers for the constructive feedback, thoughtful suggestions, and insightful questions. Your efforts and comments were instrumental in improving the quality of our work, and we have incorporated them into the camera-ready version. We also thank the Action Editor for guiding the review process.

Best regards and happy winter holidays!

---

### Decision · Action_Editor_k88Y · 2025-11-14

**Recommendation:** Accept as is

**Audience:**

Yes

**Audience Explanation:**

All reviewers agreed that the problem of preserving high-frequency information in FNOs is a relevant problem for the community. The architectural and training improvements are interesting and novel in this context.

**Claims And Evidence:**

Yes

**Claims Explanation:**

All reviewers agree that the paper provides enough experimental evidence to support their architectural and training improvements to FNOs with the additional experiments provided during the rebuttal period further strengthening the claims.